# Greenland temperature and precipitation over the last 20,000 years using data assimilation

Jessica A. Badgeley[1], Eric J. Steig[1,2], Gregory J. Hakim[2], and Tyler J. Fudge[1]

[1]Department of Earth and Space Sciences, University of Washington
[2]Department of Atmospheric Sciences, University of Washington

**Correspondence:** Jessica Badgeley (badgeley@uw.edu)

**Abstract.** Reconstructions of past temperature and precipitation are fundamental to modeling the Greenland Ice Sheet and assessing its sensitivity to climate. Paleoclimate information is sourced from proxy records and climate-model simulations; however, the former are spatially incomplete while the latter are sensitive to model dynamics and boundary conditions. Efforts to combine these sources of information to reconstruct spatial patterns of Greenland climate over glacial-interglacial cycles have been limited by assumptions of fixed spatial patterns and a restricted use of proxy data. We avoid these limitations by using paleoclimate data assimilation to create independent reconstructions of mean-annual temperature and precipitation for the last 20,000 years. Our method uses oxygen-isotope ratios of ice and accumulation rates from long ice-core records and extends this information to all locations across Greenland using spatial relationships derived from a transient climate-model simulation. Standard evaluation metrics for this method show that our results capture climate at locations without ice-core records. Our results differ from previous work in the reconstructed spatial pattern of temperature change during abrupt climate transitions; this indicates a need for additional proxy data and additional transient climate-model simulations. We investigate the relationship between precipitation and temperature, finding that it is frequency dependent and spatially variable, suggesting that thermodynamic scaling methods commonly used in ice-sheet modeling are overly simplistic. Our results demonstrate that paleoclimate data assimilation is a useful tool for reconstructing the spatial and temporal patterns of past climate on timescales relevant to ice sheets.

## 1 Introduction

Predicting the future behavior of the Greenland Ice Sheet requires understanding its sensitivity to changes in temperature and precipitation (Bindschadler et al., 2013). One important constraint on this sensitivity is the response of the paleo ice-sheet to changing climate in the past. On glacial-interglacial timescales, temperature, not precipitation, appears to be the dominant control on the size of the Greenland Ice Sheet (Alley et al., 2010), as evidenced by the fact that the ice sheet is largest during cold and arid glacial periods and smallest during warm and wet interglacials. On these timescales, precipitation over the Greenland Ice Sheet scales positively with temperature (Robin, 1977), as anticipated by the Clausius-Clapeyron relation between temperature and saturation vapor pressure. Ice-core records, however, show that this thermodynamic relation is a poor approximation on annual to multi-millennial timescales (Kapsner et al., 1995; Fudge et al., 2016). For example, the

GISP2 ice core from central Greenland shows that cooling coincided with increased snowfall between the early Holocene and present (Cuffey and Clow, 1997). Despite such evidence, paleo ice-sheet models typically assume precipitation fields that are parameterized in time using a thermodynamic relationship that is constant for all locations and timescales (e.g., Huybrechts, 2002; Greve et al., 2011).

Ice-core records provide the best empirical estimates of climate history over the Greenland Ice Sheet. For temperature, important proxies include oxygen and hydrogen isotopes of ice (e.g., Jouzel et al., 1997), nitrogen isotope ratios of gas trapped in ice (e.g., Severinghaus et al., 1998; Severinghaus and Brook, 1999), and borehole thermometry (e.g., Cuffey et al., 1995; Dahl-Jensen et al., 1998). For precipitation, the thickness of annual layers of accumulated ice, corrected for thinning, is used (e.g., Dahl-Jensen et al., 1993). Ice-core records, however, are limited in their spatial coverage. In contrast, climate-model simulations are spatially-complete estimates of past climate, but they are subject to uncertainty due to model dynamics, boundary conditions, and spatial resolution.

Efforts to combine information from proxy data and climate models have long been a part of ice-sheet modeling. The most common approach is to scale the modern spatial pattern of temperature and precipitation using data from a single ice core (e.g, Huybrechts et al., 1991; Huybrechts, 2002; Greve, 1997; Greve et al., 2011; Nielsen et al., 2018). This assumes a fixed spatial pattern through time, which is unlikely to be valid. Recently, Buizert et al. (2018) used the average of the three best-understood Greenland ice-core records to adjust the results of a transient climate-model simulation (the transient climate evolution experiment, TraCE-21ka; Liu et al., 2009; He et al., 2013). This approach allows for possible changes in spatial relationships, but focuses on ice cores in central and northern Greenland and provides no information on precipitation. Other attempts to incorporate more proxy data have been limited to short time periods (e.g., Simpson et al., 2009; Lecavalier et al., 2014).

In this study we apply paleoclimate data assimilation to obtain a new, spatially-complete reconstruction of Greenland temperature and precipitation. We focus on the last 20,000 years, which includes the end of the last glacial period, the glacial to interglacial transition, and the Holocene thermal maximum (HTM), when temperatures at the Greenland Ice Sheet summit reached 1-2 °C warmer than present (Cuffey and Clow, 1997; Dahl-Jensen et al., 1998). The climate history, and the corresponding changes in the size of the ice sheet, are well-documented over this time period (e.g., Kaufman et al., 2004; Young and Briner, 2015).

Paleoclimate data assimilation combines spatial information from a climate-model simulation and temporal data from proxy records to produce a climate "reanalysis", where the term is taken from the modern climate reanalysis methods on which the data assimilation method is based (e.g., Kalnay et al., 1996). We adapt the paleoclimate data assimilation framework developed by Hakim et al. (2016), who reconstructed annual two-meter air temperature and 500 hPa geopotential height over the last millennium using a global network of temperature and precipitation-sensitive proxy records. Here, we use oxygen isotopes of ice and layer thickness from ice cores to reconstruct temperature and precipitation, respectively. We choose these proxies for their high temporal resolution, direct relationships to climate over the ice sheet, and availability from multiple ice cores. For the climate-model simulation, we use TraCE-21ka (Liu et al., 2009; He et al., 2013), which was run with the Community Climate System Model version 3 (CCSM3) to simulate the last 22,000 years. We compare the resulting reanalysis to previously-

published climate reconstructions (Sects. 3.1, 4.2, and 4.3), and assess the precipitation-temperature relationship (Sect. 4.1). We evaluate the reanalysis with independent proxy records and sensitivity tests (Sects. 3.2 and 3.3).

## 2 Methods

Our paleoclimate reconstruction method assimilates oxygen isotope ratios and accumulation from ice cores with a transient climate-model simulation to reconstruct the last 20,000 years of Greenland temperature and precipitation. In the following subsections we discuss the ice-core data, the climate-model simulation, and the details of our paleoclimate data assimilation approach.

### 2.1 Ice-core data

We use proxy records from eight ice cores from the Greenland Ice Sheet and nearby ice caps (Fig. 1, Table 1). As a proxy for temperature, we use previously-published measurements of oxygen isotope ratios from the ice, which we discuss using the conventional $\delta^{18}$O nomenclature (Dansgaard, 1964). We note that while other proxies (such as borehole thermometry or $\delta^{15}$N of $N_2$) have been used to produce temperature estimates (e.g., Cuffey et al., 1995; Dahl-Jensen et al., 1998; Severinghaus et al., 1998; Severinghaus and Brook, 1999), they are not available for many of the core locations; we instead rely on such data to obtain independent estimates of error in the $\delta^{18}$O-temperature relationship (see Sect. 2.3.3) and as comparisons to our resulting reanalysis (see Sect. 3). The accumulation history has been estimated for five of these cores from layer thickness corrected for vertical ice-thinning due to dynamical strain in the ice sheet. We rely on previously-published accumulation histories for the GISP2 and NEEM cores (Cuffey and Clow, 1997; Rasmussen et al., 2013), and we estimate accumulation for the Dye3, GRIP, and NGRIP cores using available layer-thickness data and simple ice-thinning calculations (see below and Sect. S2). We do not use accumulation records from the Agassiz, Camp Century, or Renland cores because the ice-thinning history at these sites is not adequately constrained. Most of the ice-core data are available at 50-year or higher resolution and have been synchronized to a common depth-age scale (the Greenland ice core chronology 2005, GICC05; Andersen et al., 2006; Rasmussen et al., 2006; Svensson et al., 2006; Vinther et al., 2006, 2008). All of these ice-core records extend from the beginning of the Holocene to the present. Five $\delta^{18}$O and four accumulation records also include the last glacial period. To evaluate the impact of the differing lengths of these records, we produce a sensitivity reanalysis for which we assimilate just the fixed proxy-network (i.e., only those data that span the full reanalysis time period, the last 20,000 years).

To extract the accumulation signal from layer-thickness, the layers must be destrained using assumptions about the history of ice flow. For the Dye3, GRIP, and NGRIP cores, we use a one-dimensional ice-flow model (Dansgaard and Johnsen, 1969) to calculate the cumulative vertical strain the layers have experienced at each core site. The Dansgaard-Johnsen model requires specifying the vertical velocity at the surface and a kink height which determines the shape of the vertical velocity profile. The velocity profile below the kink height approximates the influence of greater deformation rates in deeper ice due to increased deviatoric stress and warmer ice temperature. For sites at the pressure-melting point at the bed, such as NGRIP, we also implement the basal melt-rate (e.g., Dahl-Jensen et al., 2003). Previous work on many of the Greenland ice cores has estimated

cumulative vertical thinning assuming that the accumulation history scales with $\delta^{18}$O and then found optimal parameter values by comparing the modeled and measured depth-age relationships (Dahl-Jensen et al., 1993, 2003; Rasmussen et al., 2014). Here, we wish to maintain independent determinations of the $\delta^{18}$O and accumulation proxy records. To do this, we select reasonable ice-flow parameters independently, based on the glaciological setting of each site; specifically, we use kink-height values of 0.1-0.2 for flank flow and 0.4 for ice flow near ice divides where the deviatoric stress is low (Raymond, 1983; Conway et al., 1999). Where available, we use published values or kink-height values that result in a good match to published accumulation records (Dahl-Jensen et al., 2003; Gkinis et al., 2014). Based on this range of plausible ice-flow parameters, we develop three scenarios for each site: "low", "moderate", and "high", where the names reflect the relative magnitude of accumulation in the glacial and early Holocene. We assimilate the intermediate-value ("moderate") accumulation records to produce our main precipitation reanalysis, while we assimilate the high and low accumulation records into high and low sensitivity scenarios, respectively, to provide a conservative estimate of uncertainty. Descriptions of the rationale for the parameter choices at each site are given in Sect. S2. Our method to estimate accumulation should be most accurate for the interior ice cores (i.e., GISP2, GRIP, NEEM, NGRIP); these sites are thicker and have lower accumulation rates such that layers of the same age have experienced less cumulative strain than for the more coastal cores (i.e., Agassiz, Camp Century, and Renland). We do not attempt to reconstruct accumulation from these coastal cores because the layers cannot be destrained with sufficient accuracy. Dye3 suffers from the same challenges; however, it is the only ice core with long-term climate data south of 70°N (Fig. 1). Thus, we include the Holocene Dye3 accumulation rates despite the greater uncertainty relative to the interior cores.

Because records from the Dye3 ice core are our only source of information in southern Greenland, we take the following steps to increase the data available for assimilation. The Dye3 record has not been previously assigned a depth-age scale beyond 11.7 ka (throughout this paper, "ka" refers to thousands of years before 1950 CE). We extend the depth-age scale to 20 ka to take advantage of the glacial portion of the $\delta^{18}$O record. To do this, we match the $\delta^{18}$O record from Dye3 to the $\delta^{18}$O record from NGRIP using the cross-correlation maximization procedure from Huybers and Wunsch (2004) (Sect. S3). We interpolate the glacial $\delta^{18}$O record from Dye3 (which, as measured, has an average resolution of only 85 years) to the same 50-year resolution used for our other ice-core records. Extension of the Dye3 depth-age scale also provides a layer-thickness record from 20 ka to present; however, we do not use accumulation data from Dye3 for the period 20-11.7 ka because the low resolution impedes our ability to estimate accumulation variations from layer thickness. Using this depth-age scale extension for Dye3 may introduce error that is difficult to quantify; however, we find that including Dye3 has an important impact on the resulting reanalysis of southern Greenland climate (Sect. S4).

Where possible, we account for non-local effects on the ice-core records. The global-mean $\delta^{18}$O of seawater fluctuates with global ice-volume, while on the regional scale, horizontal advection brings ice from other elevations and latitudes. We correct for changes in the oxygen-isotope composition of seawater following the methods of Stenni et al. (2010), using the benthic foraminifera dataset from Bintanja et al. (2005). For ice cores in regions of high horizontal advection, we correct for elevation and latitude differences between the site of snow deposition and the ice-core site. Following the methods from Dahl-Jensen et al. (2013), we apply corrections for advection-caused elevation changes in the $\delta^{18}$O records from Camp Century, Dye3, and NEEM and for advection-caused latitude changes in the $\delta^{18}$O record from NEEM (Vinther et al., 2009; Dahl-Jensen

et al., 2013). We do not correct the accumulation records for advection from upstream because the elevation-accumulation relationship is complicated by the prevailing wind direction (Roe and Lindzen, 2001) and the thinning function uncertainties are likely to be larger than the effects of ice advection. We also do not correct $\delta^{18}O$ or accumulation for changing elevation at the ice-core site; our goal is to reconstruct conditions at the surface, rather than at a constant reference elevation. We take the anomaly of each corrected $\delta^{18}O$ record and the ratio of each accumulation record relative to the mean of all data in the record that falls within the time period 1850-2000 CE. We then average each record to 50-year resolution, the lowest resolution in these records (with the exception of $\delta^{18}O$ from the glacial period in the Dye3 core). It is these corrected, averaged records that we use in the data assimilation (Figs. 2 and 3).

## 2.2 Climate-model simulation

We use TraCE-21ka, a simulation of the last 22,000 years of climate (22 ka to -.04 ka), which was run using the fully-coupled CCSM3 at T31 resolution (approximately 3.75 degrees horizontally) with transient ice-sheet, orbital, greenhouse gas, and meltwater flux forcings (Liu et al., 2009, 2012; He et al., 2013). For paleoclimate data assimilation, it is important that the climate simulation capture a range of possible climate states over the time period of interest. By design, TraCE-21ka captures the major glacial-to-Holocene temperature changes, as well as some of the short-term, rapid climate changes, such as the Bølling-Allerød transition (Liu et al., 2009). Many higher-resolution climate simulations are transient only over the last millennium (e.g., Bothe et al., 2015) or provide a snapshot of a certain time, such as last glacial maximum or the mid Holocene (e.g., Harrison et al., 2014). Individually, these millennial-length simulations have too little variability to capture the range of climate states across the glacial-interglacial transition. If combined, the biases in each simulation would need to be individually addressed, which is beyond the scope of this study.

From TraCE-21ka, we use two-meter air temperature for temperature ($T$) and the sum of large-scale stable precipitation and convective precipitation for precipitation ($P$). To correct for model bias in TraCE-21ka, we assume that the bias is stationary in time and apply the delta-change method (Teutschbein and Seibert, 2012) by taking the anomaly of temperature and the fraction of precipitation relative to the mean of our reference period (1850-2000 CE). An assumption of a stationary model bias is required because, with a small number of proxy records, we cannot afford to subsample them for the purposes of bias correction, data assimilation, and evaluation. After the bias correction, we average the TraCE-21ka variables (which originally have monthly resolution) to 50-year resolution, as we did for the ice-core records. In this process, we average 50 consecutive years (600 months) such that no year (or month) is used in more than one 50-year average. This averaging results in 440 time steps spaced 50 years apart.

TraCE-21ka includes changes in orbital forcing, which contribute to changes in the seasonality of temperature and precipitation. The strength of these seasonal cycles influences the mean-annual relationship between $\delta^{18}O$ and temperature (Steig et al., 1994; Werner et al., 2000; Krinner and Werner, 2003). TraCE-21ka consistently shows stronger temperature and precipitation seasonality in the glacial (20 to 15 ka) than in the Holocene (5 to 0 ka) at each of the eight ice-core sites considered in this study (Fig. S3). Relative to the annual mean, the glacial had warmer and wetter summers and colder and drier winters. The Holocene also shows such a seasonal cycle; however, there is a smaller difference between the summers and winters. Any

seasonal signal with wetter summers than winters will bias the $\delta^{18}O$ towards summer values. According to TraCE-21ka, this effect is amplified in the glacial. As we discuss in Sect. 2.3.1, the particularly strong summer bias in the glacial affects the mean-annual $\delta^{18}O$-temperature relationship in ways that are consistent with findings from borehole thermometry at the GISP2, GRIP, and Dye3 ice-core sites (Cuffey and Clow, 1997; Jouzel et al., 1997).

In addition to a change in the strength of the seasonal cycle, TraCE-21ka shows a temporal shift, with summer temperature and precipitation peaking earlier in the glacial (around June and July) than in the Holocene (from July to September) (Fig. S3). In the glacial, both variables peak around June and July, with only two exceptions: precipitation peaks in August at the Renland and Dye3 ice-core sites. In contrast, Holocene temperature peaks slightly later, in July, while precipitation peaks even later, in August and occasionally September. Both variables and both time periods show winter minimums in February, again with the

two exceptions of Renland and Dye3, which show later precipitation minimums. In this study, the timing of the seasonal peaks is relevant because it affects the precipitation-weighted temperature, defined in Eq. 7.

TraCE-21ka also includes prescribed transient ice sheets as a boundary condition. The transience is important for capturing the influence of elevation change on the ice-core records; however, the low horizontal resolution of TraCE-21ka leads to difficulties in capturing elevation changes at the edges of the ice sheet, in southern Greenland, and at coastal ice caps. In

addition, the ice sheets in TraCE-21ka are independent of climate, updated only every 500 years during the simulation, and taken from ICE-5G (Peltier, 2004), a now outdated ice-sheet reconstruction (Roy and Peltier, 2018).

## 2.3    Paleoclimate data assimilation

To combine the ice-core and climate-model data, we use an offline data assimilation method similar to that described in Hakim et al. (2016). If no covariance localization is used, as in this study, this method can be summed up as a linear combination

of randomly-selected model states that are weighted according to new information provided by the proxy records. "Offline" refers to the absence of a forecast model that evolves the climate state between assimilation time steps. The offline method is appropriate when model predictive-skill is small given the assimilation time step (Hakim et al., 2016, and references therein). Model predictive-skill is generally poor on decadal to longer timescales (Latif and Keenlyside, 2011, and references therein) except possibly during times of strong forcing, such as the Bølling-Allerød (14.7-12.7 ka) and the Younger Dryas (12.7-11.7

185    ka) (Hawkins and Sutton, 2009). Because each of our time steps is an average over 50 years, as dictated by the resolution to which we average the proxy records, the offline method is appropriate except possibly during these large-forcing events. For these events, an online method may be appropriate (assuming that the models correctly capture both the forcing and the response); however, online ensemble data assimilation over glacial-interglacial cycles using a fully-coupled earth system model is impractical due to the computational cost and is beyond the scope of this study.

Our paleoclimate data assimilation framework uses ensembles for the initial (prior) and final (posterior) estimates of the climate state, providing a probabilistic framework for interpreting and evaluating the results. To compute the posterior ensemble, we apply the Kalman update equation (Whitaker and Hamill, 2002), which spreads new information gained from proxy records

to all locations and variables in the prior ensemble:

$$\boldsymbol{x_a} = \boldsymbol{x_b} + \mathbf{K}(\boldsymbol{y} - \mathcal{H}(\boldsymbol{x_b})) \tag{1}$$

Bold lowercase letters are vectors, bold capital letters are matrices, and script capital letters are mapping functions. The posterior ensemble is $\boldsymbol{x_a}$, $\boldsymbol{x_b}$ is the prior ensemble, $\boldsymbol{y}$ is the proxy data, and $\mathcal{H}$ is the function that maps from the prior variables to the proxy variables. $\mathbf{K}$ is the Kalman gain matrix:

$$\mathbf{K} = \mathbf{B}\mathbf{H}^T(\mathbf{H}\mathbf{B}\mathbf{H}^T + \mathbf{R})^{-1} \tag{2}$$

where $^T$ indicates a matrix transpose, $\mathbf{B}$ is the covariance matrix computed from the prior ensemble, $\mathbf{H}$ is the linearization of $\mathcal{H}$ about the mean value of the prior, and $\mathbf{R}$ is a diagonal matrix containing the error variance for each proxy record, the use of which requires an assumption that error covariances between proxy records are negligible.

To compute the new information gained from the proxy records, the prior ensemble must first be mapped into proxy space to get prior estimates of the proxy ($\mathcal{H}(\boldsymbol{x_b})$). This mapping ($\mathcal{H}$) is the proxy system model (PSM). Our PSM for the $\delta^{18}$O-temperature relationship is a linear function and for accumulation is a direct comparison between ice-core-derived accumulation and precipitation from the prior. For these PSMs, both temperature and precipitation are interpolated from the climate-model grid to the geographic location of the ice core. These PSMs are detailed in Sects. 2.3.1 and 2.3.2. Comparing the prior estimates of the proxy ($\mathcal{H}(\boldsymbol{x_b})$) to the proxy data ($\boldsymbol{y}$) yields the innovation ($\boldsymbol{y} - \mathcal{H}(\boldsymbol{x_b})$), the new information gained from the proxy records.

The Kalman gain ($\mathbf{K}$, Eq. 2) weights the innovation by the relative magnitude of the ensemble covariance of the prior estimates of the proxy ($\mathbf{H}\mathbf{B}\mathbf{H}^T$), and the error covariance of the proxy records ($\mathbf{R}$). The Kalman gain spreads the weighted innovation to all locations and variables in the prior ensemble, using the covariance structure ($\mathbf{B}\mathbf{H}^T$) from the prior ensemble.

The prior ensemble is an initial estimate of possible climate states, which we form using 100 randomly-chosen 50-year averages from the TraCE-21ka simulation. States from both the glacial and the Holocene make up a prior ensemble. The same prior is used for all time steps in the reconstruction, leading to a prior that is constant in time. For a longer discussion on the reasoning behind our choice to use a constant prior, please refer to the Supplementary Information, Sect. S1. Proxy records are assimilated into the prior using Eq. 1, which produces the posterior ensemble, a new estimate of possible climate states. We assimilate $\delta^{18}$O to reconstruct temperature and separately assimilate accumulation to reconstruct precipitation. This approach maintains independence between temperature and precipitation, which avoids imposing linearity and stationarity on the relationship between these two variables. As Cuffey and Clow (1997) show, not only is this relationship non-linear on long timescales but it is also not well-approximated by simple thermodynamic expectations. Separating these variables ensures that the relationship between temperature and precipitation is consistent with the empirical relationship between $\delta^{18}$O and accumulation from ice cores, rather than being derived exclusively from the climate model.

We repeat the data assimilation process over multiple iterations, with each iteration using one of ten different 100-member prior ensembles and excluding one proxy record. Each of the ten prior ensembles is made up of a different random selection of 50-year averages from TraCE-21ka. Thus, each prior ensemble has a different variance and spatial covariance structure. Each

proxy record is excluded from a total of ten iterations, where each of these iterations uses a different one of the ten prior options. Every iteration is uniquely identifiable by which prior ensemble is used and which proxy record is excluded. For a reanalysis, the total number of iterations is thus ten times the number of proxy records, such that for temperature we have 80 iterations and for precipitation we have 50 iterations. A reanalysis is a compilation of the 100-member posterior ensembles from these

230 iterations, resulting in a temperature reanalysis having 8,000 ensemble members and a precipitation reanalysis having 5,000 ensemble members.

The proxy record that is excluded from an iteration is independent of that iteration's posterior ensemble, such that we can evaluate the posterior against this record. With our PSMs, we convert the posterior ensemble into predictions of the independent record using the mapping $\mathcal{H}$ and compare these predictions to the record along the time axis. We use four skill metrics to

235 evaluate different aspects of the predictions. The correlation coefficient (corr; Eq. 3) measures the relative timing of signals in the predictions and the proxy record:

$$corr = \frac{1}{n-1} \sum_{i=1}^{n} \left( \frac{(y_i - \bar{y})(v_i - \bar{v})}{\sigma_y \sigma_v} \right) \tag{3}$$

where $v$ is the ensemble mean of the predicted values, $y$ is the proxy record value, an overbar indicates a time mean, $n$ is the number of time steps, and $\sigma$ is the standard deviation of the variable in the subscript. The coefficient of efficiency (CE; Eq. 4)

(Nash and Sutcliffe, 1970) is affected by signal timing as well as signal amplitude and mean bias:

$$CE = 1 - \frac{\Sigma_{i=1}^{n}(v_i - y_i)^2}{\Sigma_{i=1}^{n}(v_i - \bar{v})^2} \tag{4}$$

The root mean squared error (RMSE; Eq. 5) gives an intuitive sense for the magnitude of the differences between the predictions and proxy record:

$$RMSE = \left( \frac{1}{n} \sum_{i=1}^{n} (y_i - v_i)^2 \right)^{1/2} \tag{5}$$

The ensemble calibration ratio (ECR; Eq. 6) indicates whether the ensemble has enough spread (uncertainty) given the error in the ensemble mean (e.g. Houtekamer et al., 2005):

$$ECR = \frac{1}{n} \sum_{i=1}^{n} \left( \frac{(y_i - v_i)^2}{var(\boldsymbol{v_i}) + r} \right) \tag{6}$$

where $\boldsymbol{v}$ is a vector of the ensemble members of the predicted values, $r$ is the error variance for the proxy record ($y$), and $var$ indicates the variance. Accordingly, if the ensemble variance is appropriate for the amount of error, then the ensemble

calibration ratio is near unity.

We compute all four skill metrics for both the posterior and prior ensembles, which shows whether assimilation of proxy records results in an improved estimate of the climate state over our initial estimate. We define improvement as correlation coefficient closer to 1, CE closer to 1, RMSE closer to 0, and ensemble calibration ratio closer to 1. We anticipate that our reanalysis will show improvement over the prior because the prior is constant in time and contains no information about

255 temporal climate variations; however, improvement is not gauranteed, especially if proxy records contain highly-localized

climate signals or if the prior covariance structure is unable to appropriately spread information from the proxy sites to other locations. For further comparison, we additionally compute the correlation coefficient, CE, and RMSE between the TraCE-21ka simulation and the proxy records.

We evaluate results over three time periods: 1) the full overlap between the reanalysis time period and the proxy record, 2) a time representative of the glacial, 20-15 ka, and 3) a time representative of the Holocene, 8-3 ka. Some proxy records overlap the full reanalysis period, 20-0 ka, while others overlap just the Holocene, 11.7-0 ka. The skill metrics computed for these two groups should be considered separately.

To remove mean bias from temperature, we subtract out the reference-period mean. For precipitation, we divide by the reference-period mean. It is these bias-corrected results that are referred to unless noted otherwise.

### 2.3.1 Proxy system model: $\delta^{18}$O

The isotopic composition of precipitation, as recorded in ice cores, is highly correlated with temperature at the time and location of deposition, but is also sensitive to conditions at the moisture source region (i.e. sea surface temperature and relative humidity at the ocean surface; e.g., Johnsen et al., 1989). Moisture-source conditions primarily affect the deuterium excess, which we do not use here, and the influence on $\delta^{18}$O is comparatively weak (e.g., Armengaud et al., 1998). For our $\delta^{18}$O PSM, we use a linear relationship with temperature at the ice-core drill site ($T_{site}$) that has a slope of $0.67 \pm 0.02$ ‰ °C$^{-1}$, which was calibrated using modern, spatial data (Johnsen et al., 1989). It is well known that this modern, spatially-derived slope does not necessarily apply to temporal $\delta^{18}$O-$T_{site}$ relationships, which have effective slopes that are time, frequency, and location dependent. Temporal changes in precipitation seasonality, inversion-layer thickness, and source region conditions introduce nonlinearity into the effective $\delta^{18}$O-$T_{site}$ relationship (e.g., Jouzel et al., 1997; Pausata and Löfverström, 2015). Diffusion in the firn column also affects this relationship, but it is negligible for annual and longer timescales at the locations of the ice cores we use (Cuffey and Steig, 1998). Borehole thermometry at the GISP2 and GRIP sites show that for the low-frequency changes associated with the last glacial-interglacial transition, the temporal slope is less than half the modern, spatial slope (Cuffey and Clow, 1997; Jouzel et al., 1997). Numerous studies have suggested that precipitation seasonality is the largest source of nonlinearity in the $\delta^{18}$O-$T_{site}$ relationship (e.g., Steig et al., 1994; Pausata and Löfverström, 2015); changes in precipitation seasonality are thought to be the primary reason that the effective $\delta^{18}$O-$T_{site}$ relationship for the glacial-interglacial transition has such a low slope (Werner et al., 2000; Gierz et al., 2017; Cauquoin et al., 2019).

We rely on TraCE-21ka to estimate the site-specific effects of precipitation seasonality on the $\delta^{18}$O-$T_{site}$ relationship. Site-specific effects can also be estimated using independent temperature reconstructions, e.g. from borehole thermometry or $\delta^{15}$N of N$_2$ measurements; however, such independent reconstructions for the last 20,000 years exist only for a few of the long ice-core records, GISP2, GRIP, and NGRIP (Buizert et al., 2018; Dahl-Jensen et al., 1998; Gkinis et al., 2014), limiting the utility of such records to capture the spatial variability of the $\delta^{18}$O-$T_{site}$ relationship across Greenland .

To incorporate estimates of the site-specific effects of precipitation seasonality from TraCE-21ka, we adjust the linear $\delta^{18}$O PSM by replacing $T_{site}$ with $T_{site}^{*}$, the precipitation-weighted temperature at the model grid-cell closest to the ice-core drill site. We compute $T^{*}$ across Greenland using $T$ and $P$ from TraCE-21ka (see Fig. S4 for a visual comparison of $T$ and $T^{*}$ at

the GISP2 ice-core site):

$$T^* = \sum_{i=1}^{n=12} \left( T_{mon} \frac{P_{mon}}{P_{ann}} \right) \tag{7}$$

With $T^*_{site}$ in our PSM, we find that the $\delta^{18}$O-$T_{site}$ slope is spatially variable (e.g., Fig. S5), ranging between 0.42 and 0.66 ‰°C$^{-1}$ at the ice-core sites (Table S1), and tending to be less than the modern spatial relationship of 0.67 ‰°C$^{-1}$ at most locations around Greenland. Due to the data assimilation method outlined above, these slopes vary both in space and across
iterations, the latter being due to the varying prior ensembles. These slopes do not vary in time in the prior, but they do in the posterior (note that $\delta^{18}$O-$T_{site}$ slopes mentioned throughout this paper refer to the prior ensemble). By using ten different prior ensembles, we capture the uncertainty in the $\delta^{18}$O-temperature relationship from variations in the precipitation seasonality. These TraCE-21ka-derived estimates lie within the range of slopes estimated for sites around Greenland for a variety of time periods (Table S1). Differences seen in Table S1 reflect both the different methods used and the time period considered. Some
estimates, such as Guillevic et al. (2013) and Buizert et al. (2014) are for abrupt transitions, such as Dansgaard-Oeschger events, while others find mean slopes over longer periods of time, such as Kindler et al. (2014) and this investigation.

To evaluate the sensitivity of our results to the choice of PSM, we produce four other reconstructions (S1-S4). The S1 scenario uses the PSM, $\delta^{18}$O$= 0.67T_{site}$, which is the modern (high-frequency) relationship and does not account for precipitation seasonality. The S2 scenario uses $\delta^{18}$O$= 0.5T_{site}$, the mean of the high-frequency and low-frequency temporal slopes.
The S3 scenario uses $\delta^{18}$O$= 0.335T_{site}$, which is similar to published estimates of the glacial-interglacial temporal slope (half the high-frequency slope) (Cuffey and Clow, 1997; Jouzel et al., 1997). By lowering the slope in the S2 and S3 scenarios, we implicitly account for precipitation seasonality; however, in these scenarios and S1, the $\delta^{18}$O-$T_{site}$ relationship is spatially uniform, whereas it is spatially variable in the main reanalysis because we include the spatial pattern of precipitation seasonality. The S4 scenario uses the same PSM as in the main reanalysis, $\delta^{18}$O$= 0.67T^*_{site}$, but we adjust the strength of the precipitation
seasonality in TraCE-21ka such that the average $\delta^{18}$O-$T_{site}$ slope around Greenland is approximately 0.335 ‰ °C$^{-1}$. The S4 scenario thus has the same spatially-variable $\delta^{18}$O-$T_{site}$ relationship as in the main reanalysis, but a greater influence of precipitation seasonality.

These sensitivity tests are equivalent to testing different assumptions about the $\delta^{18}$O-$T_{site}$ relationship. The availability of a 20,000 year-long isotope-enabled climate simulation would allow us to determine this relationship from model physics, which incorporate a variety of processes that can affect water isotopes, including precipitation seasonality.

### 2.3.2   Proxy system model: Accumulation

Accumulation is closely related to total precipitation at our ice-core sites, which have limited surface melting and evaporation. Simulations from the regional climate model HIRHAM5 show that for modern climate at the GISP2, GRIP, NEEM, and NGRIP ice-core sites, surface mass balance, snowfall, and precipitation are all within 1.6 cm water equivalent (w.e.) when averaged
over the years 1989 to 2012 CE (Langen et al., 2015, 2017). For this reason, and because TraCE-21ka lacks process-based ablation variables, our PSM is a direct-comparison between ice-core accumulation and simulated precipitation at the model grid-cell closest to the ice-core site.

This direct-comparison PSM may be an incomplete model of the accumulation-precipitation relationship at the Dye3 ice-core site; regional climate simulations show that modern surface mass balance is lower than precipitation due to melt rates that average 84 cm w.e. year$^{-1}$. Significant melt rates would cause the spatial covariance structure of accumulation across these sites to differ from that of precipitation; however, both models and observations lack the necessary variables or duration to show the extent to which this difference exists for 50-year timescales through the last glacial-interglacial cycle. We emphasize that we use relative, rather than absolute changes in the data assimilation, to account for the mean bias between precipitation and accumulation.

### 2.3.3 Proxy error variance estimation

In the Kalman filter (Eq. 1), the diagonal elements of $\mathbf{R}$ contain the error variance of each proxy record, which includes how we model the proxy (the PSM). We compute representative error variances for $\delta^{18}$O and accumulation, and apply them to all records and time slices. We do not include error associated with corrections applied to the ice-core data (Sect. 2.1) or associated with the accumulation PSM (Sect. 2.3.2) because we cannot characterize the statistical properties of these errors.

A universal, but typically small, error source is from the measurement of proxies. For $\delta^{18}$O, measurement error is equivalent to laboratory precision. We compute a representative measurement error from the GISP2 ice core, for which a single measurement of $\delta^{18}$O has a laboratory precision (variance) of 0.024 ‰$^2$ (Stuiver and Grootes, 2000). Assuming independent error and annual measurements, the 50-year average error variance reduces to 0.0034 ‰$^2$, which is insignificant compared to other sources of error. For accumulation, the measurement error is from the measurement of layer thickness, which is related to the error in annual-layer counts per unit depth. Again, we assume GISP2 is a representative core and estimate the layer-thickness error from Table 3 in Alley et al. (1997), which provides repeat annual-layer counts for several depth intervals. From this table, we find the standard deviation of counted years in each depth interval, divide by the average number of years, average across all depth intervals, and square the result. This computation results in a layer-thickness error variance of 0.0015, a unitless number due to our use of fractional accumulation records.

Another source of error is the extent to which a model grid-cell may misrepresent a point proxy-measurement. In the innovation, there is an implicit assumption that the proxy ($\boldsymbol{y}$) and the prior estimate of the proxy ($\mathcal{H}(\boldsymbol{x_b})$) are representative of the same processes. However, an ice core is about 100 cm$^2$, an area that is affected by processes at all scales, from regional change to local, sub-meter-scale topography, while a model grid-cell in TraCE-21ka can cover tens of thousands of square kilometers, an area that is affected by only the largest scales, from global to regional. Thus, there is an inherent inability of the prior to represent local processes at the ice-core site, which we refer to as the spatial representation error. To estimate this error, we compute the variance of the local noise (e.g., Reeh and Fisher, 1983) using the GISP2 and GRIP ice-core records, which are located about 30 km apart within the same model grid cell. For $\delta^{18}$O, our estimate is 0.21 ‰$^2$ which is about half the value determined by Fisher et al. (1985) at several locations around Greenland. Our estimate is relatively conservative, considering that we are using 50-year averages rather than annual averages as in Fisher et al. (1985). For accumulation, we estimate a spatial representation error variance of 0.0023 using the same method as for $\delta^{18}$O.

A third source of error is the extent to which the $\delta^{18}$O PSM may be an inaccurate model of the $\delta^{18}$O-$T_{site}$ relationship. The less accurate the PSM, the less weight that should be given to the innovation. We estimate PSM error variance by calculating the mean squared error (MSE) between a $\delta^{18}$O record and an independent temperature record mapped to $\delta^{18}$O using the PSM, $\delta^{18}$O$= 0.67T_{site}$. We use independent datasets taken from the GISP2 ice core: the $\delta^{18}$O record and three $\delta^{15}$N-derived temperature estimates for the Holocene, a mean estimate and the two-standard-deviation uncertainty bounds (Kobashi et al., 2017). From these datasets, we estimate three PSM error variances that range from 0.56 to 1.1 ‰$^2$, from which we choose the largest.

For the assimilation of each $\delta^{18}$O record, we use an estimated total error variance of 1.3 ‰$^2$, which is a sum of the measurement, spatial representation, and PSM error variances. For the assimilation of each accumulation record, we use an estimated total error variance of 0.0038, which is a sum of the measurement and spatial representation error variances.

## 3 Results

### 3.1 Overview

Through the assimilation of ice-core data with a prior ensemble that is constant in time, we produce a spatially-complete, mean-annual Greenland temperature and precipitation reanalysis at 50-year resolution (Figs. 4 and 5). Here we focus on results for the late glacial and the Holocene thermal maximum (HTM).

In our reanalysis, late glacial (20-15 ka) mean-temperature anomalies range from about $-20$ °C in northern Greenland to less than $-10$ °C in southern Greenland (Fig. 4c). At the GRIP and GISP2 ice-core sites, the reanalysis has a $-14$ °C anomaly with a standard deviation of 2 °C. This is in excellent agreement with the mean-temperature anomaly of $-14$ °C for the same period at the GISP2 site, which was derived from $\delta^{18}$O calibrated with borehole thermometry (Cuffey et al., 1995; Cuffey and Clow, 1997). Also in agreement with borehole thermometry, this period is warmer than the last glacial maximum (Dahl-Jensen et al., 1993; Cuffey and Clow, 1997). Average late-glacial precipitation in the reanalysis ranges from a third to half of modern with the highest values on the coasts around southern Greenland (Fig. 4d).

Our reanalysis shows warmest temperatures occurred across Greenland between 7 and 3 ka, reaching a temperature maximum around 5 ka (Fig. 5). Although this timing tends to be later than many estimates of the HTM, it lies within the ranges reported in the literature; for example, a summary of proxy records from around Greenland shows peak warmth usually occurring around 9-5 ka, but also as early as 10.8 ka and as late as 3 ka (Kaufman et al., 2004). Borehole thermometry shows that temperatures peaked around 6-7.7 ka at Summit and 4.5 ka at the Dye3 ice-core site (Cuffey and Clow, 1997; Dahl-Jensen et al., 1998; Kaufman et al., 2004). Temperature estimates from $\delta^{15}$N of N$_2$ show an earlier HTM peak at Summit around 8 ka (Kobashi et al., 2017). In northwest Greenland, $\delta^{18}$O measurements from lake sediments show the HTM starting before 7.7 ka and ending around 6 ka (Lasher et al., 2017), while chironomid assemblages show peak warmth around 10-8 ka (McFarlin et al., 2018).

Mean-annual HTM temperature anomalies in our reanalysis range from nearly $+2$ °C in northern Greenland to about $+1$ °C in southern Greenland (Fig. 4a). Similar to the late-glacial temperature anomalies, the pattern of the HTM is dominated by a north-

south trend that has the greater temperature changes to the north, especially in northwest Greenland. While this spatial pattern agrees well with previous studies which have noted especially warm Holocene temperature anomalies in northwest Greenland (Lasher et al., 2017; Lecavalier et al., 2017; McFarlin et al., 2018), many estimates of HTM anomalies around Greenland are higher than our reanalysis indicates. Our low temperature estimates (compared to previous work) may be in part due to our reconstructing the annual mean rather than the summer mean; the greatest temperature anomalies in the HTM are thought to have occurred in the summer months and many proxies are more sensitive to summer than annual temperature (Kaufman et al., 2004). Importantly, in our reanalysis, the higher HTM temperatures do not translate to a marked increase in precipitation as would be expected from a thermodynamically-scaled relationship between temperature and precipitation. Instead, we find fractional precipitation within 2% of modern values (1.0±0.02) during the HTM (Fig. 4b), with slightly higher-than-modern precipitation in central Greenland and slightly lower-than-modern precipitation in northwestern Greenland.

## 3.2 Independent proxy evaluation

Here we evaluate our results against proxy records that are excluded from ten of the iterations that make up the temperature and precipitation reanalyses. For this evaluation, we use the raw results (without a mean-bias correction). We find, however, that the mean biases are small relative to climate changes over the last 20,000 years; there is little difference between our bias-corrected and uncorrected results and it is unlikely that the mean bias has a large effect on our evaluation.

Evaluation against independent proxy data shows that our reanalysis captures the timing and magnitude of low-frequency climate changes (Figs. 6 and 7) and is an improvement over both the prior ensemble and TraCE-21ka (Figs. S6-S7 and S8-S9). Evaluation over the full 20,000 years of the temperature and precipitation results shows high, positive correlation coefficients (ranging from a minimum of 0.97 to maximum of 0.99), which indicate that the reanalysis captures both the timing and sign of climate events, while high CE (0.87-0.98) and low RMSE values (0.62-1.2 ‰ for $\delta^{18}O$ and 0.04-0.08 for accumulation) indicate that the reanalysis captures the magnitude of these events. Our skill during this longest evaluation period is primarily due to the presence of low-frequency climate changes, which tend to be coherent across Greenland, such that evaluation over this full 20,000-year period shows more skill than evaluation over the full Holocene, which shows more skill than evaluation over just 5,000 years in the Holocene (or 5,000 years in the glacial) (Figs. 6 and 7).

Our posterior ensemble consistently shows improvement over the uninformed, constant prior ensemble during the 20,000-year evaluation period (Figs. S6 and S7). The TraCE-21ka simulation is also uninformed by the ice-core data, but it is transient and generally captures glacial to Holocene changes. Over the 20,000-year evaluation period, relative to the reconstructions, we find that TraCE-21ka has consistently lower correlation coefficients (0.86-0.96), lower CE values (0.50-0.86), and higher RMSE values (1.9-2.8 ‰ for $\delta^{18}O$ and 0.11-0.15 for accumulation) (Figs. S8 and S9). This comparison suggests that our reconstruction captures the timing and magnitude of the glacial to Holocene transition better than TraCE-21ka.

Even for the shorter evaluation periods, which are dominated by high-frequency, spatially-incoherent noise, the reanalysis shows improvement over both the prior ensemble and TraCE-21ka (Figs. S6-S7 and S8-S9); however, the improvement is not as consistent as for the 20,000-year evaluation period. For our reconstruction, correlation is positive except for three locations in the temperature reconstruction, with Holocene precipitation showing the largest correlation values (up to 0.60) and the total

range being -0.30 to 0.60. The prior correlation is zero for these shorter evaluation periods and locations; however, TraCE-21ka shows correlation values ranging from -0.29 to 0.69, with eight negative correlations (more than we find for our reconstruction), but generally higher correlations in the Holocene than our reconstruction.

For the shorter evaluation periods, the reconstruction CE is generally negative (ranging from -82 to 0.17) with a few exceptions; however, the reconstruction may still be skillful (e.g., Cook et al., 1999). The skill of the reconstruction is better measured by the difference between prior and posterior CE due to the strong influence that bias can have on CE (Hakim et al., 2016). There is consistent improvement of the posterior CE over that of the prior (ranging from an increase of 4.7 to 3200) and over that of TraCE-21ka (ranging from an decrease of -6.9 to an increase of 230). RMSE is the most consistent of the skill metrics for these shorter evaluation periods, with our reconstruction showing improvement over both the prior and TraCE-21ka, the one exception being that TraCE-21ka has greater skill at the Agassiz ice-core site in the Holocene. For the reconstruction, RMSE values range from 0.24 to 1.8 $‰^2$ for temperature and 0.025 to 0.10 for precipitation.

For all evaluation periods and both variables, the ensemble calibration ratio (ECR) for the prior is skewed towards values greater than unity (0.66-8.7), which suggests that the prior ensemble tends to have too little spread. Conversely, for the posterior, the ECR is generally less than unity (0.10-1.7) (Figs. 6 and 7), suggesting that the posterior ensemble has more spread than the error in the ensemble mean (as compared to the proxy records). This result indicates that the reanalysis ensemble encompasses the climate as recorded by the proxy records for most times and locations over the last 20,000 years.

### 3.3 Sensitivity evaluation

Proxy networks that change in time, such as ours, can introduce artificial discontinuities into the reanalysis, especially if the number of proxies is low or the proxy uncertainty values are inappropriate. We produce another reconstruction with a fixed proxy network, in which all assimilated proxy records participate in every time step in the reconstruction (see Sect. 2.3). A comparison of these results with our main reanalysis shows no apparent discontinuities for the ensemble mean and 5th to 95th percentiles at Summit (Fig. 8) or other locations around Greenland.

Our results are sensitive to the $\delta^{18}$O-$T$ relationship. To test this, we compare the main reanalysis to the four scenarios (S1-S4, as described in Sect. 2.3.1) that use different $\delta^{18}$O PSMs, each of which assumes a different slope and spatial pattern of the $\delta^{18}$O-$T$ relationship. We show this comparison for Summit as an example (Fig. 9), but the findings are applicable for all locations. As discussed previously, scenarios S1-S3 all assume that the $\delta^{18}$O-$T$ relationship has a uniform spatial pattern, but they each assume a different influence of precipitation seasonality. From these scenarios, we find that the temperature reconstruction is sensitive to the assumed precipitation seasonality, especially in the glacial where a stronger seasonality results in a greater glacial temperature anomaly. At Summit, this difference is nearly 10 °C between S1, which assumes no influence, and S3, which assumes the most influence of precipitation seasonality (Fig. 9). Similarly, the main reanalysis and S4 scenario both assume that the $\delta^{18}$O-$T$ relationship has a spatially-variable pattern, but S4 assumes a greater influence of precipitation seasonality. Again we find that the results are sensitive to assumed seasonality, with the greatest impact on the glacial-to-interglacial change.

The temperature results are also sensitive to the spatial pattern of the $\delta^{18}$O-$T$ relationship. We find this by comparing the results from the S1-S3 scenarios that assume a spatially-uniform relationship to results from the main reanalysis and S4 scenario that assume a spatially-variable relationship. The S1-S3 scenarios have a characteristic shape to their time series (Fig. 9), and, although the main reanalysis and S4 scenario generally fit this characteristic shape in the glacial and middle-late Holocene, in the early Holocene the main reanalysis and S4 diverge and show slower warming trends than the S1-S3 scenarios (Fig. 9b). In addition, the reconstructions with spatially-varying $\delta^{18}$O-$T$ relationships show stronger north-south gradients during times of abrupt temperature change than those with spatially-constant relationships (Table S2). These findings indicate that there is new information added by using a PSM that includes spatial variability in precipitation seasonality.

For precipitation, we find that the results are sensitive to which accumulation record is assimilated at each ice-core site. As explained in Sect. 2.1 and S1, we use a low, moderate, and high accumulation record for most of the ice-core sites to produce the low, main, and high precipitation scenarios, respectively (Fig. 10). The largest spread among the scenarios is in the earliest part of the reconstruction, i.e. the last glacial through the early Holocene, since uncertainties in the ice thinning history have the greatest impact at greater depths (and hence, greater ages). There is also a larger spread among the scenarios at more southern locations because the accumulation record at Dye3 has both the most influence on southern Greenland (Sect. S4) and the largest uncertainty in the ice thinning history (Fig. 3, Sect. S2).

## 4 Discussion

### 4.1 Precipitation-temperature relationship

Our results allow us to investigate the relationship between temperature and precipitation. To facilitate comparison with thermodynamic scalings widely used by ice-sheet models, we define the relationship as exponential and find the best fit for our reanalysis:

$$P_{fraction} = \frac{P_{past}}{P_{modern}} = e^{\beta(T_{past} - T_{modern})} \tag{8}$$

where $P$ is the precipitation rate, $T$ is the temperature, and $\beta$ is a scaling factor (Greve et al., 2011). For a given temperature change, a higher value of $\beta$ results in a larger change in precipitation (orange in Fig. 11a). In ice-sheet models that use this scaling, it is commonly applied with a uniform $\beta$ value for all locations (e.g., Huybrechts et al., 1991; Huybrechts, 2002; Greve, 1997; Greve et al., 2011; Pollard and DeConto, 2012; Cuzzone et al., 2019). The best-fit scaling-factors ($\beta$) center on the Greve et al. (2011) value of 0.07 for locations around Greenland, but the scaling factors tend to be larger ($\beta$ >0.10) where late-glacial precipitation is lowest and smaller ($\beta$ <0.05) where late-glacial precipitation is highest (Figs. 4d and 11a).

Spatial varibility in the temperature-precipitation relationship was also found by a previous study, which looked at decadal averages over a recent 110 year period (Buchardt et al., 2012). Also using an exponential to represent the relationship, Buchardt et al. (2012) similarly found weaker relationships in northern than in central and southern Greenland (compare their Fig. 3 to our Fig. 11). In southern Greenland, they found a negative $\beta$ value in southwest and a positive value in southeast Greenland, which they attribute to the Foehn effect. We find the opposite pattern, with larger, positive $\beta$ values in southwest Greenland

as compared to southeast Greenland. These differences may result from the time periods, timescales, and spatial resolutions considered in each investigation and from methodology. The spatial pattern found by (Buchardt et al., 2012) is limited to the ice-core locations they considered, such that they have little information about the relationship near the ice-sheet edges. The spatial pattern in our reanalysis is influenced by both the ice-core records and the spatial covariance structures of the prior ensembles. With only one long ice core in southern Greenland, the east-west gradient in $\beta$ values is highly influenced by the spatial patten inherited from TraCE-21ka.

The precipitation-temperature relationship in our reanalysis is driven by the assimilated ice-core records, though, as mentioned above, the spatial pattern of this relationship is also influenced by the spatial covariance structures of the prior ensembles. Previous work with ice-core records has found that the relationship between temperature and precipitation is frequency dependent, with a stronger relationship at lower frequencies (Cuffey and Clow, 1997); as expected, there is a similar frequency-dependence in our reanalysis. We find that an exponential scaling captures the low-frequency glacial to Holocene precipitation change; however, this fails at higher frequencies (Fig. 11b-e). To evaluate this frequency-dependence, we filtered our results using 6th order, low-pass and high-pass Butterworth filters with 5,000 year$^{-1}$ cutoff frequencies. The low-pass filtered dataset shows the same precipitation-temperature relationship as the unfiltered dataset (Fig. 11c-d), while the high-pass filtered dataset shows that precipitation is less sensitive to changes in temperature (i.e., the value of $\beta$ is lower) at these higher frequencies (Fig. 11e). This decoupling of temperature and precipitation is apparent in the amplitude difference of high-frequency signals in the glacial and the Holocene. In our temperature reanalysis, we find that high frequencies in the glacial have a greater amplitude than those in the Holocene, while in our precipitation reanalysis, we find the opposite. A single scaling, as is typically used in ice-sheet modeling, cannot capture this difference.

We examine how the sensitivity experiments (Figs. 9 and 10) affect the scaling factor ($\beta$) in the precipitation-temperature relationship. We pair the five temperature reconstructions (main, S1-S4) and three precipitation reconstructions (low, moderate, and high) into fifteen possible combinations and conduct the same analysis as described above. Across these fifteen combinations, we find that the spatial pattern of $\beta$ is robust (Fig. 11a). The exact magnitude depends primarily on the temperature reconstruction and how cold it is in the glacial, with colder temperatures giving lower $\beta$ values. To a lesser degree, the magnitude also depends on the precipitation reconstruction, with wetter scenarios giving lower $\beta$ values. As previously, we find that the low-pass filtered datasets have the same or nearly the same $\beta$ value as the unfiltered dataset, while the high-pass filtered datasets have consistently lower $\beta$ values. As an example of this, Table S3 shows the $\beta$ value found for the Kangerlussuaq region for all fifteen combinations and three filtering options.

## 4.2 Spatial patterns during abrupt climate change events

Our paleoclimate data assimilation framework is not limited to the assimilation of $\delta^{18}O$ and accumulation rate records. In this section we examine how our reanalysis compares to reconstructions driven by another type of proxy, $\delta^{15}N$ of $N_2$. In particular, we focus on abrupt temperature events, for which there is previous work using $\delta^{15}N$ (e.g., Severinghaus et al., 1998; Guillevic et al., 2013; Buizert et al., 2014). Abrupt temperature events increase the thermal gradient in the firn – the upper, porous portion of the ice column – which leads to fractionation of the stable isotopes of $N_2$ (Severinghaus et al., 1998). Using a firn

compaction model, temperature can be derived from $\delta^{15}$N measurements with inverse methods (e.g., Severinghaus et al., 1998; Severinghaus and Brook, 1999; Guillevic et al., 2013; Kindler et al., 2014; Buizert et al., 2014; Kobashi et al., 2017). We assimilate temperatures derived from $\delta^{15}$N data collected from the GISP2, NGRIP, and NEEM ice cores (Buizert et al., 2014).

Our reanalysis and the Buizert et al. (2014) records cover three abrupt temperature events – the Bølling-Allerød warming (14.6 ka), cooling into the Younger Dryas (12.9 ka), and warming at the end of the Younger Dyras (11.5 ka). For these three abrupt temperature events, the $\delta^{15}$N-derived temperature records show larger temperature changes at GISP2 in central Greenland (Buizert et al., 2014), while the $\delta^{18}$O changes are larger at NEEM in northwest Greenland (Fig. 2). We perform three sets of experiments to investigate how these two sets of proxy records affect the mean spatial pattern indicated by our reanalysis

during abrupt temperature events. The first experiment, "O8", assimilates all eight $\delta^{18}$O records with one 100-member prior ensemble from TraCE-21ka. This results in a 100-member reanalysis ensemble. The second experiment, "N3O5", assimilates all three of the $\delta^{15}$N-derived temperature records and the five remaining $\delta^{18}$O records (those that do not overlap with the $\delta^{15}$N sites), using the same 100-member prior ensemble as used in the O8 experiment. Finally, we perform a modified experiment, "N3O5_BA", with both $\delta^{18}$O and $\delta^{15}$N records, but using a prior ensemble selected from the 1,000 years surrounding the

Bølling-Allerød warming. To maintain a 100-member prior ensemble, we use decadal rather than 50-year averages of TraCE-21ka for this experiment. This adjustment does not affect the comparison (results not shown). Detailed methods for these experiments are given in Sect. S6.

    For both the "O8" and "N3O5" experiments, the spatial pattern for the abrupt climate change events are similar to our main reanalysis, with the largest magnitude temperature changes in northern and northwestern Greenland, decreasing magnitude

with decreasing latitude, and slightly larger change in the central east and southeast than corresponding western regions. For example, the spatial pattern of the Younger Dryas cooling is nearly the same regardless of which grouping of records is assimilated (Figs. 12a and b). This overall finding is robust to different combinations of these proxy records; for example, if we assimilate just the three $\delta^{15}$N-derived temperature records and no $\delta^{18}$O records (results not shown) the pattern is not substantially changed. This pattern of temperature change differs from spatial patterns inferred previously from $\delta^{15}$N for

various abrupt climate events (Guillevic et al., 2013; Buizert et al., 2014); however, the O8 and N3O5 experiments show that these differences are not due to the assimilation of $\delta^{18}$O rather than $\delta^{15}$N-derived temperatures. The replacement of $\delta^{18}$O records with $\delta^{15}$N-derived temperature records does not change the overall spatial pattern of these abrupt events, though it does result in a reconstruction with slightly larger temperature changes in the south and slightly smaller changes in the north. This effect, however, is less important than the choice of prior ensemble, as is shown by the third experiment, N3O5_BA.

For N3O5_BA, we restrict the prior ensemble to the Bølling-Allerød warming, which produces a reconstructed spatial pattern similar to those reported by Buizert et al. (2014) for each of the three abrupt climate events. In TraCE-21ka, the Bølling-Allerød warming is forced by a sudden termination of freshwater forcing in the North Atlantic (Liu et al., 2009). This forcing leads to large temperature fluctuations in southern Greenland that decrease with increasing latitude. With this covariance pattern dominating the prior ensemble, the N3O5_BA reconstruction indeed shows the largest temperature changes

in southern Greenland, followed by central and then northern Greenland (Figs. 12c, S12c, and S13c).

These experiments suggest that current ice-core records are insufficient to place a strong constraint on the spatial pattern of abrupt climate events. Reconstructions of these events may be improved with additional data, especially from southern Greenland (Sect. S4), and with prior ensembles that are designed to sample over uncertainty in the forcing and boundary conditions unique to these events.

## 4.3 Climate and the ice sheet: A case study of Southwest Greenland

Our main reanalysis is one of very few spatially-complete time series of Greenland climate over the last 20,000 years. Here, we compare our reanalysis with other Greenland climate histories, and suggest that, together, they should be treated as an ensemble of climate boundary conditions that can be used to produce ensembles of ice-sheet model simulations. These climate histories can also be further evaluated using a combination ice-sheet models and independent constraints from the glacial-geologic record of past ice-sheet configurations.

We compare our results with the recent reconstruction of Buizert et al. (2018), hereafter referred to as the B18 reconstruction. The B18 temperature reconstruction was produced by adjusting a part of the TraCE-21ka temperature field that is affected by changes in the Atlantic meridional overturning circulation such that the full temperature field provides a good match to an average of three $\delta^{15}$N-derived temperature records recovered from ice cores. The B18 snow-accumulation reconstruction is simply a reference climatology scaled to accumulation rates from the GISP2 ice core. We treat this as a precipitation reconstruction, but note that accumulation may be less than precipitation at some locations around Greenland, especially near the coast. It is also informative to compare these results with our S4 temperature and high precipitation reconstructions (hereafter referred to as S4 and high P or simply 'sensitivity' as in Fig. 13), as well as the TraCE-21ka simulation itself (i.e., the climate model output, unconstrained by data). For brevity, we focus on the area around Kangerlussuaq in southwest Greenland, but the comparisons are generally applicable to any region of Greenland. Southwest Greenland is of interest because the ice-sheet behavior here is primarily a response to changes in surface forcing (i.e., temperature and precipitation) because there are few tidewater glaciers (Cuzzone et al., 2019). Furthermore, the Kangerlussuaq region has a particularly well-documented ice-sheet retreat history through the Holocene (Young and Briner, 2015; Lesnek and Briner, 2018).

In the Kangerlussuaq region, the B18 reconstruction shows more extreme temperature changes than our reconstructions, with late-glacial (20-15 ka) anomalies of about $-20$ °C and peak HTM temperature anomalies of about $+2$ °C at 9 ka (Fig. 13). B18 also shows a faster rate of transition between the glacial and Holocene, reaching temperatures close to modern by 10 ka. In contrast, TraCE-21ka shows more moderate temperature anomalies and a slower transition, with late-glacial anomalies of about $-8.6$ °C and near-modern temperatures that first appear around 7 ka. TraCE-21ka has no obvious HTM in this location or any location in Greenland. Our main reanalysis and the S4 version of our temperature reanalysis both lie between B18 and TraCE-21ka, with late-glacial anomalies of about $-12$ and $-14$ °C, respectively, Holocene peak temperature anomalies of $+1$ °C around 5 ka, and temperatures close to modern first appearing around 8 ka.

For precipitation, the B18 reconstruction again tends to show the largest fluctuations and fastest transition, with a late-glacial precipitation fraction of about 0.26 and precipitation rates close to modern first appearing just after 10 ka. TraCE-21ka again shows the most moderate fluctuations and a slower transition, with a late-glacial fraction of about 0.38 and rates close to modern

not appearing until around 5 ka. Our main reanalysis and high P lie in the middle during the late-glacial, with fractions of about 0.32 and 0.36, respectively; however, our main reanalysis has a slow transition into the Holocene, similar to TraCE-21ka, while high P has a fast transition similar to B18. In the Holocene, high P shows the most elevated precipitation out of all the reconstructions, with 10-15% more precipitation than modern occurring around 7-3 ka. B18 shows precipitation values similar to modern for the last 10,000 years of the Holocene, while TraCE-21ka and our main reanalysis show lower-than-modern precipitation throughout most of the Holocene.

All of these paleoclimate reconstructions – our main reanalysis, the sensitivity scenarios, and B18 – are plausible histories of temperature and precipitation over Greenland. Given any past change in the ice-sheet, each of these histories has a different implication for ice-sheet sensitivity to climate, the veracity of which could be tested by using them to force an ice-sheet model and comparing this ensemble of results to the geologic record.

Our results have potentially important implications for the response of the Greenland Ice Sheet to climate change. In particular, we find maximum Holocene temperatures were reached around 5 ka, which is between 500 years earlier and 4,000 years later than most previous estimates. Moreover, there is little corresponding change in precipitation in our main reanalysis and the low sensitivity scenario. If these findings are correct, they imply a relatively rapid response to temperature forcing for sections of the ice sheet margin that retreated less than a century later (Young and Briner, 2015). A caveat is that proxy data remains very sparse, particularly in southern Greenland, where the poorly-resolved Dye3 core is the only long record. Future work to obtain improved measurements on the Dye3 core, or to gather new data from southern Greenland, would help to alleviate this limitation, as would the incorporation of data from off the ice sheet, such as from lake and ocean sediment cores.

An important distinction among various different paleoclimate reconstructions for Greenland is in the treatment of elevation changes. Any paleoclimate reconstruction from ice-core records is complicated by ice-sheet elevation changes. In Vinther et al. (2009), it is assumed that the climate history is the same at all locations around Greenland, and that any differences among the ice core paleotemperature records is a result of that elevation change. In B18, past elevation changes are assumed to be negligible. In our reconstruction, the impact of elevation change on the spatial covariances of temperature and precipitation is implicitly accounted for as part of the data assimilation methodology. Formally, our reconstruction is of surface climate, not climate at a fixed elevation. Consequently, our reanalysis may not be directly comparable to other paleoclimate reconstructions. For example, the HTM is commonly reconstructed as an early Holocene event in records that are at a fixed or nearly-fixed elevation. In our reanalysis, the early Holocene is cooler than the mid Holocene. Changes in the ice-surface elevation could account for this apparent discrepancy. Thinning in the early Holocene (Vinther et al., 2009) would result in a lowering of the ice surface and an apparent warming at the ice surface due to lapse rate effects. This warming signal would be captured in ice-core records. If the warming trend due to surface lowering occurs at the same time as an overall climate cooling, then the climate signal would be dampened or possibly reversed.

Our method depends on the accuracy of the climate-elevation relationships in our prior – i.e. in the TraCE-21ka climate model simulation, which probably does not capture such relationships with particularly high fidelity since the model resolution is low and the climate and ice-sheet models are not coupled. Future work could take advantage of the probabilistic relationships

among accumulation, temperature, and surface elevation as simulated in fine-scale regional climate models (Edwards et al., 2014).

## 5    Conclusions

Paleoclimate data assimilation is a novel method for reconstructing climate fields over the Greenland Ice Sheet. Our approach, combining ice-core records with a climate-model simulation, provides complete spatial reanalyses of mean-annual temperature and precipitation covering the last 20,000 years. Evaluation against independent proxy records shows that this methodology leads to significant and meaningful improvement over the prior ensemble (drawn from a climate simulation) and TraCE-21ka. Between the posterior ensemble and sensitivity experiments, our results provide a range of climate scenarios for ice-sheet modeling. Moreover, independently reconstructing both precipitation and temperature allows the assumption of purely thermodynamic control on precipitation to be relaxed, and an examination of the relationship between these quantities over a range of timescales. Specifically, we find that the Clausius-Clapeyron scaling is a good approximation over glacial-interglacial cycles, but not for shorter timescales where precipitation variability partially decouples from temperature.

Paleoclimate reconstructions would benefit from a larger selection of long climate-model simulations at higher resolution. Particularly valuable would be transient simulations that include water isotopes as prognostic variables, which allows for direct assimilation of water isotope ratios (Steiger et al., 2017; Okazaki and Yoshimura, 2017), rather than the use of an explicit proxy system model between temperature and $\delta^{18}$O. Recent work shows significant improvements to the realism of water-isotope enabled models in the polar regions (Nusbaumer et al., 2017; Dütsch et al., 2019; Cauquoin et al., 2019; Okazaki and Yoshimura, 2019), and longer simulations, once available, should allow us to further improve upon the results we have presented here. In principle, our method could also be applied to climate-model simulations that include a fully-coupled Greenland Ice Sheet. At present, fully-coupled simulations of Greenland over thousands of years are prohibitively expensive except at low resolution, and the limited work that has been done shows significant biases (Vizcaino et al., 2015). Nevertheless, incorporating data assimilation into such models would provide the groundwork for more-complete data-constrained simulations as computing power becomes less of a limiting factor in the future.

*Code and data availability.*    The paleoclimate reconstructions in this paper made use of code from the Last Millennium Reanalysis project, which is publicly available at https://github.com/modons/LMR (Hakim, 2019). The reconstructions, along with the new accumulation histories for Dye3, GRIP, and NGRIP, will be made publicly available in the Arctic Data Center at the time of publication.

*Author contributions.*    JAB and EJS conceived the idea for the study. JAB wrote code improvements necessary for this work, completed the paleoclimate reconstructions with guidance from EJS and GJH, conducted all analyses of the results, and wrote the first draft of the paper. GJH provided expert advice on data assimilation methodology and code development. TJF made the calculations of ice flow used to model accumulation for the Dye3, GRIP, and NGRIP cores. All authors contributed to the final version of the manuscript.

*Competing interests.* The authors declare that they have no conflict of interest.

*Disclaimer.* Any opinion, findings, and conclusions or recommendations expressed in this material are those of the authors(s) and do not necessarily reflect the views of the National Science Foundation.

*Acknowledgements.* Funding for this study was provided by the National Science Foundation Grant ARCSS no. 1503281 awarded to the University of Washington. In addition, this material is based upon work supported by the National Science Foundation Graduate Research Fellowship under Grant no. DGE-1256082 awarded to JAB. GJH also acknowledges support from the Heising Simons Foundation through

grant 2016-14 and the NSF through grant AGS-1602223. We thank Robert Tardif for help with code development, and we thank Joshua Anderson, Bo Vinther, Christo Buizert for their help compiling the ice-core data. We also thank the Snow on Ice project members for their discussions and support, especially Joshua Cuzzone, Jason Briner, and Elizabeth Thomas.

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

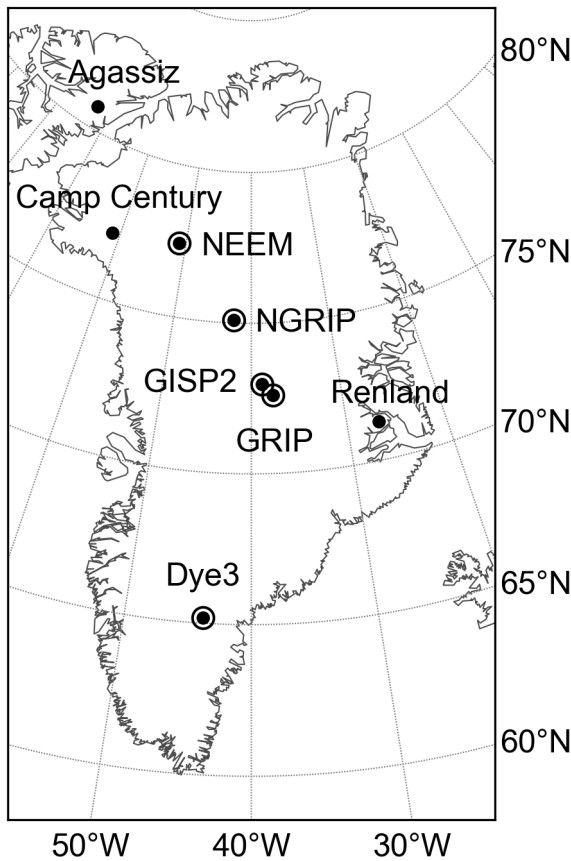

**Figure 1.** Locations of the ice-core sites referenced in this study. We use oxygen isotope ($\delta^{18}$O) records from all eight sites and accumulation records from the five circled sites.

**Table 1.** Metadata for the water isotope ($\delta^{18}$O) and accumulation (accum) records referenced in this study. "NBI" refers to the Niels Bohr Institute data access site (http://www.iceandclimate.nbi.ku.dk/data/) and "Pangaea" refers to the Pangaea data access site (https://www.pangaea.de/). Latitude and longitude are in units of decimal degrees (dd) and dates are in thousands of years before 1950 CE (ka).

| Ice core name | Latitude (dd) | Longitude (dd) | Variables | Oldest (ka) | Youngest (ka) | Source | Citations |
|---|---|---|---|---|---|---|---|
| Agassiz | 80.7 | 286.9 | $\delta^{18}$O | 11.64 | −0.02 | NBI | 1 |
| Camp Century | 77.18 | 298.88 | $\delta^{18}$O | 11.64 | −0.02 | NBI | 1 |
| NEEM | 77.45 | 308.94 | $\delta^{18}$O | >20 | −0.0108 | NBI | 2, 3, 4 |
| | | | accum | >20 | −0.04 | NBI | 5 |
| NGRIP | 75.1 | 317.7 | $\delta^{18}$O | >20 | −0.04 | NBI | 6 |
| | | | accum | >20 | −0.02 | this study | 7, 8, 9, 10 |
| GISP2 | 72.97 | 321.2 | $\delta^{18}$O | >20 | −0.04 | NBI | 11, 12 |
| | | | accum | >20 | −0.0375 | Pangaea | 13 |
| GRIP | 72.6 | 322.4 | $\delta^{18}$O | >20 | −0.02 | NBI | 14 |
| | | | accum | >20 | −0.02 | this study | 7, 8, 9, 10 |
| Renland | 71.27 | 333.27 | $\delta^{18}$O | 11.64 | −0.02 | NBI | 1 |
| Dye3 | 65.18 | 316.18 | $\delta^{18}$O | >20 | −0.02 | NBI | 1, 15 |
| | | | accum | 11.640 | 0 | this study | 16, 17 |

[1] Vinther et al. (2009), [2] Dahl-Jensen et al. (2013), [3] Schüpbach et al. (2018), [4] personal comm. Bo Vinther, [5] Rasmussen et al. (2013), [6] Andersen et al. (2004), [7] Vinther et al. (2006), [8] Rasmussen et al. (2006), [9] Andersen et al. (2006), [10] Svensson et al. (2006), [11] Grootes and Stuiver (1997), [12] Stuiver and Grootes (2000), [13] Cuffey and Clow (1997), [14] Johnsen et al. (1997), [15] Dansgaard et al. (1982), [16] Vinther et al. (2009), [17] this study

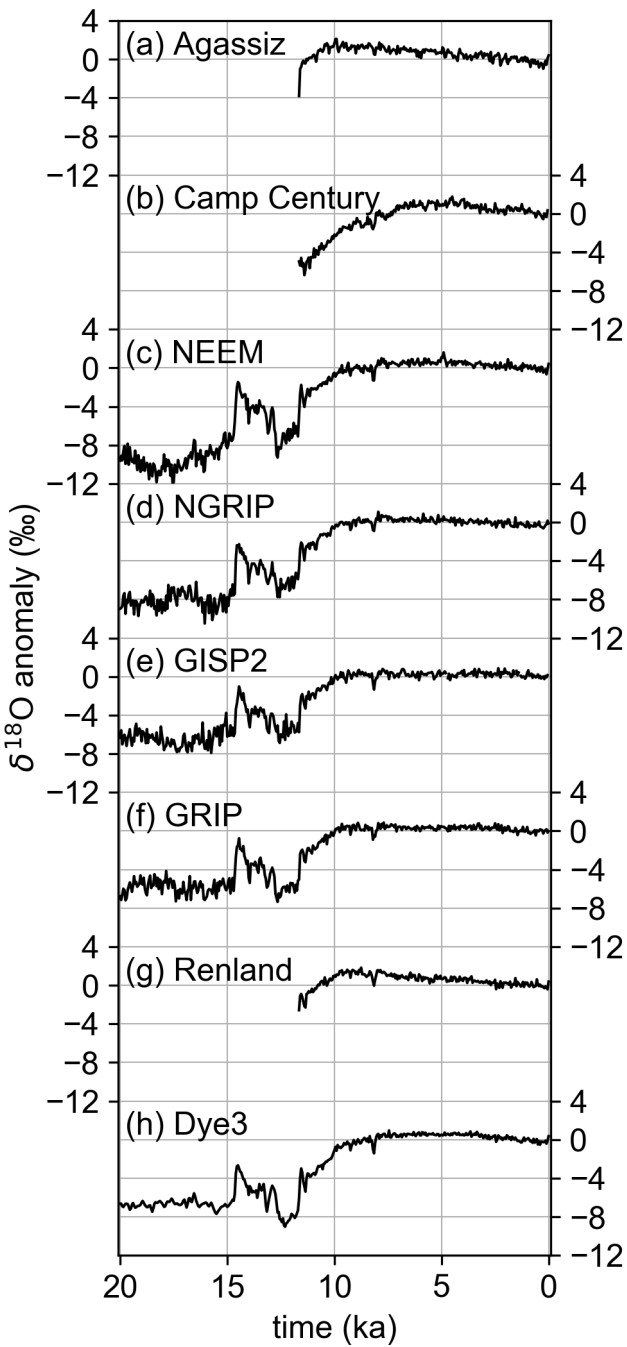

**Figure 2.** $\delta^{18}$O records assimilated into the temperature reconstruction. Records are shown as anomalies relative to the mean of 1850-2000 CE and are ordered top to bottom from northernmost to southernmost. Ice-core site names are given above each record.

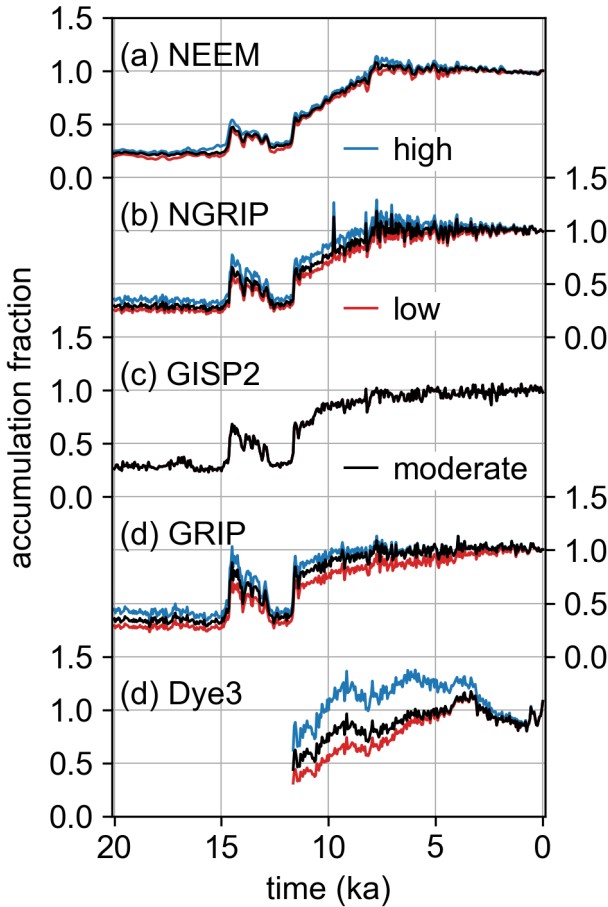

**Figure 3.** Accumulation records assimilated to reconstruct precipitation for the main reanalysis and two sensitivity scenarios. Records are shown as fractions relative to the mean of 1850-2000 CE and are ordered top to bottom from northernmost to southernmost. Black lines are the moderate records which are included in the main precipitation reanalysis, red lines are the low records which are included in the low sensitivity scenario, and blue lines are the high records which are included in the high sensitivity scenario. Note that we use the same GISP2 accumulation record for the main, high, and low scenarios. Ice-core site names are given above each set of records.

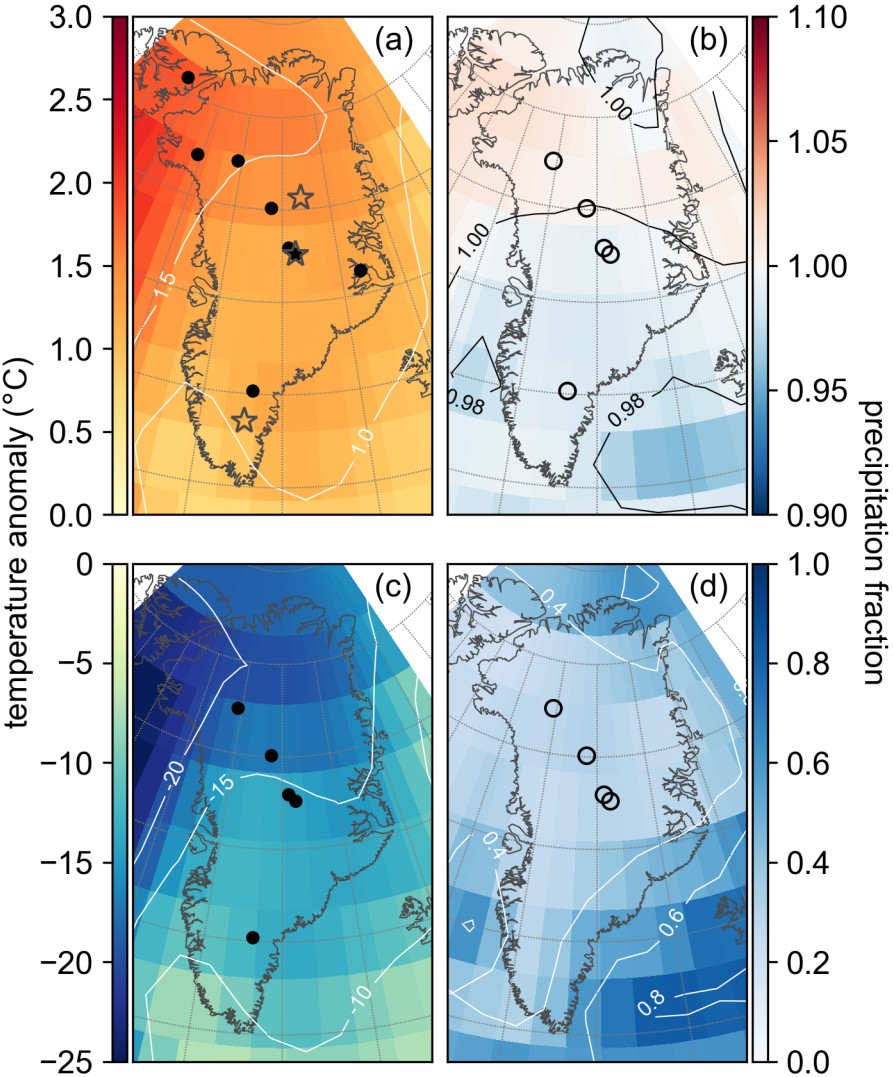

**Figure 4.** Spatial pattern of the reanalysis mean for temperature (panels (a), (c)) and precipitation (panels (b), (d)). (a) and (b) are averaged over 1,000 years around the peak warmth in the Holocene, 5.5-4.5 ka, while (c) and (d) are averaged over 5,000 years in the late glacial, 20-15 ka. Anomalies and fractions are with respect to the mean of 1850-2000 CE. Points show ice-core locations used for each reanalysis with closed circles indicating $\delta^{18}$O records and open circles indicating accumulation records. Grey stars show the locations of the EGRIP ice-core site, Summit, and South Dome, which are referenced in Figs. 5 and 10.

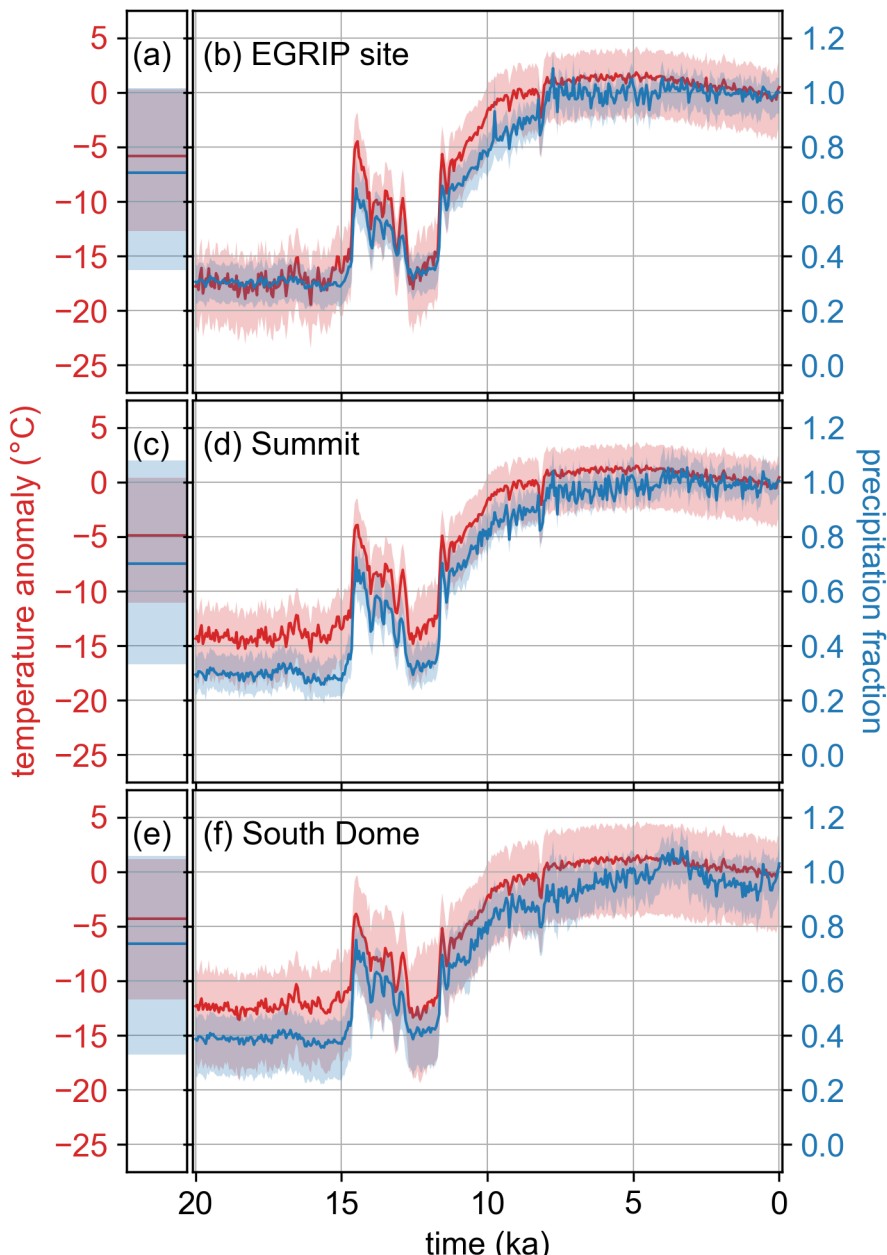

**Figure 5.** Time series of the prior (panels (a), (c) and (e)) and reanalysis (panels (b), (d) and (f)) ensemble mean and $5^{th}$ to $95^{th}$ percentile shading for temperature (red) and precipitation (blue) at three locations. Anomalies and fractions are with respect to the mean of 1850-2000 CE. (a) and (b) show these time series for the location closest to the EGRIP ice-core site, (c) and (d) show for the location closest to Summit, and (e) and (f) show for the location closest to South Dome. These locations are ordered from northernmost (top) to southernmost (bottom) and are shown on a map in Fig. 4.

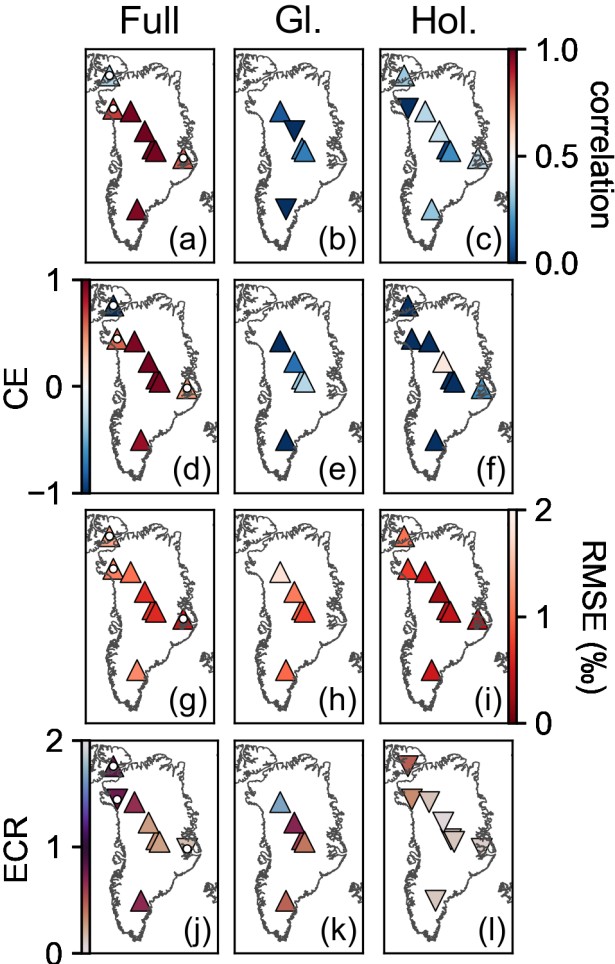

**Figure 6.** Skill metrics averaged over iterations and time for the temperature reanalysis. The first column (panels (a), (d), (g), and (j)) shows the skill metrics for the full overlap (Full) between the proxy record and reanalysis. A white dot indicates evaluation against proxy records that overlap only the Holocene (11.7-0 ka). The middle column (panels (b), (e), (h), and (k)) shows the skill metrics for a period in the glacial (Gl.) (20-15 ka), while the right column (panels (c), (f), (i), and (l)) is for a period in the Holocene (Hol.) (8-3 ka). The first row (panels (a)-(c)) reports the correlation coefficient, the second row (panels (d)-(f)) the coefficient of efficiency (CE), the third (panels (g)-(i)) the root mean square error (RMSE), and the fourth row (panels (j)-(l)) the ensemble calibration ratio (ECR). Triangle symbols pointing up indicate that the posterior ensemble evaluates better than the prior ensemble for that location and statistic. Triangle symbols pointing down indicate the opposite. We define better evaluation as correlation coefficient closer to 1, CE closer to 1, RMSE closer to 0, and ECR closer to 1.

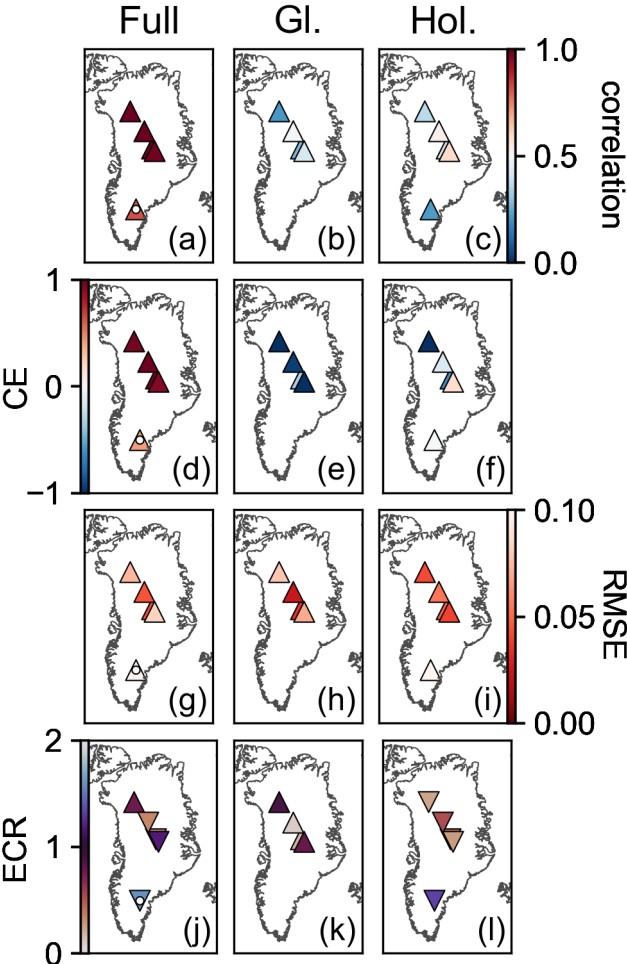

**Figure 7.** Skill metrics averaged over iterations and time for the precipitation reanalysis. The first column (panels (a), (d), (g), and (j)) shows the skill metrics for the full overlap (Full) between the proxy record and reanalysis. A white dot indicates evaluation against proxy records that overlap only the Holocene (11.7-0 ka). The middle column (panels (b), (e), (h), and (k)) shows the skill metrics for a period in the glacial (Gl.) (20-15 ka), while the right column (panels (c), (f), (i), and (l)) is for a period in the Holocene (Hol.) (8-3 ka). The first row (panels (a)-(c)) reports the correlation coefficient, the second row (panels (d)-(f)) the coefficient of efficiency (CE), the third (panels (g)-(i)) the root mean square error (RMSE), and the fourth row (panels (j)-(l)) the ensemble calibration ratio (ECR). Triangle symbols pointing up indicate that the posterior ensemble evaluates better than the prior ensemble for that location and statistic. Triangle symbols pointing down indicate the opposite. We define better evaluation as correlation coefficient closer to 1, CE closer to 1, RMSE closer to 0, and ECR closer to 1.

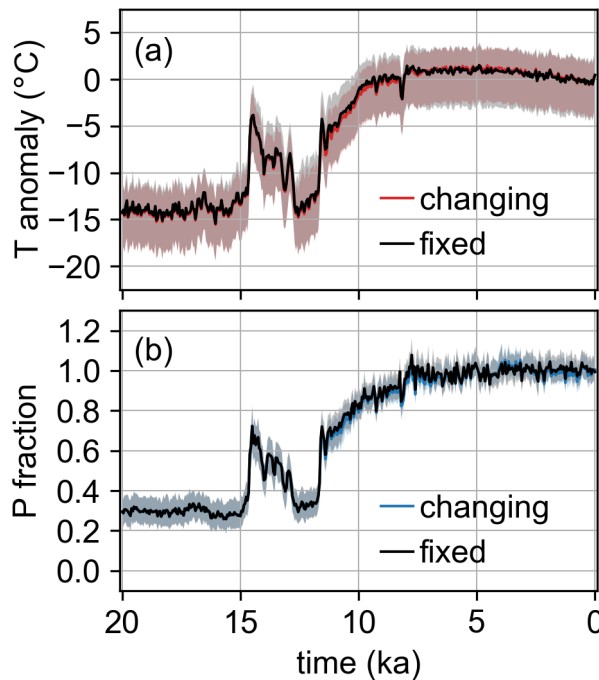

**Figure 8.** Changing (red and blue) vs. fixed (black) proxy-network for the (a) temperature (T) and (b) precipitation (P) reanalysis mean and $5^{th}$ to $95^{th}$ percentile shading. Anomalies and fractions are with respect to the mean of 1850-2000 CE. These time series are for the location closest to Summit, which is representative of the results around Greenland.

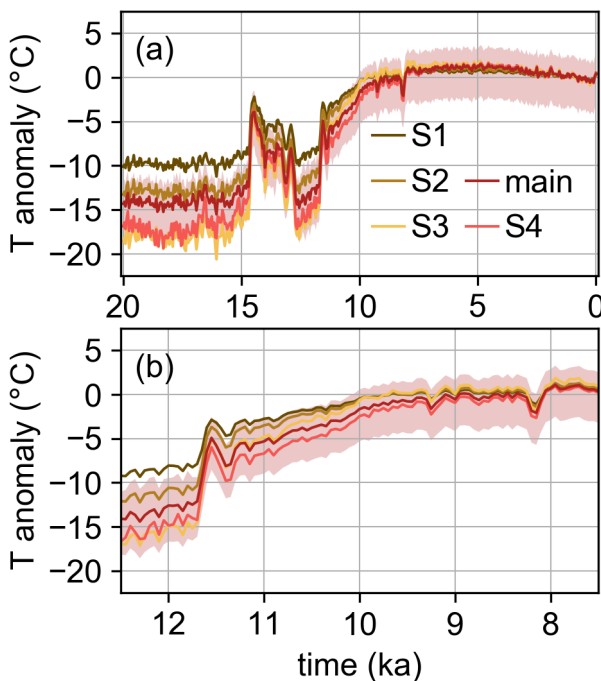

**Figure 9.** The main temperature (T) reanalysis (ensemble mean and $5^{th}$ to $95^{th}$ percentile shading) and ensemble mean for four sensitivity scenarios, S1-S4. Panel (a) shows the full 20,000 year reconstruction, while panel (b) shows a the Younger Dryas to early Holocene period (13 to 7 ka). Each sensitivity scenario reflects a different assumption about precipitation seasonality, with S1-S3 assuming a spatially-uniform seasonality and S3-S4 assuming stronger seasonality than the main reanalysis. Anomalies are with respect to the mean of 1850-2000 CE. These time series are for the location closest to Summit, which is representative of the results around Greenland.

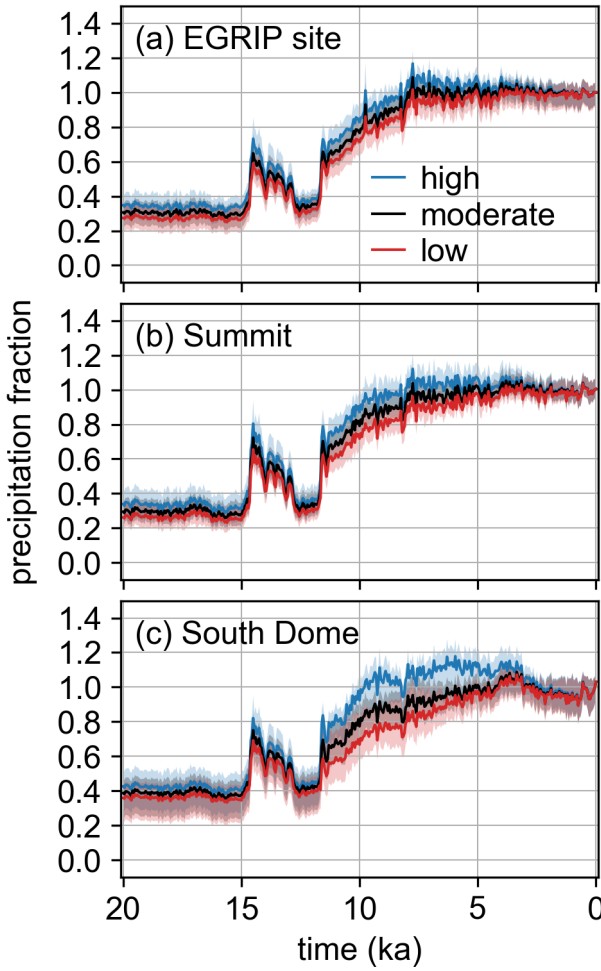

**Figure 10.** Ensemble mean and $5^{th}$ to $95^{th}$ percentile shading for the main precipitation reanalysis (black), high sensitivity scenario (blue), and low sensitivity scenario (red). Fractions are with respect to the mean of 1850-2000 CE. (a) is the time series for the location closest to the EGRIP ice-core site, (b) is closest to Summit, and (c) is closest to South Dome, which are representative of northern, central, and southern Greenland and are shown on a map in Fig. 4.

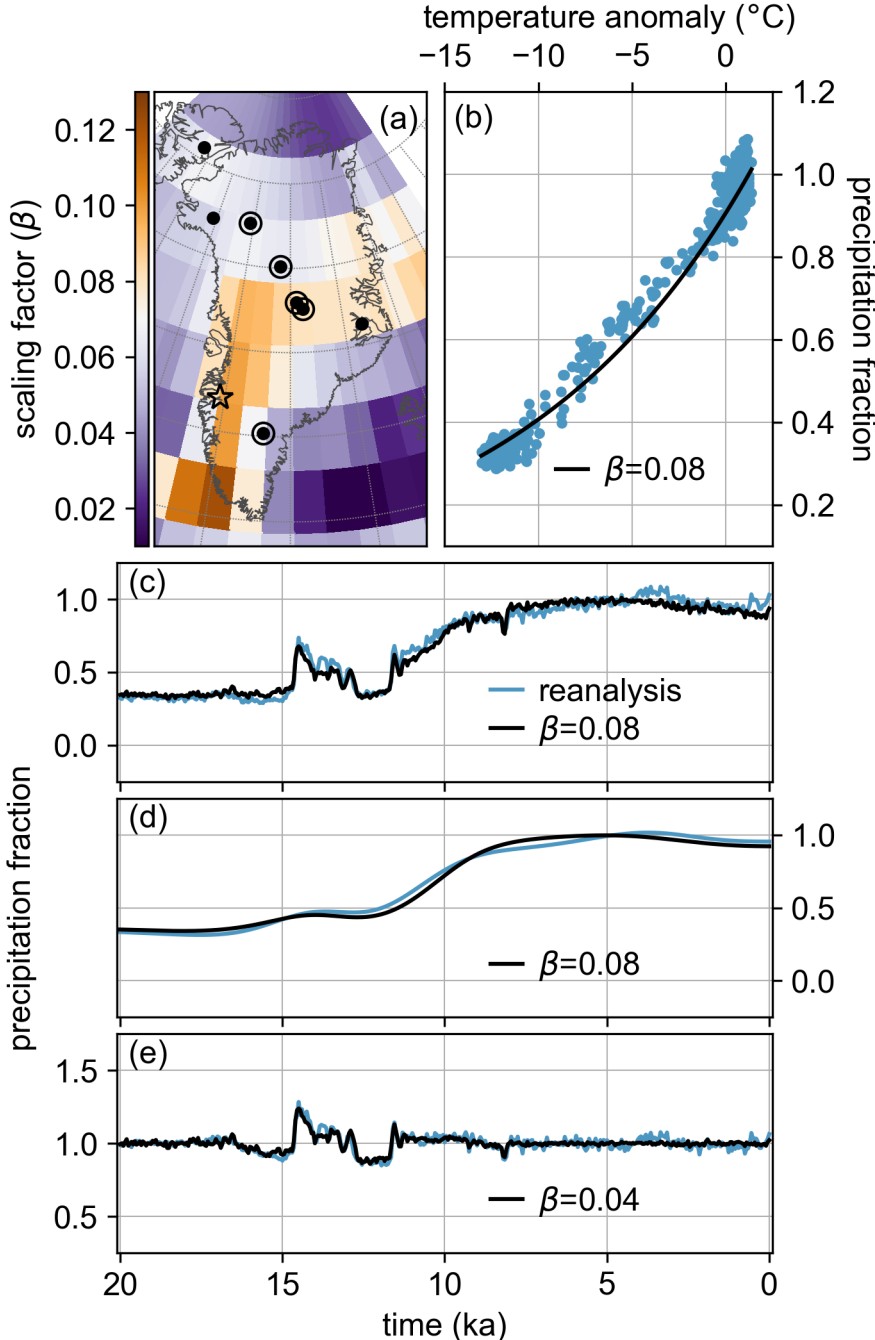

**Figure 11.** The precipitation-temperature relationship in our reanalysis. (a) shows the spatial pattern of the scaling factor ($\beta$) for the best-fit thermodynamic scaling. The colorbar is centered on 0.07, the value used by Greve et al. (2011). Points indicate ice-core locations used for each reanalysis with closed circles indicating $\delta^{18}$O records and open circles indicating accumulation records. The star is at the center of the area used in panels (b)-(e) (65°N to 68.7°N and 48.5°W to 52.5°W). (b) is a scatter plot of temperature anomaly vs. precipitation fraction from the reanalysis (blue points). The black line shows the best-fit exponential scaling. (c) shows the time series of the precipitation reanalysis (blue line) and precipitation scaled from temperature using the best-fit scaling (black line). (d) and (e) are the same as (c) except low-pass and high-pass filtered, respectively, with a cutoff frequency of 5,000 year$^{-1}$.

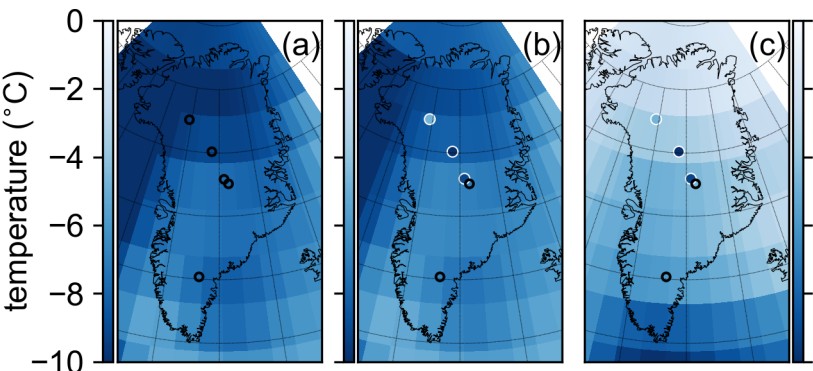

**Figure 12.** Spatial pattern of the abrupt cooling event into the Younger Dryas. Panel (a) shows results from experiment O8, assimilating all eight $\delta^{18}$O records, panel (b) shows results from experiment N3O5, assimilating all three $\delta^{15}$N-derived temperature records and the remaining five $\delta^{18}$O records (those that do not overlap with the $\delta^{15}$N sites), and panel (c) shows results from experiment N3O5_BA, which is similar to the N3O5 experiment except the prior ensemble is selected from the 1,000 years surrounding the Bølling-Allerød warming. Unfilled black circles show locations of assimilated $\delta^{18}$O records, while filled circles with white outlines show locations of assimilated $\delta^{15}$N-derived temperature records. Filled circles in panels (b) and (c) show the $\delta^{15}$N-derived temperature values as reported by Buizert et al. (2014) on the same color scale as the rest of the panel. The temporal definition of this event is the same as defined in Buizert et al. (2014).

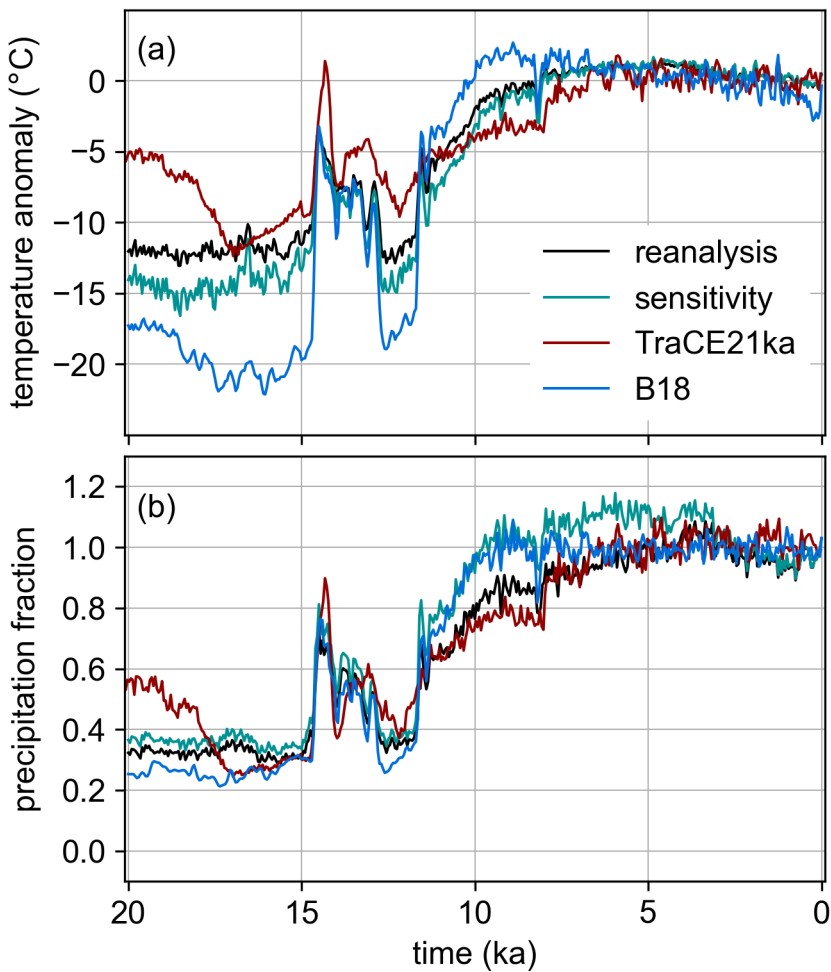

**Figure 13.** Temperature (a) and precipitation (b) reconstructions from our main reanalysis (black), our sensitivity scenarios S4 and high P (green), TraCE-21ka (red), and B18 (blue) (Buizert et al., 2018). Each reconstruction is averaged to a 50-year time resolution and averaged over a spatial domain in the Kangerlussuaq region, defined by the latitude-longitude box 65°N to 68.7°N and 48.5°W to 52.5°W, the center of which is located at the star in Fig. 11a. Temperature anomalies and precipitation fraction are defined with reference to the mean of 1850-2000 CE.