# Peer review of "Greenland temperature and precipitation over the last 20,000 years using data assimilation"

_Climate of the Past, 2019_

## Referee Comment (RC1) · Anonymous Referee #1 · 30 Jan 2020

General comments

The authors present a new reconstruction of temperature and precipitation over Greenland covering the past 20 000 years using for the first time over such a long period a data assimilation technique successfully applied recently over the past millennia. The paper is very clear, justify nearly all the choices in a very rigorous way and provides comprehensive estimates of the uncertainties. I have thus no doubt that, in addition to the new reconstruction that can be used for instance to drive ice sheet models, this study opens new fields of application of data assimilation of multi-millennial timescales.

However, I consider that the impact of the choice of the prior is not enough discussed and this issue must be addressed before publication. If I understand well, the prior ensemble is made of 100 states obtained by averaging 50 years of model data. Those

states are selected randomly over the full length of the simulation (line 175). This method is reasonable if the climate variations are weak, such as during the past millennia, but is it valid for very large changes as observed during the glacial interglacial periods? I may have missed something but, if I am right, a state obtained in the model in the late Holocene can be used to reconstruct the last glacial climate, which may be hard to justify. For instance, the authors argue that it is important to take into account the changes in seasonality of precipitation (e.g. line 233) but I wonder how this could be achieved by selecting model states that are coming from very different periods. I would suggest using as prior only years that are close to the period that is reconstructed so that only glacial states are used to reconstruct glacial climate for instance.

Specific comments

1. More specifically, still related to the prior, the authors explain (line 135) that 'For paleoclimate data assimilation, it is important that the climate simulation capture a range of possible climate states over the time period of interest '.They should thus first discuss the results of the Trace21ka simulation as it seems from Figure 12 that it underestimates the magnitude of the changes. More generally, the authors do not discuss at all the biases of the climate model. They correct for biases in the modern state by using anomalies compared to 1850-2000 (line 146) but this seems to be a small change compared to the signal during the whole simulation (line 366). Besides, the response to forcing is very different between different models as illustrated by the Paleoclimate Model Intercomparison Project. How this model behavior, which can also bias results for distant past, is influencing the results? Another way to phrase this point is that the model biases are not constant over time while the proposed correction assumes the stationarity of the biases.

2. Estimating the skill of the reconstruction compared to a constant prior (line 204) is a too low target for me. If the reconstruction was only showing a warming between the glacial period and the Holocene, it would already be skillful compared to this initial estimate and this does not require a very sophisticated technique. The skill of the

reconstruction should be evaluated against the transient Trace21ka simulation to see if the data assimilation brings some skill compared to the simulation not constrained by data

3. The authors explain at the end of the conclusion (line 485) that using a model that directly simulates isotopes would likely improve their results. It would be interesting to discuss that earlier because, for instance, they mention a different relationship between reconstructed precipitation and temperature at different time scales (line 352) but what is the potential role of a different relationship between temperature and d18O on this conclusion?

4. If I am right, when the technique described in section 2.3 is applied for the past millennia, records related to both the temperature and hydrology are assimilated together, as the covariance between the variables can bring interesting information and reduces the uncertainties. Here, it is claimed that having independent temperature and precipitation reconstructions is an advantage. This also means that precipitation and temperature changes could not be dynamically consistent in the proposed reconstruction? Maybe the authors do not want to rely on the covariance between those two variables as simulated by the climate model but they should explain why and, in that case, explain in a bit more detail the added value brought by the assimilation using this model results as prior.

5. Line 153, it is said that 'The offline method is appropriate when characteristic memory in the system is significantly shorter than the time step' (here 50 years). Is this valid here, for Bølling-Allerød and Younger Dryas events for instance?

6. Line 375. The reanalysis skill over the full period is clear compared to a constant climate but this would be informative to quantify it more precisely for the two selected 5000-year periods. Stating that it is lower than for the full period is not enough I think. From the figures 7 and 8, it seems that the CE is negative for nearly all the points. Stating line 377 that 'the reanalysis shows overall improvement over the prior ensemble' is

also a weak conclusion as discussed in point 2.

---

## Referee Comment (RC2) · Anonymous Referee #2 · 12 Feb 2020

The authors present new reconstructions of temperature and precipitation over the last 20,000 years over Greenland. For this, they apply a data assimilation technique on $\delta^{18}$O and accumulation records from Greenland ice cores and use the temperature and precipitation outputs from the TraCE-21ka simulation to extend the information to all the continent. The paper is in general clear enough for that people not having skills in data assimilation can read and understand quite easily the methodology presented in this manuscript. In my knowledge, the technique presented here is innovative for such a long period, and the different assumptions are presented and tested in a very rigorous way. This manuscript is worthy for publication in Climate of the Past, after having considered the comments below.

[Figure]

**General comments**

1. Compared to the ice cores part, which is well discussed in the Methods section and in the Supplementary Material, the results of the TraCE-21ka simulation are not discussed enough in my opinion. More details about the similarities/differences with other PMIP simulations and/or climate reconstructions for PI and LGM could be discussed for example. How do the last 1000/2000 years fit well with last millennium simulations or reconstructions from isotopic proxies? Moreover, the rapid climate transitions are not so well captured by the TraCE-21ka, especially the Younger Dryas. This point should be discussed in terms of potential consequences for the reconstructions by data assimilation. The spatial resolution (T31 and 26 atmospheric vertical levels for the atmosphere if I am not wrong) should be clearly stated, and the uncertainties related to this aspect could be discussed (even if it is mentioned later). For example, is the limited number of grid points over Greenland a problem for the paleo DA technique? The ice sheet boundary conditions are also of major importance in this type of simulation. The expected differences if a more recent ice-sheet reconstruction would be prescribed could be discussed. Last point: what about the precipitation seasonality in TraCE-21ka over Greenland? Is it consistent with observations? Is the seasonality different for Holocene and glacial periods? I guess it could have an important impact on the $\delta^{18}$O PSM and the final temperature reconstruction. . .

2. The way how the prior ensemble is made should be clarified. To avoid misunderstanding, the authors should state at line 175 that the prior is constant in time (and not later at line 206). If I understand clearly, 10 different 100-member prior ensembles are made (it should be said directly at the beginning). To form a prior ensemble, do you then randomly pick up 100 snapshots from the resampled TraCE-21ka temperature and precipitation outputs (see my minor comment for the line 146)? Or do you take randomly from the yearly TraCE-21ka outputs 50 consecutive years of data that you average in time for a member, other 50 consecutive years of model outputs for another

time period for the second member, and so on...? Or another way? Anyway, it needs clarification (in the way of the section 2.3 of Hakim et al. 2016 for example). I have the same remark as the first referee: what would be the difference if you would use, for instance, a "glacial prior" to reconstruct climate variables from glacial period and a "Holocene prior" for the warmer period instead of a constant prior for all the 20,000 years? In link with my first major remark, what would be the impact on the seasonality of precipitation, that influences the reconstructions of the authors?

**Minor comments**

Line 14: "requires understanding its sensitivity to changes..."

Line 18: I would put into brackets the terms "and arid" and "and wet".

Line 31: the spatial resolution, especially for paleoclimate simulations, brings also uncertainties.

Line 37 and passim: I think this is TraCE-21ka and not TraCE21ka.

Paragraph lines 106-116: Does the matching of $\delta^{18}$O from Dye3 to the $\delta^{18}$O record from NGRIP bring a dependency when evaluating the posterior against Dye3 $\delta^{18}$O record?

Section 2.2: I understand when you use the term "transient ice-sheet" that the prescribed ice-sheet is changed over time. But some people can misunderstand and think that it is done dynamically with a coupled ice-sheet model. I would use an expression like "prescribed transient ice-sheet boundary conditions" for example.

Line 146: What is the initial temporal resolution of TraCE-21ka outputs? Monthly mean? When you talk about "average of 50-year resolution", do you mean "resampling" every 50 model years? If you take the last 20,000 years, it makes something like 400 time steps, right?

Lines 230-232: Other model studies like Gierz et al. 2017 (JAMES) for the LIG and Cauquoin et al. 2019 (CP) for 6k-PI climates have shown that the seasonality of precipitation affects the $\delta^{18}$O-temperature relationship over Greenland.

Line 286: "that the proxy $y$ and prior estimate of the proxy $\mathcal{H}(x_b)$".

Lines 311-316: What does it give compared to the TraCE-21ka results?

Line 326: "from nearly $+2°$C in northern..."

Line 367: "has a large effect on our evaluation."

Line 380: "the ECR is..."

Line 404: the slower warming trends are hard to see. Make a zoom in the figure or give numbers.

Line 428: For S4 and high P cases, say clearly that it refers to the "sensitivity" curves on figure 12.

Line 486: you can add the reference Okazaki and Yoshimura 2017 (CP).

Line 488: Add the references Cauquoin et al. 2019 (CP) and Okazaki and Yoshimura 2019 (JGR Atmos).

Figure 4: add maybe contours for more clarity. And change the scale for the precipitation fraction at the peak warmth in the Holocene (panel b).

Figures 7,8, S3 and S4: quite normal that the correlation is improved for the full period compared to the constant prior climate state.

---

## Author Comment (AC1) · 6 Mar 2020

**Reply to Referee Comments #1 and #2**

We thank both referees for their time and insights. In the following, we address each of the referees' comments and, in most cases, provide an edited section of the paper.

**Referee #1**

**General Comment**

"The authors present a new reconstruction of temperature and precipitation over Greenland covering the past 20 000 years using for the first time over such a long period a data assimilation technique successfully applied recently over the past millennia. The paper is very clear, justify nearly all the choices in a very rigorous way and provides comprehensive estimates of the uncertainties. I have thus no doubt that, in addition to the new reconstruction that can be used for instance to drive ice sheet models, this study opens new fields of application of data assimilation of multi-millennial timescales.

However, I consider that the impact of the choice of the prior is not enough discussed and this issue must be addressed before publication. If I understand well, the prior ensemble is made of 100 states obtained by averaging 50 years of model data. Those states are selected randomly over the full length of the simulation (line 175). This method is reasonable if the climate variations are weak, such as during the past millennia, but is it valid for very large changes as observed during the glacial interglacial periods? I may have missed something but, if I am right, a state obtained in the model in the late Holocene can be used to reconstruct the last glacial climate, which may be hard to justify. For instance, the authors argue that it is important to take into account the changes in seasonality of precipitation (e.g. line 233) but I wonder how this could be achieved by selecting model states that are coming from very different periods. I would suggest using as prior only years that are close to the period that is reconstructed so that only glacial states are used to reconstruct glacial climate for instance."

**Reply to the General Comment**

The reviewer's general comment broadly concerns our selection of the prior ensemble from the entire TraCE-21ka simulation rather than from specific time periods that align with the reconstruction time. This is an excellent point, which we have thought about carefully, but had not elaborated upon in the paper. In reply, we will elaborate on this in the paper and the supplementary information, as follows below.

The reviewer states that, "If I understand well, the prior ensemble is made of 100 states obtained by averaging 50 years of model data. Those states are selected randomly over the full length of the simulation (line 175)." If we are understanding each other correctly, then yes, one state in the 100-member prior ensemble is an average over 50 years of the model data; these 100 states are selected randomly from the full length of the simulation. This implies that both glacial and Holocene states are likely to be contained within the same prior ensemble that is used to reconstruct all time steps over the last 20,000 years. To be clear, a prior ensemble could in principle contain only Holocene states; however, this is not the case for any of the ten prior ensembles we use in the paper. Thus, in reconstructing a time step in the glacial, for example, both glacial and Holocene states are part of the prior ensemble.

We agree with the reviewer that conditionally chosen prior ensembles would be preferable, but for the timescale under consideration this is not yet feasible. To explain the reasoning behind our choices in the paper, we elaborate on the pros and cons of four methods we considered before deciding on the one that we use in this study (#4).

1. For offline data assimilation (i.e., no information passed between assimilation time steps), a justifiable method for choosing the prior ensemble would be to use a 100-member ensemble of 20,000-year climate simulations. These climate simulations would be TraCE-21ka-like (i.e., results from fully-coupled GCMs at T31 resolution or higher), and have varied initial conditions, boundary conditions, and model physics. The prior ensemble for any assimilation time step would be taken from the

same time step in the climate simulations, which would lead to a prior ensemble that varies smoothly in time and is a justifiable
45  initial guess for the climate evolution over the past 20,000 years. Though this option is simple, it is not feasible because the
computational cost of running even one TraCE-21ka-like simulation remains near computational limits.

2. Given that there is only one TraCE-21ka-like simulation, another method would be to select states from TraCE-21ka that
are closest in time to the reconstruction time step. For example, if we were reconstructing the 50-year average centered on the
50  year 5,000 CE, then we would select the 100 states from TraCE-21ka that are closest in time to 5,000 CE. Given that we are
working with 50-year averages, this means we would select all the states between 7,500 and 2,500 CE. This method, which
we call the "running-window" method, provides a prior that varies smoothly in time and is a justifiable initial estimate for the
climate evolution.

55  For the running-window method, the variance of the prior ensemble would tend to be small. A prior with small variance
would lead to underweighting of the proxy records during assimilation. To avoid this issue, we could use the well-accepted ap-
proach of inflating the prior variance (Anderson and Anderson, 1999). However, the use of inflation adds an additional tunable
parameter; in this case, it would add an additional parameter per time step. Although inflation can, in principle, be constrained
using the ensemble calibration ratio (computed for excluded proxies), we have too few proxy records to meaningfully constrain
60  this parameter without overfitting.

In addition to estimating numerous inflation factors, the running-window method limits us to one estimate of the spatial
covariance structure per time step. Thus, we have no way to quantify the uncertainty associated with the prior covariance
structure. This could be fixed by expanding the running window and randomly selecting multiple prior ensembles; however, if
65  the running window is expanded enough to create meaningfully different prior ensembles, then Holocene states will leak into
glacial prior ensembles (and *vice versa*) and the method essentially becomes the method we use in the paper.

3. To reduce the number of inflation parameters, we could split TraCE-21ka into several distinct time periods. From these
time periods, we would randomly select prior ensembles that are only used for the reconstruction of associated assimilation
70  time steps. For example, if we split TraCE-21ka into glacial, transitional, and Holocene periods, then we'd make a glacial
prior ensemble that is only used to reconstruct the glacial, a transitional prior ensemble that is only used to reconstruct the
transition, and a Holocene prior ensemble that is only used to reconstruct the Holocene. This reduces the number of inflation
factors we must estimate to a total of three. A disadvantage, however, is that this makes the prior discontinuous in time, which
frequently leads to a discontinuous reconstruction. To adjust the reconstruction and make it continuous requires another source
75  of information. Such post-processing adds an extra layer of complexity.

4. The method used in our study ensures a continuous reconstruction and removes the need for inflation factors. This method
uses the same prior ensemble for all time steps (thus it is continuous) and the includes both glacial and Holocene states, which
provides enough variance to appropriately weight the proxy records (thus no inflation is needed). Though the time-invariant
80  prior is a poor estimate of the climate evolution over the last 20,000 years, the proxy records are given enough weight to result
in a posterior that captures the large climate changes. In addition, we can quantify the uncertainty associated with the spatial
covariance pattern by producing multiple posterior ensembles that each stem from a different prior ensemble. In the paper, we
use ten different prior ensembles to quantify this uncertainty. Overall, this method is both feasible and simple, thus providing
a first step in developing paleoclimate data assimilation for applications on glacial-interglacial timescales.
85

As mentioned above, we will include a discussion of these points in the revised paper.

We would also like to specifically address the following comment: "the authors argue that it is important to take into account
the changes in seasonality of precipitation (e.g. line 233) but I wonder how this could be achieved by selecting model states that
90  are coming from very different periods." We agree with the reviewer that the best way to account for changes in precipitation
seasonality is to do it in a time-varying manner, but we use a time-invariant prior for reasons given above. This means that
the mean precipitation seasonality is constant in time and determined by the states in our prior ensemble. What is new in our

paper is that for each reconstruction, we use ten different prior ensembles, which gives us ten different estimates of the mean precipitation seasonality. In addition, our approach accounts for spatial variations in precipitation seasonality.

We do not wish to mislead readers into thinking that we account for time-varying precipitation seasonality. Instead, we account for a mean precipitation seasonality, which is determined by the states in our prior ensemble. To clarify this in the paper, we will edit lines 242-244 in the paper to say the following:

With $T^*_{site}$ in our PSM, we find that the $\delta^{18}O$-$T_{site}$ slope is spatially variable. Using our prior ensembles, which are selected randomly from the full TraCE-21ka simulation, the average slope around Greenland is about 75% of modern, or about 0.5 ‰°$C^{-1}$. Thus, in using this prior ensemble, we capture the mean of the modern (i.e. high-frequency) relationship and that of the glacial-interglacial (i.e. low-frequency) effective temporal slope. Due to our use of TraCE-21ka, this mean $\delta^{18}O$-$T_{site}$ relationship is relevant both for our reconstruction period (the last 20,000 years) and for our reconstruction method, which uses a mean spatial covariance pattern from TraCE-21ka for all time steps.

**Specific Comment #1**

"More specifically, still related to the prior, the authors explain (line 135) that 'For paleoclimate data assimilation, it is important that the climate simulation capture a range of possible climate states over the time period of interest.' They should thus first discuss the results of the TraCE-21ka simulation as it seems from Figure 12 that it underestimates the magnitude of the changes. More generally, the authors do not discuss at all the biases of the climate model. They correct for biases in the modern state by using anomalies compared to 1850-2000 (line 146) but this seems to be a small change compared to the signal during the whole simulation (line 366). Besides, the response to forcing is very different between different models as illustrated by the Paleoclimate Model Intercomparison Project. How this model behavior, which can also bias results for distant past, is influencing the results? Another way to phrase this point is that the model biases are not constant over time while the proposed correction assumes the stationarity of the biases."

**Reply to Specific Comment #1**

The reviewer makes a good point that we should expand our discussion of the TraCE-21ka simulation, especially with respect to model biases.

Model-bias corrections rely on observations. In the modern, there are numerous observations from ground-based and satellite systems. From the past, there are relatively few observations, which are from proxy records. Given an assumption of stationary model bias (unchanging in time), we can use the modern observations to compute the bias correction; however, given an assumption of non-stationary model bias, we must rely on paleoclimate proxy records as well. In our paper, we assume a stationary model bias and apply the delta-change method (Teutschbein and Seibert, 2012). This leaves the proxy records available for data assimilation. Ideally, we would subsample the proxy records and use one subsample for data assimilation, another to correct for model bias *a priori*, and another to assess the influence of model bias on our reconstruction *a posteriori*. This, however, is not possible with a small number of proxy records. Therefore, we have chosen to reserve all proxy records for data assimilation and to assume a stationary bias correction.

In addition to a mean bias in the model, there may also be biases in the variance. The reviewer specifically points out this issue with TraCE-21ka: it has a small glacial-Holocene climate change relative to the ice-core records. This is a bias that is best addressed with information from proxy records, which we have reserved as independent observations for data assimilation.

As the reviewer alluded to, another opportunity to assess the influence of model bias is to simply select our prior ensemble from a different model. By examining how the results are affected by a variety of different model simulations, we could assess the sensitivity of our results to different models (and thus model biases). Doing this analysis in a rigorous manner is not yet possible because TraCE-21ka is the only-available continuous 20,000-year simulation completed with a fully-couple GCM at

a T31 resolution or higher.

We will include this discussion as a paragraph in section 2.2 (lines 132-147):

> To correct for model bias, we assume it is stationary in time and apply the delta-change method (Teutschbein and Seibert, 2012) by taking the anomaly of temperature and the fraction of precipitation relative to the mean of our reference period (1850-2000 CE). Our assumption of a stationary model bias is required because, with a small number of proxy records, we cannot afford to subsample them for the purposes of bias correction, data assimilation, and evaluation.

**Specific Comment #2**

"Estimating the skill of the reconstruction compared to a constant prior (line 204) is a too low target for me. If the reconstruction was only showing a warming between the glacial period and the Holocene, it would already be skillful compared to this initial estimate and this does not require a very sophisticated technique. The skill of the reconstruction should be evaluated against the transient TraCE-21ka simulation to see if the data assimilation brings some skill compared to the simulation not constrained by data."

**Reply to Specific Comment #2**

We agree with the reviewer that we should compare our reconstruction skill against that of other 20,000-year reconstructions or simulations. The skill metrics are most comparable if there is either a $\delta^{18}O$ or T* variable available in the reconstruction or simulation. Thus, this comparison is straightforward with TraCE-21ka, but not for other reconstructions like that of Buizert et al. (2018). We have computed the correlation coefficient, coefficient of efficiency (CE), and root mean square error (RMSE) for TraCE-21ka, and will add this information both as text and figures.

We will add the following text to line 209 (the end of a paragraph) to explain our evaluation against TraCE-21ka:

> We additionally compute the correlation coefficient, CE, and RMSE on the TraCE-21ka simulation as compared to the proxy records.

We will also edit section 4.1, which discusses the results of the evaluation. Please see our reply to specific comment #6 for the edited text (lines 284 to 340 of this document) and additional figures.

**Specific Comment #3**

"The authors explain at the end of the conclusion (line 485) that using a model that directly simulates isotopes would likely improve their results. It would be interesting to discuss that earlier because, for instance, they mention a different relationship between reconstructed precipitation and temperature at different time scales (line 352) but what is the potential role of a different relationship between temperature and $\delta^{18}O$ on this conclusion?"

**Reply to Specific Comment #3**

We agree with the reviewer that the isotope-temperature relationship is an interesting one to explore, a topic which we touch on with our sensitivity experiments described in section 2.3.1 of the paper (lines 245-255). We tested the sensitivity of our results to different isotope-temperature relationships, primarily the magnitude of the slope in a linear relationship, but also the spatial pattern of that slope. The reconstructions resulting from these experiments are discussed in section 4.2 and shown in Fig. 10.

An isotope-enabled model, as we mention in the conclusion, would provide another estimate of the magnitude and spatial pattern of the isotope-temperature relationship. The advantage of an isotope-enabled model is that it incorporates the variety of processes that can affect water isotope ratios, whereas, with our experiments, we focus on one primary process: precipi-

180  tation seasonality. Isotope-enabled models may have biases, and the experiments we perform are important for assessing the sensitivity of reconstructions to the assumed isotope-temperature relationships. As the reviewer suggests, we will bring up this discussion earlier in the paper by adding the following sentences to the last paragraph in section 2.3.1 (lines 245-255):

185  > These sensitivity tests are equivalent to testing different assumptions about the $\delta^{18}$O-temperature relationship. The availability of a 20,000 year-long isotope-enabled climate simulation would allow us to determine this relationship from model physics, which incorporate a variety of processes that can affect water isotopes, including precipitation seasonality.

190  The reviewer also astutely suggests that we test the effect of the isotope-temperature relationship on the precipitation-temperature relationship. We have analyzed the temperature-precipitation relationship between all possible combinations of our main reanalysis and sensitivity reanalyses and will add the following paragraph to the paper at the end of section 4.2 (line 413). Table S1 is included at the end of this document.

195  > Finally, we test how these sensitivity results affect the scaling factor ($\beta$) in the precipitation-temperature relationship. We pair the five temperature reconstructions (main, S1-S4) and three precipitation reconstructions (low, moderate, and high) into fifteen possible combinations and conduct the same analysis as described above in Sect. 3. Across these fifteen combinations, we find that the spatial pattern of $\beta$ is robust (Fig. 6a). The exact magnitude depends primarily on the temperature reconstruction and how cold it is in the glacial, with colder temperatures giving lower $\beta$ values. To a lesser degree, the magnitude also depends on the precipitation reconstruction, with wetter scenarios giving lower $\beta$ values. As previously, we find that the low-pass filtered datasets have the same or nearly the same $\beta$ value as the unfiltered dataset, while the high-pass filtered datasets have consistently lower $\beta$ values. As an example of this consistency, Table S1 shows the $\beta$ value found for the Kangerlussuaq region for all fifteen combinations and three filtering options.

**Specific Comment #4**

205  "If I am right, when the technique described in section 2.3 is applied for the past millennia, records related to both the temperature and hydrology are assimilated together, as the covariance between the variables can bring interesting information and reduces the uncertainties. Here, it is claimed that having independent temperature and precipitation reconstructions is an advantage. This also means that precipitation and temperature changes could not be dynamically consistent in the proposed reconstruction? Maybe the authors do not want to rely on the covariance between those two variables as simulated by the climate model but they should explain why and, in that case, explain in a bit more detail the added value brought by the assimilation
210  using this model results as prior."

**Reply to Specific Comment #4**

The reviewer is correct that previous work (e.g., Hakim et al., 2016; Tardif et al., 2019) has used a similar method to assimilate both temperature and precipitation-sensitive proxy records into a reconstruction of multiple climate variables over the last millennium. We agree with the reviewer that our choice to separate the proxy records and reconstruct temperature and precipitation
215  independently requires more explanation in the paper.

We have chosen to independently reconstruct temperature and precipitation because the relationship between these two variables is highly non-linear and non-stationary over the last 20,000 years. Precipitation generally follows the expected thermodynamic relationship on glacial-interglacial timescales (Robin, 1977); however, there have been times in the last 20,000
220  years, as shown by ice-core records, that precipitation has deviated significantly (even having opposite sign) from the thermodynamic expectation (Cuffey and Clow, 1997). For this reason, we choose not to impose the climate-model-derived mean temperature-precipitation relationship on our reconstruction.

We would also like to address the reviewer's concern that the temperature and precipitation may not be dynamically consistent given our method of reconstructing them independently. The reviewer is correct that our reconstructions may not achieve dynamic consistency through the modeled relationships. However, the reconstructions are dynamically consistent insofar as the empirical $\delta^{18}$O and accumulation records from the ice-core records are dynamically consistent, as they must be.

We will make edits to the paragraph at lines 175-183 to reflect this discussion. Note that some edits to this section are in reply to the general comment #2 from Referee #2 (lines 409 to 410 of this document). New text is in *italics*.

The prior ensemble is an initial estimate of possible climate states, which we form using 100 randomly-chosen 50-year averages from the TraCE-21ka simulation. *States from both the glacial and the Holocene make up a prior ensemble. The same prior is used for all time steps in the reconstruction, leading to a prior that is constant in time.* Proxy records are assimilated into the prior using Eq. 1, which produces the posterior ensemble, a new estimate of possible climate states. We assimilate $\delta^{18}$O to reconstruct temperature and separately assimilate accumulation to reconstruct precipitation. *This approach maintains independence between temperature and precipitation, which avoids imposing linearity and stationarity on the relationship between these two variables. As Cuffey and Clow (1997) show, not only is this relationship non-linear on long timescales but it is also not well-approximated by simple thermodynamic expectations. Separating these variables ensures that the relationship between temperature and precipitation is consistent with the empirical relationship between $\delta^{18}$O and accumulation from ice cores, rather than being derived from the climate model.*

*We repeat the data assimilation process* over multiple iterations, with each iteration using one of ten different 100-member prior ensembles and excluding one proxy record. *Each of the ten prior ensembles is made up of a different random selection of 50-year averages from TraCE-21ka. Thus, each prior ensemble has a different variance and spatial covariance structure.* Each proxy record is excluded from a total of ten iterations, where each of these iterations uses a different one of the ten prior options. *Every iteration is uniquely identifiable by which prior ensemble is used and which proxy record is excluded.* For a reanalysis, the total number of iterations is thus ten times the number of proxy records, such that for temperature we have 80 iterations and for precipitation we have 50 iterations. A reanalysis is a compilation of the *100-member* posterior ensembles from these iterations, resulting in a temperature reanalysis having 8,000 ensemble members and a precipitation reanalysis having 5,000 ensemble members.

**Specific Comment #5**

"Line 153, it is said that 'The offline method is appropriate when characteristic memory in the system is significantly shorter than the time step' (here 50 years). Is this valid here, for Bølling-Allerød and Younger Dryas events for instance?"

**Reply to Specific Comment #5**

The reviewer raises an excellent point that periods with strong forcing, such as the Bølling-Allerød and Younger Dryas, have a longer characteristic memory than periods with weaker forcing, such as the late Holocene. In general, models have little predictive skill on decadal or longer timescales, except perhaps for areas strongly influenced by the thermohaline circulation (Latif and Keenlyside, 2011, and references therein). During periods of strong forcing, model predictive skill may increase if both the forcing and the response are appropriately represented by the model; however, the predictive skill may also decrease if model uncertainty is large (Hawkins and Sutton, 2009). Rather than assuming our method is appropriate for all times in the last 20,000 years, we would ideally make use of an ensemble of long climate simulations or online data assimilation; however, both alternative approaches are not feasible at this time due to computational cost.

To clarify this point in the paper, we will edit the first paragraph of section 2.3 (lines 149 to 153) to say the following (new text is in *italics*):

To combine the ice-core data and climate-model data, we use an offline data assimilation method similar to that described in Hakim et al. (2016). "Offline" refers to the absence of a forecast model that evolves the climate state between *assimilation* time steps, such that in offline data assimilation the same initial climate state is used for every time step. The offline method is appropriate when model predictive-skill is significantly shorter than the *assimilation* time step (Hakim et al., 2016, and references therein). *Model predictive-skill is generally poor on decadal to longer timescales (Latif and Keenlyside, 2011, and references therein) except possibly during times of strong forcing, such as the Bølling-Allerød (14.7-12.7 ka) and the Younger Dryas (12.7-11.7 ka) (Hawkins and Sutton, 2009). Because each of our time steps is an average over 50 years, as dictated by the resolution to which we average the proxy records, the offline method is appropriate except possibly during these large-forcing events. For these events, an online method would be most appropriate (assuming that the models correctly capture both the forcing and the response); however, online data assimilation over glacial-interglacial cycles is not feasible at this time given the computational cost.*

**Specific Comment #6**

"Line 375. The reanalysis skill over the full period is clear compared to a constant climate but this would be informative to quantify it more precisely for the two selected 5000-year periods. Stating that it is lower than for the full period is not enough I think. From the figures 7 and 8, it seems that the CE is negative for nearly all the points. Stating line 377 that 'the reanalysis shows overall improvement over the prior ensemble' is also a weak conclusion as discussed in point 2."

**Reply to Specific Comment #6**

We agree with the reviewer that the evaluation over the two 5,000-year periods warrants more discussion. In section 4.1, we will insert this discussion as well as the comparison of our reconstruction skill to that of TraCE-21ka (in reply to specific comment #2, lines 146 to 165). New text is in *italics* and new figures, S5-S8, are at the end of this document.

4.1 Independent proxy evaluation

[revised manuscript text omitted]

**Referee #2**

"The authors present new reconstructions of temperature and precipitation over the last 20,000 years over Greenland. For this, they apply a data assimilation technique on $\delta^{18}O$ and accumulation records from Greenland ice cores and use the temperature and precipitation outputs from the TraCE-21ka simulation to extend the information to all the continent. The paper is in general clear enough for that people not having skills in data assimilation can read and understand quite easily the methodology presented in this manuscript. In my knowledge, the technique presented here is innovative for such a long period, and the different assumptions are presented and tested in a very rigorous way. This manuscript is worthy for publication in Climate of the Past, after having considered the comments below."

**General Comment #1**

"Compared to the ice cores part, which is well discussed in the Methods section and in the Supplementary Material, the results of the TraCE-21ka simulation are not discussed enough in my opinion. More details about the similarities/differences with other PMIP simulations and/or climate reconstructions for PI and LGM could be discussed for example. How do the last 1000/2000 years fit well with last millennium simulations or reconstructions from isotopic proxies? Moreover, the rapid climate transitions are not so well captured by the TraCE-21ka, especially the Younger Dryas. This point should be discussed in terms of potential consequences for the reconstructions by data assimilation. The spatial resolution (T31 and 26 atmospheric vertical levels for the atmosphere if I am not wrong) should be clearly stated, and the uncertainties related to this aspect could be discussed (even if

it is mentioned later). For example, is the limited number of grid points over Greenland a problem for the paleo DA technique? The ice sheet boundary conditions are also of major importance in this type of simulation. The expected differences if a more recent ice-sheet reconstruction would be prescribed could be discussed. Last point: what about the precipitation seasonality in TraCE-21ka over Greenland? Is it consistent with observations? Is the seasonality different for Holocene and glacial periods? I guess it could have an important impact on the $\delta$18O PSM and the final temperature reconstruction. . . "

**Reply to General Comment #1**

We agree with the referee that our discussion of the TraCE-21ka simulation is limited compared to our discussion of the ice-core records. This is primarily because TraCE-21ka is described extensively in the literature (e.g., Liu et al., 2009, 2012; He, 2011; He et al., 2013), whereas a number of aspects of the ice-core network, particularly the accumulation records, are novel or have been discussed little in previous work. In section 2.2 of the paper, we do discuss attributes of TraCE-21ka that are especially relevant to our method, including, the glacial to Holocene mean state changes, temperature and precipitation seasonality, and the ice-sheet boundary conditions. We agree with the referee that we are missing a statement about the spatial resolution of TraCE-21ka (indeed it is T31) and more details concerning seasonality and the ice sheets. We will include these details in the revised paper.

We want to emphasize that our paper represents the first attempt to use the ensemble Kalman filter approach to reconstruct climate over glacial-interglacial timescales. Ideally, there would be more 20,000-year (or longer) TraCE-21ka-like simulations (i.e., from fully-coupled GCMs at T31 resolution or higher). With other simulations from different models, we could address the influence of model bias, spatial resolution, boundary conditions, and initial conditions (for a discussion of model bias, see our reply to Referee #1's specific comment #1, lines 116 to 145 of this document). If we had other simulations, then we agree with the referee that it would warrant a comparison between simulations and a discussion of how the differences affect our results. With only one TraCE-21ka-like simulation available, we cannot conduct a meaningful comparison with how other climate-model simulations would affect our results. Our focus instead is on demonstrating that the method is feasible and skillful. Thus, we do not see it as productive to elaborate on how TraCE-21ka compares with PMIP or other simulations. In addition, the previous literature has extensively interrogated the results of the TraCE-21ka simulation (e.g., He, 2011; Buizert et al., 2014; Pedro et al., 2016; Zhang et al., 2017, 2018; Marsicek et al., 2018). We will add new text to refer the reader to this previous work.

The referee additionally suggests that we include a discussion of how well TraCE-21ka captures rapid climate transitions, such as the Younger Dryas. We agree that this would be necessary if we were to choose our prior ensemble exclusively from time periods that are close to our reconstruction time (for examples of this, see our reply to Referee #1's general comment, lines 21 to 105 of this document); however, our prior ensemble is time-invariant and thus our method is insensitive to the temporal evolution of TraCE-21ka (as long as the variance and spatial covariance structures of TraCE-21ka are preserved).

**General Comment #2**

"The way how the prior ensemble is made should be clarified. To avoid misunderstanding, the authors should state at line 175 that the prior is constant in time (and not later at line 206). If I understand clearly, 10 different 100-member prior ensembles are made (it should be said directly at the beginning). To form a prior ensemble, do you then randomly pick up 100 snapshots from the resampled TraCE-21ka temperature and precipitation outputs (see my minor comment for the line 146)? Or do you take randomly from the yearly TraCE-21ka outputs 50 consecutive years of data that you average in time for a member, other 50 consecutive years of model outputs for another time period for the second member, and so on. . .? Or another way? Anyway, it needs clarification (in the way of the section 2.3 of Hakim et al. 2016 for example). I have the same remark as the first referee: what would be the difference if you would use, for instance, a "glacial prior" to reconstruct climate variables from glacial period and a "Holocene prior" for the warmer period instead of a constant prior for all the 20,000 years? In link with my first major remark, what would be the impact on the seasonality of precipitation, that influences the reconstructions of the authors?"

**Reply to General Comment #2**

We thank the referee for calling our attention to these points of confusion. We will clearly state earlier in the paper that the prior is constant in time. We will also clarify how we average TraCE-21ka in line 146 of the paper by making the following edits (new text is in *italics*):

405

> We then average to a 50-year resolution, as for the ice-core records. *This averaging results in 401 time steps spaced 50 years apart.*

We will also reorganize and edit the paragraph starting at line 175 to clarify how we form our ten different 100-member
410  prior ensembles. Please see lines 232 to 251 in this document for the edits.

The referee additionally raises concerns over our random selection of the prior ensemble from the full TraCE-21ka simulation and the effect this has on the seasonality of precipitation and our reconstructions. For this we refer to our reply to the general comments from Referee #1 (lines 21 to 105 in this document).

415  **Minor Comments**

We thank the referee for these specific edits and the effort to make the paper clearer and more concise.

**Line 14: "requires understanding its sensitivity to changes. . ."**
Thank you. This edit will be made in the revised paper.
420

**Line 18: I would put into brackets the terms "and arid" and "and wet".**
We think that it is important to equally emphasize the thermal and hydrologic differences between glacial and interglacial periods. For this reason, we'd rather not make the suggested revision.

425  **Line 31: the spatial resolution, especially for paleoclimate simulations, brings also uncertainties.**
We will edit this to say: "In contrast, climate-model simulations are spatially-complete estimates of past climate, but they are subject to uncertainty due to model dynamics, boundary conditions, and spatial resolution."

**Line 37 and passim: I think this is TraCE-21ka and not TraCE21ka.**
430  We will make this change throughout the paper.

**Paragraph lines 106-116: Does the matching of $\delta^{18}$O from Dye3 to the $\delta^{18}$O record from NGRIP bring a dependency when evaluating the posterior against Dye3 $\delta^{18}$O record?**
While dating uncertainty is non-zero, it is clear from multiple independent lines of evidence that the major $\delta^{18}$O variations
435  in all Greenland cores are nearly synchronous. Most important is the variance, rather than the timing, of $\delta^{18}$O, and this is not affected by the minor changes to the timescales imposed here. (Note that our matching of the $\delta^{18}$O from Dye3 to the $\delta^{18}$O record from NGRIP is similar to the methods used to create the GICC05 depth-age scale and apply it to other ice cores.)

**Section 2.2: I understand when you use the term "transient ice-sheet" that the prescribed ice-sheet is changed over**
440  **time. But some people can misunderstand and think that it is done dynamically with a coupled ice-sheet model. I would use an expression like "prescribed transient ice-sheet boundary conditions" for example.**
We will change the sentence over lines 141-142 to say: "TraCE-21ka also includes prescribed transient ice-sheets as a boundary condition, the transient nature of which is important for capturing the influence of elevation change on the ice-core records."

445  **Line 146: What is the initial temporal resolution of TraCE-21ka outputs? Monthly mean? When you talk about "average of 50-year resolution", do you mean "resampling" every 50 model years? If you take the last 20,000 years, it**

**makes something like 400 time steps, right?**

We thank the referee for calling our attention to this point of confusion. We will edit section 2.2 to clarify the initial temporal resolution of TraCE-21ka (which is monthly) and what we mean by averaging it to a 50-year resolution. The referee is correct that we end up with about 400 time steps after averaging. Our method is to take 50 consecutive years (600 months) of TraCE-21ka and to average them. No year (or month) is used in more than one 50-year average.

450

**Lines 230-232: Other model studies like Gierz et al. 2017 (JAMES) for the LIG and Cauquoin et al. 2019 (CP) for 6k-PI climates have shown that the seasonality of precipitation affects the $\delta^{18}$O-temperature relationship over Greenland.**

455 We thank the referee for this comment. We did not mean to imply that Werner et al. (2000) is the only modeling study that has shown that precipitation seasonality affects the $\delta^{18}$O-temperature relationship in Greenland. We will add the recommended citations.

**Line 286: "that the proxy y and prior estimate of the proxy H(xb)".**

460 Thank you. This edit will be made in the revised paper.

**Lines 311-316: What does it give compared to the TraCE-21ka results?**

We agree with the referee that it's important to compare our results to that of TraCE-21ka; however, we have made a point of saving comparisons with TraCE-21ka and Buizert et al. (2018) for the discussion (section 5).

465

**Line 326: "from nearly +2 °C in northern. . ."**

Thank you. This edit will be made in the revised paper.

**Line 367: "has a large effect on our evaluation."**

470 Thank you. This edit will be made in the revised paper.

**Line 380: "the ECR is. . ."**

Thank you. This edit will be made in the revised paper.

475 **Line 404: the slower warming trends are hard to see. Make a zoom in the figure or give numbers.**

Thank you for the suggestion. We have edited Fig. 10 to show a zoom-in on the Younger Dryas to Holocene transition. Please see the edited version at the end of this document.

**Line 428: For S4 and high P cases, say clearly that it refers to the "sensitivity" curves on figure 12.**

480 Thank you, we will clarify that "sensitivity" in the Fig. 12 legend label refers to S4 and high P.

**Line 486: you can add the reference Okazaki and Yoshimura 2017 (CP).**

We agree that this is a relevant citation and will add it to the revised paper.

485 **Line 488: Add the references Cauquoin et al. 2019 (CP) and Okazaki and Yoshimura 2019 (JGR Atmos).**

We agree that these are relevant citations and will add them to the revised paper.

**Figure 4: add maybe contours for more clarity. And change the scale for the precipitation fraction at the peak warmth in the Holocene (panel b).**

490 Thank you for the suggestion. We have added contours for clarification and changed the color scale in panel (b). Please see the edited version at the end of this document.

**Figures 7, 8, S3, and S4: quite normal that the correlation is improved for the full period compared to the constant prior climate state.**

495 Yes, we agree and have stated this in the paper (lines 205-209). If the referee thinks it is necessary, we can state it again later

in the text or in the figure captions. We will additionally be comparing the skill of our results to TraCE-21ka, as suggested by Referee #1.

**Figures**

**Table S1.** Scaling factors ($\beta$) for the temperature-precipitation relationship in the Kangerlussuaq region. The results for the main reanalysis are in bold.

| Temperature scenario | Precipitation scenario | No Filtering | Low-Pass (5,000 years$^{-1}$) | High-Pass (5,000 years$^{-1}$) |
|---|---|---|---|---|
| Main | Low | 0.09 | 0.09 | 0.04 |
| **Main** | **Moderate** | **0.08** | **0.08** | **0.04** |
| Main | High | 0.08 | 0.08 | 0.05 |
| S1 | Low | 0.12 | 0.11 | 0.05 |
| S1 | Moderate | 0.11 | 0.11 | 0.06 |
| S1 | High | 0.11 | 0.11 | 0.07 |
| S2 | Low | 0.09 | 0.09 | 0.04 |
| S2 | Moderate | 0.09 | 0.08 | 0.04 |
| S2 | High | 0.09 | 0.08 | 0.05 |
| S3 | Low | 0.06 | 0.06 | 0.03 |
| S3 | Moderate | 0.06 | 0.06 | 0.03 |
| S3 | High | 0.06 | 0.06 | 0.04 |
| S4 | Low | 0.07 | 0.07 | 0.03 |
| S4 | Moderate | 0.07 | 0.07 | 0.03 |
| S4 | High | 0.07 | 0.07 | 0.04 |

[Figure]

**Figure S5.** Temperature skill metrics for TraCE-21ka simulation. The first column (panels (a), (d), and (g)) shows the skill metrics for the full overlap (Full) between the proxy record and reanalysis. A white dot indicates evaluation against proxy records that overlap only the Holocene (11.7-0 ka). The middle column (panels (b), (e), and (h)) shows the skill metrics for a period in the glacial (Gl.) (20-15 ka), while the right column (panels (c), (f), and (i)) is for a period in the Holocene (Hol.) (8-3 ka). The first row (panels (a)-(c)) reports the correlation coefficient, the second row (panels (d)-(f)) the coefficient of efficiency (CE), and the third (panels (g)-(i)) the root mean square error (RMSE). Triangle symbols pointing up indicate that the posterior ensemble evaluates better than TraCE-21ka for that location and statistic. Triangle symbols pointing down indicate the opposite. A result is considered improved where the correlation coefficient closer to 1, CE closer to 1, and RMSE closer to 0.

[Figure]

**Figure S6.** Precipitation skill metrics for TraCE-21ka simulation. The first column (panels (a), (d), and (g)) shows the skill metrics for the full overlap (Full) between the proxy record and reanalysis. A white dot indicates evaluation against proxy records that overlap only the Holocene (11.7-0 ka). The middle column (panels (b), (e), and (h)) shows the skill metrics for a period in the glacial (Gl.) (20-15 ka), while the right column (panels (c), (f), and (i)) is for a period in the Holocene (Hol.) (8-3 ka). The first row (panels (a)-(c)) reports the correlation coefficient, the second row (panels (d)-(f)) the coefficient of efficiency (CE), and the third (panels (g)-(i)) the root mean square error (RMSE). Triangle symbols pointing up indicate that the posterior ensemble evaluates better than TraCE-21ka for that location and statistic. Triangle symbols pointing down indicate the opposite. A result is considered improved where the correlation coefficient closer to 1, CE closer to 1, and RMSE closer to 0.

[Figure]

**Figure S7.** Difference in skill metrics, for temperature, between the posterior ensemble (averaged over iterations and time) and the TraCE-21ka simulation (difference = posterior − TraCE-21ka). Description of individual panels as in Figure S6.

[Figure]

**Figure S8.** Difference in skill metrics, for precipitation, between the posterior ensemble (averaged over iterations and time) and the TraCE-21ka simulation (difference = posterior – TraCE-21ka). Description of individual panels as in Figure S7.

[Figure]

**Figure 10.** The main temperature (T) reanalysis (ensemble mean and $5^{th}$ to $95^{th}$ percentile shading) and ensemble mean for four sensitivity scenarios, S1-S4. Each sensitivity scenario reflects a different assumption about precipitation seasonality, with S1-S3 assuming a spatially-uniform seasonality and S3-S4 assuming stronger seasonality than the main reanalysis. Anomalies are with respect to the mean of 1850-2000 CE. These time series are for the location closest to Summit, which is representative of the results around Greenland. Panel (a) shows the full time period, and panel (b) the Younger-Dryas through the Holocene, showing that S1, S2, and S3 warm more quickly than the main reconstruction and S4.

[Figure]

**Figure 4.** Spatial pattern of the reanalysis mean for temperature (panels (a), (c)) and precipitation (panels (b), (d)) with contours for clarity. (a) and (b) are averaged over 1,000 years around the peak warmth in the Holocene, 5.5-4.5 ka, while (c) and (d) are averaged over 5,000 years in the late glacial, 20-15 ka. Anomalies and fractions are with respect to the mean of 1850-2000 CE. Points show ice-core locations used for each reanalysis with closed circles indicating $\delta^{18}$O records and open circles indicating accumulation records. Grey stars show the locations of the EGRIP ice-core site, Summit, and South Dome, which are referenced in Figs. 5 and 11.

---

## Author Comment (AC2) · 6 Mar 2020

For the reply, please see Author Comment #1 (AC1).

———————————————

---

## Referee Comment (RC3) · Anonymous Referee #3 · 12 Mar 2020

Review of Badgeley et al. 2020 on Greenland paleo data assimilation

Badgeley et al. present temperature and precipitation fields for the last 20,000 years over Greenland generated using a paleo data-assimilation technique. This is an interesting and potentially very valuable new approach to investigating past climates. The paper is well written and clearly illustrated, and I am generally enthusiastic about the work.

While the methodology represents a big step forward, the paper is also a step backwards in other regards as it assumes a constant linear scaling of d18O to site temperature for all sites and periods based on the spatial d18O-T relationship. This assumption has been disproven in the last 2 decades through careful work in the ice core community (including some of the papers cited here). This assumption will dominate all the

spatial and temporal patterns in the temperature reconstructions, and deserves more careful consideration than it is given here. The authors suggest that this problem is alleviated by using the precipitation weighted temperature, but they do not demonstrate this. Below I recommend some comparisons that should be done before the paper is suitable for publication.

My main concern is the use of a single linear d18O-T scaling based on the spatial d18O-T pattern at all sites and locations. While water isotopes are a valuable proxy, its temperature interpretation has proved very difficult. Borehole thermometry and d15N gas thermometry are the most reliable methods to get absolute (calibrated) temperature changes, and both suggest a d18O slope that is around half of the spatial relationship (0.67 permil/K) used here (as the authors acknowledge).

I suspect this assumption will lead to underestimated temperature variability in the posterior. The authors should check this for the abrupt transitions at the three sites (GISP2, NEEM, NGRIP) where d15N-based temperature changes are known (Buizert et al. 2014).

However, there is also a clear spatial gradient, as first noted by [Guillevic et al., 2013], a paper that should be cited and discussed. Guillevic observes that d18O changes are largest towards the north (i.e. NEEM), and smaller towards the south (i.e. Summit). However, the actual temperature changes have the opposite gradient – smallest in the north and largest in the south. This means that the d18O-T relationship has an enormous spatial gradient, from ∼0.6 at NEEM to ∼0.3 at Summit. The Guillevic temperature gradient is seen in many (all?) climate model simulations and should thus be considered very robust.

These patterns are such that when using a single constant slope (as the authors do), the larger temperature changes would appear to be in the north, as is indeed the case in their reconstructions (Fig 4a, 4c). However, the Guillevic result would actually suggest the opposite pattern in temperature. The authors need to plot the magnitude

of abrupt climate warming in their reanalysis (either the 14.7 ka or 11.6 ka transition), and compare it to the d15N-based values. My hunch is that they will find the opposite pattern from the Guillevic result.

It has also been documented that the d18O-T slope is strongly variable in time, changing by almost a factor of 2 [Kindler et al., 2014].

It would be unreasonable to ask the authors to redo all the work abandoning a key assumption; rather I think they should do a careful comparison to d15N-based estimates of abrupt climate change to assess how well their method captures both the magnitude and spatial pattern of abrupt temperature changes – and the implications this may have for the LGM and Holocene optimum patterns shown in Fig 4a and 4c. Perhaps they can provide some suggestions for future work on ways to assimilate the d15N-based climate constraints directly.

If the reconstructed N-S temperature gradient during abrupt change is indeed opposite to the Guillevic gradient, this should be clearly stated in the abstract.

The authors suggest that using precipitation-weighted temperatures alleviates the problems associated with using a linear d18O-T scaling. To validate this claim, at the very least they should show a comparison of the 21ka histories of TraCE 2m temperature and TraCE precipitation-weighted temperature at a key site (e.g. Summit), to show how different these two really are. Ideally, they would show more clearly how this impacts the reconstructed magnitude of the abrupt climate change events (that are most strongly constrained by the d15N data).

General comments:

Please describe the data assimilation method in more general terms understandable to the non-initiated, so the reader won't have to track down the Hakim reference. Can we think of the posterior as a cleverly weighted sum of the randomly selected model timesteps put into the prior?

Is there some relationship between the posterior and the 21ka climate simulation – for example, is the posterior solution for the LGM very similar to the TraCE simulation of the LGM? Is the posterior LGM solution strongly weighted towards LGM model years randomly selelcted in the prior?

The TraCE simulation has quire a coarse grid I imagine? Please specify the exact resolution. I imagine it may even put multiple of the ice core sites in a single grid box. Perhaps the grid box resolution could be drawn onto figure 1? It seems that the spatial fields in Fig 4 are much smoother than the model would be. Did you apply smoothing or some other technique?

How meaningful is it to use global climate simulations and constrain them only in Greenland? From a global perspective, Greenland is essentially a single location and the global climate field is not at all constrained. How well-behaved is the far-field response in the reanalysis? And does this somehow impact the reconstruction? I think doing this with global proxy databases (such as [Shakun et al., 2012] would be a great next step (beyond the scope of this paper of course).

Seasonality is very briefly addressed, but it deserves more attention as it is an important climate parameter. Please specifically address seasonality in both the prior and posteriors. Will the reconstructions made available online have T and/or P seasonality in them, and if so, describe how this seasonality is derived. I imagine the seasonality of the posterior can be derived via the assimilation method?

The authors find an unusually late timing of the Holocene optimum around 5ka – much later than other ice-core based estimate from both d18O and melt layers. Looking at Fig 2, it appears that Camp Century (and perhaps Dye 3) are the only cores that suggest such timing, and since the temperature reanalysis is fully determined by ice core d18O, it follows that those two cores must be responsible for this timing (do you agree with this assessment?). However, as pointed out by [Vinther et al., 2009], these sites experience strong thinning in the first half of the Holocene, which will shift their

apparent climatic optimum towards a later age (as early Holocene climatic warmth is masked by a cooler site temperature at higher elevation). Could the late (5ka) timing of the climatic optimum in your reanalysis be an artifact of the thinning history of the Greenland ice sheet? Please discuss briefly in the text.

The data assimilation is fully dependent upon the accuracy of the TraCE-21 climate model simulation in capturing Greenland climate. Therefore, the paper needs a short evaluation of how well this model actually simulates Greenland T and P in the modern day. The TraCE T and P fields should be compared to modern-day Greenland reconstructions thereof; I would recommend the works by Box et al. on this topic [Box, 2013; Box and Colgan, 2013; Box et al., 2009; Box et al., 2013], but general reanalysis products such as NCEP or ERA5 are suitable also.

All the figures show relative temperature changes and accumulation changes (relative to the reference period, which is not defined as far as I can tell). But when forcing ice sheet models absolute values are needed. Are these absolute values taken from the last time-slice of the TraCE simulations, or is something better used?

Minor comments

L8: What are "independent ice core records"? d18O? Again, I think the reconstructions should be compared during the abrupt temperature transitions at NEEM NGRIP and GISP2, which is where d15N-N2 provides a very robust estimate of the magnitude of change. Those are the truly independent ice core records to compare to.

L24: This is somewhat misleading, because you'll always need to do such precip corrections unless you are doing a fully coupled ice-climate simulation. As the ice elevation in the ice sheet simulation evolves, it differs from the reference elevation at which the climate field is defined; this needs to be corrected for via clausius-clapeyron or similar. So also with your forcing the ice sheet models will need to apply thermodynamic precip corrections.

L28: Many more d15N studies to cite here: [Guillevic et al., 2013; Kindler et al., 2014; Severinghaus and Brook, 1999; Severinghaus et al., 1998]

L38: "restricted to a single climate model realization"; wouldn't this critique apply to your study as well? It appears that both use the exact same climate model run.

L70: "measured layer thickness" is not really true. For several cores you use volcanic ties, in which case the layer thicknesses are not measured but inferred

L136: "captures the...." This is in the eye of the beholder. With the exception of the Bolling warming itself the TraCE run matches the abrupt transitions poorly – there is no YD to speak of.

L145: why not use P-E? is evaporation negligible?

L217: "highly correlated" is a strong statement. Do you have a reference? Normally d18O and site temperature are not highly correlated at most sites on observational time scales (< 0.5).

L224: Based on the recent literature, I think that post-depositional alternation may be the largest complication in interpreting the d18O record. Please mention.

L241: Can you plot T_site and Tˆ*_site together for the last 21ka at a key site (e.g. Summit). That will let the reader judge the impact of using T* instead of T.

How is the seasonality of the posterior linked to the seasonality of the prior?

L261: "grid-cell closest to site" is this also done for T, or do you use 2D linear interpolation or similar? Are there cases where multiple cores share a closest grid cell?

L295: maybe a sentence on how this was estimated?

L313-314: But [Dahl-Jensen et al., 1998] estimates it a lot colder at GRIP, more like -22K cooling at the LGM (25ka). This should be mentioned.

L340: Maybe reference [Buchardt et al., 2012] who did very detailed analyses of this.

L416: are other d18O records really independent? They suffer the same biases from seasonality, source effects, etc. For true independence, compare to d15N-N2.

L438: TraCE has no HTM anywhere! (one of its many problems. . .)

L476: This is more of a discussion than a conclusion item. Consider moving it. Also, see my comment above, the 5ka timing could be an artifact of ice sheet elevation changes.

Figure 4: please add panels (e) and (f) with the T and P change over an abrupt transition (e.g. the Bolling onset). In panel (c), only show the cores that actually constrain the LGM (so not Agassiz, camp century and Renland). Why are the field so much smoother than the TraCE CCSM3 model resolution? Baffin bay has a large temp response with no cores to constrain it – can we trust this?

Fig 5: the "noise" in T (i.e. high frequency signals) at all core sites seem strongly correlated. How come? Could it be that the posterior is more or less reflecting the mean d18O of the various sites?

Fig 6: The largest features in the plot are not directly constrained by any cores. Do you trust these?

Figs 7 and 8 are very technical and could be moved to the supplement.

References:

Box, J. E. (2013), Greenland Ice Sheet Mass Balance Reconstruction. Part II: Surface Mass Balance (1840–2010)*, J. Clim., 26(18), 6974-6989, doi: 10.1175/jcli-d-12-00518.1.

Box, J. E., and W. Colgan (2013), Greenland ice sheet mass balance reconstruction. Part III: Marine ice loss and total mass balance (1840–2010), J. Clim., 26(18), 6990-7002.

Box, J. E., L. Yang, D. H. Bromwich, and L.-S. Bai (2009), Greenland Ice Sheet

Surface Air Temperature Variability: 1840–2007*, J. Clim., 22(14), 4029-4049, doi: 10.1175/2009jcli2816.1.

Box, J. E., N. Cressie, D. H. Bromwich, J.-H. Jung, M. van den Broeke, J. van Angelen, R. R. Forster, C. Miège, E. Mosley-Thompson, and B. Vinther (2013), Greenland ice sheet mass balance reconstruction. Part I: Net snow accumulation (1600–2009), J. Clim., 26(11), 3919-3934.

Buchardt, S. L., H. B. Clausen, B. M. Vinther, and D. Dahl-Jensen (2012), Investigating the past and recent δ18O-accumulation relationship seen in Greenland ice cores, Clim. Past, 8(6), 2053-2059, doi: 10.5194/cp-8-2053-2012.

Dahl-Jensen, D., K. Mosegaard, N. Gundestrup, G. D. Clow, S. J. Johnsen, A. W. Hansen, and N. Balling (1998), Past Temperatures Directly from the Greenland Ice Sheet, Science, 282(5387), 268-271, doi: 10.1126/science.282.5387.268.

Guillevic, M., et al. (2013), Spatial gradients of temperature, accumulation and delta18O-ice in Greenland over a series of Dansgaard-Oeschger events, Clim. Past, 9(3), 1029-1051, doi: 10.5194/cp-9-1029-2013.

Kindler, P., M. Guillevic, M. Baumgartner, J. Schwander, A. Landais, and M. Leuenberger (2014), Temperature reconstruction from 10 to 120 kyr b2k from the NGRIP ice core, Clim. Past, 10(2), 887-902, doi: 10.5194/cp-10-887-2014.

Severinghaus, J. P., and E. J. Brook (1999), Abrupt climate change at the end of the last glacial period inferred from trapped air in polar ice, Science, 286(5441), 930-934.

Severinghaus, J. P., T. Sowers, E. J. Brook, R. B. Alley, and M. L. Bender (1998), Timing of abrupt climate change at the end of the Younger Dryas interval from thermally fractionated gases in polar ice, Nature, 391(6663), 141-146.

Shakun, J. D., P. U. Clark, F. He, S. A. Marcott, A. C. Mix, Z. Liu, B. Otto-Bliesner, A. Schmittner, and E. Bard (2012), Global warming preceded by increasing carbon dioxide concentrations during the last deglaciation, Nature, 484(7392), 49-54, doi:

http://www.nature.com/nature/journal/v484/n7392/abs/nature10915.html#supplementary-information.

Vinther, B. M., et al. (2009), Holocene thinning of the Greenland ice sheet, Nature, 461(7262), 385-388, doi: 10.1038/nature08355.

---

## Author Comment (AC3) · 7 Apr 2020

**Reply to Referee Comment #3**

We thank the referee for their time and insights. In the following, we address the referee's comments, which are in black. Our replies are inline in blue. Note that figures and tables in this reply are labeled with "R" preceding the number.

5    Review of Badgeley et al. 2020 on Greenland paleo data assimilation

Badgeley et al. present temperature and precipitation fields for the last 20,000 years over Greenland generated using a paleo data-assimilation technique. This is an interesting and potentially very valuable new approach to investigating past climates. The paper is well written and clearly illustrated, and I am generally enthusiastic about the work.

Thank you.

While the methodology represents a big step forward, the paper is also a step backwards in other regards as it assumes a constant linear scaling of d18O to site temperature for all sites and periods based on the spatial d18O-T relationship. This
15    assumption has been disproven in the last 2 decades through careful work in the ice core community (including some of the papers cited here). This assumption will dominate all the spatial and temporal patterns in the temperature reconstructions, and deserves more careful consideration than it is given here. The authors suggest that this problem is alleviated by using the precipitation weighted temperature, but they do not demonstrate this. Below I recommend some comparisons that should be done before the paper is suitable for publication.

My main concern is the use of a single linear d18O-T scaling based on the spatial d18O-T pattern at all sites and locations. While water isotopes are a valuable proxy, its temperature interpretation has proved very difficult. Borehole thermometry and d15N gas thermometry are the most reliable methods to get absolute (calibrated) temperature changes, and both suggest a d18O slope that is around half of the spatial relationship (0.67 permil/K) used here (as the authors acknowledge).

25

We agree with the referee that it would be overly-simplistic to assume "a constant linear scaling of $\delta^{18}$O to site temperature for all sites and periods based on the spatial $\delta^{18}$O-T relationship". This is not, however, what we do; we allow the $\delta^{18}$O-temperature relationship to vary spatially by relying on precipitation-weighted temperature (T*) from TraCE-21ka. As we write in our paper (lines 229-232), "Numerous studies have suggested that precipitation seasonality is the largest source of
30    nonlinearity in the $\delta^{18}$O-T site relationship (e.g., Steig et al., 1994; Pausata and Löfverström, 2015); changes in precipitation seasonality are thought to be the primary reason that the effective $\delta^{18}$O-T site relationship for the glacial-interglacial transition has such a low slope (Werner et al., 2000)." We convert T* to $\delta^{18}$O using the equation $0.67T^* = \delta^{18}$O, and then compute the best-fit slope between $\delta^{18}$O and temperature to find their relationship. This effectively assumes a constant linear scaling between the *instantaneous* $\delta^{18}$O and site temperature, while allowing for changes in precipitation to affect the time-averaged
35    relationship. These TraCE-21ka-derived slopes vary between 0.42 and 0.66 ‰ °C$^{-1}$ at the core sites (Table R1), and are less than the modern spatial relationship of 0.67 ‰ °C$^{-1}$ at most locations around Greenland (e.g., Fig. R1).

The TraCE-21ka-derived slopes vary both in space and across prior ensembles (Fig. R1 and Table R1). By using ten different prior ensembles, we capture the uncertainty in the $\delta^{18}$O-temperature relationship, as determined by TraCE-21ka. We also ex-
40    amine a wider range of slope estimates in our sensitivity experiments. The slopes for S4 are shown in Table R1, and the slopes for S1, S2, and S3 are 0.67, 0.5, and 0.335 ‰ °C$^{-1}$, respectively. As described in our paper (lines 389 to 405), the magnitude of the slope affects the magnitude of the anomalies (Fig. 10 in the paper). The spatial pattern of the slope also has an effect. For example, in the early Holocene, the spatially-variable slopes result in a reconstruction that warms more slowly. In addition, the reconstructions with spatially-varying slopes show stronger north-south gradients than those with spatially-constant slopes.
45    This north-south gradient shows up especially in the abrupt transitions, with larger changes in the north relative to the south (Table R2). In the paper, we had not discussed the impact of the spatially-varying slopes on the spatial pattern of the reconstructions; we will include this discussion in the revised paper by making the following revisions to Sect. 4.2, lines 400 to 405

New text is in *italics*.

*The temperature results are also* sensitive to the spatial pattern of the $\delta^{18}$O-T relationship. *We find this by comparing the results from the S1-S3 scenarios that assume a spatially-uniform relationship* to results from the main reanalysis and S4 scenario that assume a spatially-variable *relationship*. The S1-S3 scenarios have a characteristic shape to their time series (Fig. 10), and, although the main reanalysis and S4 scenario generally fit this characteristic shape in the glacial and middle-late Holocene, in the early Holocene the main reanalysis and S4 diverge and show slower warming trends than the S1-S3 scenarios. *In addition, the reconstructions with spatially-varying $\delta^{18}$O-T relationships show stronger north-south gradients during times of abrupt temperature change than those with spatially-constant relationships (e.g., Table R2). These findings indicate* that there is new information added by using a PSM that accounts for spatial variability in precipitation seasonality.

The reviewer is correct that we effectively assume that the $\delta^{18}$O-temperature relationship is constant in time. We could in principle account for temporal changes in the $\delta^{18}$O-temperature relationship by using a time-varying prior; however, this method is complicated by discontinuities and/or extra assimilation parameters. We use the same prior ensemble for all time steps of the reconstruction, which avoids discontinuities in the reconstruction and does not require us to constrain extra assimilation parameters with so few proxy records. A consequence of this method is that the $\delta^{18}$O-temperature relationship, which is derived from the prior ensemble, is constant in time. Unfortunately, we are restricted in our ability to both use a time-varying prior and avoid the complications stated above until more long, fully-coupled climate simulations become available (see our reply to Reviewer #1, lines 22-105, for more details).

As the referee mentioned, borehole thermometry and $\delta^{15}$N gas thermometry are reliable methods to getting at the $\delta^{18}$O-temperature relationship; however, as we say on lines 233-237 in the paper, we do not rely on these methods because borehole thermometry and $\delta^{15}$N gas thermometry are not available at all sites. Instead, we rely on the TraCE-21ka-derived relationships described above, which we compare to relationships found previously using borehole and $\delta^{15}$N gas thermometry (Table R1). It is known that the $\delta^{18}$O-temperature relationship varies temporally depending on the length of time considered and the date (Jouzel et al., 1997). Table R1 shows that indeed, different investigations have estimated a variety of slopes for a variety of time periods. Differences in the estimated slopes are likely to result from the method used in a particular investigation and the time period considered. Some estimates, such as Guillevic et al. (2013) and Buizert et al. (2014) are for abrupt transitions, such as Dansgaard-Oeschger events, while others find mean slopes over longer periods of time, such as Kindler et al. (2014) and our own paper. The slopes that we derive from TraCE-21ka mostly fall within the ranges found by previous studies, even though our slopes are estimated for a time period that is not addressed in the other investigations, the last 20,000 years.

We will include this discussion, along with the associated figures and tables, in the revised paper and supplementary information. One of these revisions will be edits to lines 242 to 244 in Sect. 2.3.1 of the paper. New text is in *italics*.

With T$^*_{site}$ in our PSM, we find that the $\delta^{18}$O-T$_{site}$ slope is spatially variable *(e.g., Fig. R1), ranging from 0.42 and 0.66 ‰ °C$^{-1}$ at the ice-core sites (Table R1), and tending to be less than the modern spatial relationship of 0.67 ‰ °C$^{-1}$ at most locations around Greenland. These slopes vary both in space and across prior ensembles. By using ten different prior ensembles, we capture the uncertainty in the $\delta^{18}$O-temperature relationship resulting from variations in precipitation seasonality. These TraCE-21ka-derived estimates lie within the range of slopes estimated for sites around Greenland for a variety of time periods (Table R1). Differences seen in Table R1 reflect both the different methods used and the time period considered. Some estimates, such as Guillevic et al. (2013) and Buizert et al. (2014) are for abrupt transitions, such as Dansgaard-Oeschger events, while others find mean slopes over longer periods of time, such as Kindler et al. (2014) and this investigation.*

I suspect this assumption will lead to underestimated temperature variability in the posterior. The authors should check this for the abrupt transitions at the three sites (GISP2, NEEM, NGRIP) where d15N-based temperature changes are known (Buizert et al. 2014).

We thank the referee for this suggestion. In Table R3 we compare the magnitudes of three abrupt transitions (warming into the Bølling-Allerød, cooling into the Younger Dryas, and warming into the Holocene) against the $\delta^{15}$N-derived temperature estimates from Buizert et al. (2014) at three locations. We note that these three sites are all in central and northern Greenland and are not necessarily representative of southern Greenland or the coastal ice caps (e.g., Agassiz and Renland). The comparison shows that our approach does not underestimate temperature variability as compared to Buizert et al. (2014); for the NEEM and NGRIP sites, the mean values of each study are within one standard deviation, while for the GISP2 site, the one standard deviation uncertainty bounds overlap for two of the three transitions. The specifics of the comparison are dependent on the location and the time period. For example, our reconstruction shows greater variability than the $\delta^{15}$N-derived reconstruction at NEEM, but less variability at NGRIP and GISP2. In addition, the reconstructions agree best during the Younger Dryas at the NGRIP and GISP2 sites, but are in better agreement during the Holocene and Bølling-Allerød transitions at NEEM.

However, there is also a clear spatial gradient, as first noted by [Guillevic et al., 2013], a paper that should be cited and discussed. Guillevic observes that d18O changes are largest towards the north (i.e. NEEM), and smaller towards the south (i.e. Summit). However, the actual temperature changes have the opposite gradient – smallest in the north and largest in the south. This means that the d18O-T relationship has an enormous spatial gradient, from ~0.6 at NEEM to ~0.3 at Summit. The Guillevic temperature gradient is seen in many (all?) climate model simulations and should thus be considered very robust.

Thank you for the suggestion. We agree that the Guillevic et al. (2013) paper should be cited. We are aware of the discrepancy between the spatial gradient in the $\delta^{18}$O and in the $\delta^{15}$N-derived temperature; however, we do not agree that this discrepancy has been resolved. The Guillevic temperature gradient (larger temperature changes in the south than in the north) is based on four central to northern ice core records (GISP2 and GRIP, however, are at essentially the same site near Summit). Guillevic et al. (2013) find that the temperature changes at NGRIP and the Summit cores are statistically indistinguishable; thus most of their spatial pattern is driven by the difference between NEEM and NGRIP/GISP2/GRIP. We cannot know without more constraints whether this north-south pattern that appears between NEEM and these three other cores holds for other parts of Greenland, such as southern Greenland. As we show in Sect. S3 of our supplementary information, a southern data point is key to reconstructing southern Greenland climate.

The referee notes the reproduction of Guillevic et al.'s north-south pattern by climate models as evidence for the interpretation that this pattern extends to southern Greenland. We acknowledge that this has been found in some climate simulations, for example, Buizert et al. (2014) show that the TraCE-21ka simulation has this pattern during the Bølling-Allerød transition. Our prior ensembles are selected randomly from all time-steps in TraCE-21ka, such that the covariance patterns reflect the dominant patterns of climate change in TraCE-21ka. These dominant patterns primarily show that the highest-magnitude temperature changes are in nortnern Greenland and, as a weaker signal, that there are higher-magnitude changes in southern than in central Greenland (e.g., Fig. R2). Thus, the TraCE-21ka model suggests that the pattern during the Bølling-Allerød transition is not the dominant spatial pattern of Greenland climate for the last 20,000 years. When combined with the covariance pattern of our proxy records, the result is our reconstruction, which shows larger changes to the north. We acknowledge that the spatial patterns of our results may be different with a time-varying prior ensemble. We will add this discussion to our revised paper (see lines 188 to 217 in this document).

These patterns are such that when using a single constant slope (as the authors do), the larger temperature changes would appear to be in the north, as is indeed the case in their reconstructions (Fig 4a, 4c). However, the Guillevic result would actually suggest the opposite pattern in temperature. The authors need to plot the magnitude of abrupt climate warming in their reanalysis (either the 14.7 ka or 11.6 ka transition), and compare it to the d15N-based values. My hunch is that they will find the opposite pattern from the Guillevic result.

As noted above, the reviewer is not correct that we use a single slope for all ice-core sites. We use a spatially-varying slope (for an example, see Fig. R1). We will be sure to clarify this in the revised paper.

Figs. R3, R4, and R5 show the spatial pattern of three abrupt transitions in our main reconstruction: warming into the Bølling-Allerød, cooling into the Younger Dryas, and warming into the Holocene, respectively. These same results are shown in Table R3 for the eight ice core locations and are compared to results from Buizert et al. (2014). Our results show increasing variability to the north accross seven of the eight cores, while the Guillevic et al. (2013) and Buizert et al. (2014) results show increasing variability to the south across just three cores. So yes, the referee is correct that our results show the opposite pattern from the Guillevic et al. (2013) results; however, the reason for this difference is not due to a spatially-constant $\delta^{18}$O-tempreature relationship. Instead, our spatial pattern is a result of how the spatial patterns in the proxy records are spread throughout the domain via the spatial covariance pattern between temperature and precipitation-weighted temperature in the prior ensemble.

Despite the opposite gradient in north-south variability during abrupt transitions, there are some similarities between our reconstruction and the Buizert et al. (2014) reconstructions. Using the same time definitions, we computed the difference between the Younger Dryas and Older Dryas temperatures (Fig. R6). We find a similar north-south pattern, with a greater difference in the north than in central and southern Greenland. We find even better correspondance between the Buizert et al. (2014) results and our S3 sensitivity experiment (Fig. R7), which has a spatially-constant $\delta^{18}$O-temperature relationship. We will add a brief discussion about these comparisons to the revised version of the manuscript.

It has also been documented that the d18O-T slope is strongly variable in time, changing by almost a factor of 2 [Kindler et al., 2014].

Please refer to lines 60 to 67 of this document for an explanation of why we use a method that has a constant $\delta^{18}$O-T relationship in time in the prior ensemble. We note, however, at the $\delta^{18}$O-T relationship is free to vary in time in the posterior ensemble.

It would be unreasonable to ask the authors to redo all the work abandoning a key assumption; rather I think they should do a careful comparison to d15N-based estimates of abrupt climate change to assess how well their method captures both the magnitude and spatial pattern of abrupt temperature changes – and the implications this may have for the LGM and Holocene optimum patterns shown in Fig 4a and 4c. Perhaps they can provide some suggestions for future work on ways to assimilate the d15N-based climate constraints directly.

We agree with the referee that it would be ideal to assimilate more than just $\delta^{18}$O records. We thank the referee for their suggestion of including this in a discussion of future work, and we will do so in the revised paper.

In the preceding replies, we have done a careful comparison between our reconstructions and $\delta^{15}$N-based reconstructions. We will include these comparisons in the revised paper. Our comparisons show that our main reconstruction is within error of the $\delta^{15}$N-derived estimates for the three abrupt temperature transitions that occured in the last 20,000 years (Table R3). At the three locations where we can compare to $\delta^{15}$N-derived estimates – NEEM, NGRIP, and Summit – we find some discrepency in the spatial pattern of these abrupt transitions; however, because there are so few $\delta^{15}$N-derived estimates, we cannot say whether this disparity extends to southern Greenland or the coastal ice caps.

We will add the following revisions to lines 306 to 316 in Sect. 3 of the paper. New text is in *italics*. We are not certain yet where we will put the figures mentioned in this revised text; we may merge them into one figure to include in the paper or put them into the Supplementary Information.

Through the assimilation of ice-core data with a prior ensemble that is constant in time, we produce a spatially-complete Greenland temperature and precipitation reanalysis (Figs. 4 and 5). Here we focus on results relevant to the evolution and sensitivity of the Greenland Ice Sheet, including the late glacial anomaly, *the three periods of abrupt temperature*

*change,* and the Holocene thermal maximum (HTM), and the relationship between temperature and precipitation.

In our reanalysis, late glacial (20-15 ka) mean-temperature anomalies range from about -20 °C in northern Greenland to less than -10 °C in southern Greenland (Fig. 4c). At the GRIP and GISP2 ice-core sites, the reanalysis has a -14 °C anomaly with a standard deviation of 2 °C. This is in excellent agreement with the mean-temperature anomaly of -14 °C for the same period at the GISP2 site, which was derived from $\delta^{18}O$ calibrated with borehole thermometry (Cuffey et al., 1995; Cuffey and Clow, 1997). Average late-glacial precipitation in the reanalysis ranges from a third to half of modern with the highest values on the coasts around southern Greenland (Fig. 4d).

*Our reanalysis covers three periods of abrupt temperature change: the Bølling-Allerød waming, cooling into the Younger Dryas, and warming into the Holocene. Figs. R3, R4, and R5, respectively, show the spatial patterns of these abrupt changes for our main reanalysis. Our results consistently show the largest-magnitude temperature changes in northern Greenland for each of these events. This finding is in contrast with results from $\delta^{15}N$-derived temperature re-constructions that show larger changes in central Greenland cores than in north-central Greenland cores (e.g., Guillevic et al., 2013; Buizert et al., 2014). It has been argued that this $\delta^{15}N$-derived pattern may be extended to southern Green-land because some climate simulations replicate this pattern (e.g., Buizert et al., 2014). For example, TraCE-21ka shows the largest temperature changes in southern Greenland during the Bølling-Allerød transition (Liu et al., 2009).*

*Our prior ensembles are selected randomly from all time steps in TraCE-21ka, such that the covariance patterns re-flect the dominant patterns of climate change in TraCE-21ka. These dominant patterns primarily show that the highest-magnitude temperature changes are in nortnern Greenland and, as a weaker signal, that there are higher-magnitude changes in southern than in central Greenland. Thus, the TraCE-21ka model suggests that the pattern during the Bølling-Allerød transition is not the dominant spatial pattern of Greenland climate for the last 20,000 years. When combined with the covariance pattern of our proxy records, the result is our reconstruction, which shows larger changes to the north. We acknowledge that the spatial patterns of our results may be different with a time-varying prior ensemble. Esti-mates of the abrupt temperature events in our current reanalysis, however, are within error of the estimates from Buizert et al. (2014) for most of the three events at each of the three sites, GISP2, NGRIP, and NEEM (Table R3).*

If the reconstructed N-S temperature gradient during abrupt change is indeed opposite to the Guillevic gradient, this should be clearly stated in the abstract.

We will edit the relevant part of our abstract to say the following. Note that new text is in *italics* and that some of the edits are in response to later comments (see lines 384 to 386).

Reconstructions of past temperature and precipitation are fundamental to modeling the Greenland Ice Sheet and assess-ing its sensitivity to climate. Paleoclimate information is sourced from proxy records and climate-model simulations; however, the former are spatially incomplete while the latter are sensitive to model dynamics and boundary condi-tions. Efforts to combine these sources of information to reconstruct spatial patterns of Greenland climate over glacial-interglacial cycles have been limited by assumptions of fixed spatial patterns and a restricted use of proxy data. We avoid these limitations by using paleoclimate data assimilation to create independent reconstructions of temperature and precipitation for the last 20,000 years. Our method uses *oxygen-isotope ratios of ice and accumulation rates* from long ice-core records and extends *this information* to all locations across Greenland using spatial relationships derived from a transient climate-model simulation. *Standard evaluation metrics for this method show that our results capture climate at locations without ice-core records. A comparison to other paleoclimate proxy records shows that our results generally agree with previous findings. There are some differences, however, especially relating to the spatial pattern of abrupt cli-mate transitions, for which our results show greater temperature changes in the north, while temperature reconstructions from $\delta^{15}N$ of $N_2$ show the opposite. We additionally investigate the relationship between precipitation and temperature, finding that it* is frequency dependent and spatially variable, suggesting that thermodynamic scaling methods commonly

used in ice-sheet modeling are overly simplistic. Our results demonstrate that paleoclimate data assimilation is a useful tool for reconstructing the spatial and temporal patterns of past climate on timescales relevant to ice sheets.

The authors suggest that using precipitation-weighted temperatures alleviates the problems associated with using a linear d18O-T scaling. To validate this claim, at the very least they should show a comparison of the 21ka histories of TraCE 2m temperature and TraCE precipitation-weighted temperature at a key site (e.g. Summit), to show how different these two really are. Ideally, they would show more clearly how this impacts the reconstructed magnitude of the abrupt climate change events (that are most strongly constrained by the d15N data).

We have provided a figure as the referee suggests (Fig. R8), which shows that there is a difference between the temperature and precipitation-weighted temperature. The figure is only for the grid cell closest to Summit, but this is the case for all locations around Greenland, though the magnitude of the difference varies by location. We will include this figure in the revised paper or Supplementary Information.

Through our temperature sensitivity experiments, we have tested the impact that precipitation-weighted temperature has on our reconstructions (see lines 389-405 in our paper for a discussion of the results of our sensitivity experiments). We thank the reviewer for their suggestion that we discuss this in more detail. Previously in this reply (lines 39 to 58 of this document), we discussed the impact that a spatially-varying $\delta^{18}$O-temperature relationship has on our results, and provided a revised section of the paper. This is relevant because the spatial variations in the $\delta^{18}$O-temperature relationship are a result of using precipitation-weighted temperature.

General comments:

Please describe the data assimilation method in more general terms understandable to the non-initiated, so the reader won't have to track down the Hakim reference. Can we think of the posterior as a cleverly weighted sum of the randomly selected model timesteps put into the prior? Is there some relationship between the posterior and the 21ka climate simulation – for example, is the posterior solution for the LGM very similar to the TraCE simulation of the LGM? Is the posterior LGM solution strongly weighted towards LGM model years randomly selelcted in the prior?

The referee is correct that this data assimilation method can be thought of as "a cleverly weighted sum of the randomly selected model timesteps put into the prior" because the ensemble mean is indeed a weighted sum of the ensemble members *if* there is no covariance localization, as is the case in our paper. The referee is also correct that it can be determined whether the posterior mean for a time step in the glacial is more strongly weighted towards ensemble members that were selected from the glacial period in TraCE-21ka; however, these weights are not a direct output of our method.

Thank you for the suggestion to describe data assimilation in more general terms. We agree that it's a good idea to provide a summary of the method. We will make the following edit to the first paragraph in Sect. 2.3, which describes the data assimilation method. The edit is in *italics*.

To combine the ice-core data and climate-model data, we use an offline data assimilation method similar to that described in Hakim et al. (2016). *This method can be summed up as a linear combination of randomly-selected model states that are weighted according to new information provided by the proxy records (if no covariance localization is used).*

The TraCE simulation has quite a coarse grid I imagine? Please specify the exact resolution. I imagine it may even put multiple of the ice core sites in a single grid box. Perhaps the grid box resolution could be drawn onto figure 1? It seems that the spatial fields in Fig 4 are much smoother than the model would be. Did you apply smoothing or some other technique?

Yes, we agree that we should specify the exact resolution of TraCE-21ka. As we replied to Referee #2, we will add this to Sect. 2.2 of the paper (lines 132 to 147). The spatial resolution is T-31, or about 3.75 degrees, and the temporal resolution is monthly, but we average it to 50-year resolution (resulting in about 440 time steps).

We thank the reviewer for calling our attention to the fact that we forgot to say in the figure captions that the spatial fields are smoothed. We agree that it could be helpful to see the original resolution, so we will convert our plots back to the original resolution.

How meaningful is it to use global climate simulations and constrain them only in Greenland? From a global perspective, Greenland is essentially a single location and the global climate field is not at all constrained. How well-behaved is the far-field response in the reanalysis? And does this somehow impact the reconstruction? I think doing this with global proxy databases (such as [Shakun et al., 2012]) would be a great next step (beyond the scope of this paper of course).

Our paper is on reconstructing Greenland climate using proxy records from Greenland. We agree that a next step in applying paleoclimate data assimilation to galcial-interglacial timescales is to use a global proxy database to reconstruct global climate variables. We will menton this in a paragraph about future work in the conclusion of the revised paper.

Seasonality is very briefly addressed, but it deserves more attention as it is an important climate parameter. Please specifically address seasonality in both the prior and posteriors. Will the reconstructions made available online have T and/or P seasonality in them, and if so, describe how this seasonality is derived. I imagine the seasonality of the posterior can be derived via the assimilation method?

We agree that seasonality deserves more attention. As we note in our reply to Referee #2, we plan to include a description of TraCE-21ka's seasonality in Sect. 2.2 of the paper (lines 132 to 147). Seasonality is only used to compute $T^*$; it is in no other way included in the data assimilation process or results. Our results are mean-annual 50-year averages, which we will be sure to state early on in the paper. The referee is correct that climate variables for specific seasons or the seasonal cycle itself can theoretically be reconstructed using this data assimilation method.

The authors find an unusually late timing of the Holocene optimum around 5ka – much later than other ice-core based estimate from both d18O and melt layers. Looking at Fig 2, it appears that Camp Century (and perhaps Dye 3) are the only cores that suggest such timing, and since the temperature reanalysis is fully determined by ice core d18O, it follows that those two cores must be responsible for this timing (do you agree with this assessment?). However, as pointed out by [Vinther et al., 2009], these sites experience strong thinning in the first half of the Holocene, which will shift their apparent climatic optimum towards a later age (as early Holocene climatic warmth is masked by a cooler site temperature at higher elevation). Could the late (5ka) timing of the climatic optimum in your reanalysis be an artifact of the thinning history of the Greenland ice sheet? Please discuss briefly in the text.

We agree with the referee that the timing of the Holocene thermal maximum (HTM) in our reconstructions tends to be on the later end of the range of previous findings (see lines 317 to 325 of the paper). We also agree that this signal appears to result from the Camp Century, Dye3, and perhaps the NEEM $\delta^{18}O$ records. The referee mentions how thinning in the early Holocene (Vinther et al., 2009) would affect our results. As we state in lines 127 and 460 of the paper, our reconstructions are for climate at the ice-sheet surface, meaning that they include the lapse-rate effect of changing surface elevation. This is opposed to climate reconstructions at a reference elevation. Thus, our reconstruction of a later HTM is consistent with previous findings of an earlier HTM (given a fixed reference elevation) (e.g., McFarlin et al., 2018) and early Holocene thinning (Vinther et al., 2009). This is something we have thought about in great detail, but decided not to include in the paper; however, as suggested by the referee, we will make the following edits to lines 455 to 465 in Sect. 5 of the paper. New text is in *italics*.

An important distinction among various different paleoclimate reconstructions for Greenland is in the treatment of elevation changes. Any paleoclimate reconstruction from ice-core records is complicated by ice-sheet elevation changes.

In (Vinther et al., 2009), it is assumed that the climate history is the same at all locations around Greenland, and that any differences among the ice core paleotemperature records is a result of that elevation change. In B18, past elevation changes are assumed to be negligible. In our reconstruction, the impact of elevation change on the spatial covariances of temperature and precipitation is implicitly accounted for as part of the data assimilation methodology. Formally, our reconstruction is of surface climate, not climate at a fixed elevation. *Consequently, our reanalysis may not be directly comparable to other paleoclimate reconstructions. For example, the HTM is commonly reconstructed as an early Holocene event in records that are at a fixed or nearly-fixed elevation. In our reanalysis, the HTM occurs later, in the mid-Holocene. This is likely a result of thinning in the early Holocene (Vinther et al., 2009), which is captured in the ice-core records and acts to dampen early-Holocene warming signals in our reanalysis.* Our method depends on the accuracy of the climate-elevation relationships in our prior – i.e. in the TraCE-21ka climate model simulation, which probably does not capture such relationships with particularly high fidelity since the model resolution is low and the climate and ice-sheet models are not coupled. Future work could take advantage of the probabilistic relationships among accumulation, temperature, and surface elevation as simulated in fine-scale regional climate models (Edwards et al., 2014).

The data assimilation is fully dependent upon the accuracy of the TraCE-21 climate model simulation in capturing Greenland climate. Therefore, the paper needs a short evaluation of how well this model actually simulates Greenland T and P in the modern day. The TraCE T and P fields should be compared to modern-day Greenland reconstructions thereof; I would recommend the works by Box et al. on this topic [Box, 2013; Box and Colgan, 2013; Box et al., 2009; Box et al., 2013], but general reanalysis products such as NCEP or ERA5 are suitable also.

The statement that "data assimilation is fully dependent upon the accuracy of the TraCE-21ka climate model simulation in capturing Greenland climate" is incorrect. Our results are dependent only on the spatial covariance patterns of the temperature anomalies and fractional precipitation in TraCE-21ka (referenced to 1850-2000 CE) and the covariance pattern of the proxy records. For our use of TraCE-21ka, what matters is how well TraCE-21ka captures the spatial pattern and variance of past climate anomalies, which is unknown except by comparison with the ice-core data (which our method does implicitly). The fidelity of the modern-day climatology is not particularly relevant.

All the figures show relative temperature changes and accumulation changes (relative to the reference period, which is not defined as far as I can tell). But when forcing ice sheet models absolute values are needed. Are these absolute values taken from the last time-slice of the TraCE simulations, or is something better used?

Before we use TraCE-21ka in our assimilation, we subtract (for temperature) or divide (for precipitation) by the mean of TraCE-21ka over the period 1850-2000 CE. Before we assimilate each proxy record, we subtract (for $\delta^{18}$O) or divide (for accumulation rate) by the mean of that record over the same period, 1850-2000 CE. Thus, the reconstructions are of temperature anomalies and fractional precipitation, which are referenced to the 1850-2000 CE mean climate as recorded by the ice-core records. We thank the referee for pointing out that this was not entirely clear. We will state this more clearly and earlier on in the revised paper.

Now that our reconstructions are complete, they may be turned into absolute values for applications such as ice-sheet modeling. This can be done by adding a 1850-2000 CE temperature climatology to the temperature fields and multiplying the precipitation fields by a 1850-2000 CE precipitation climatology (e.g., from Box, 2013). We do not do this in our paper.

Minor comments

L8: What are "independent ice core records"? d18O? Again, I think the reconstructions should be compared during the abrupt temperature transitions at NEEM NGRIP and GISP2, which is where d15N-N2 provides a very robust estimate of the magnitude of change. Those are the truly independent ice core records to compare to.

Our use of the term "independent ice-core records" refers to the comparison of our results against $\delta^{18}$O records that are excluded from that reconstruction iteration. This is explained later in the text around line 164, so we will edit the abstract to clarify what is meant. See lines 225 to 240 of this document for these edits.

In our results and discussion we compare to findings from previous work; however, the referee is correct that we do not directly compare to temperature reconstructions derived from $\delta^{15}$N of $N_2$. We have now done this comparison, discussed it previously in this reply, and provided a revision to include this discussion in Sect. 3 of the paper (see lines 188 to 217 of this document for the revisions).

L24: This is somewhat misleading, because you'll always need to do such precip corrections unless you are doing a fully coupled ice-climate simulation. As the ice elevation in the ice sheet simulation evolves, it differs from the reference elevation at which the climate field is defined; this needs to be corrected for via clausius-clapeyron or similar. So also with your forcing the ice sheet models will need to apply thermodynamic precip corrections.

We agree that ice-sheet models need to correct precipitation fields for changing surface elevation using some assumption about the relationship between precipitation and elevation. Our point, however, is about ice-sheet simulations that base their precipitation histories entirely on their temperature histories using a thermodynamic relation. Thus, the time series and spatial pattern the precipitation anomalies perfectly match those from the temperature fields, which we know to be false from ice-core records. We will clarify this distinction in the revised paper.

L28: Many more d15N studies to cite here: [Guillevic et al., 2013; Kindler et al., 2014; Severinghaus and Brook, 1999; Severinghaus et al., 1998]

Thank you. We appreciate the recommended citations and will include them in the revised paper.

L38: "restricted to a single climate model realization"; wouldn't this critique apply to your study as well? It appears that both use the exact same climate model run.

Yes, the referee is correct, and we will remove the quoted language from the revised paper. Our current results rely on a single climate simulation; however, our method is easily generalizable to any climate simulation of the past 20,000 years. The method in Buizert et al. (2018) is also generalizable to other climate simulations as long as these simulations are accompanied by single-fociring experients.

L70: "measured layer thickness" is not really true. For several cores you use volcanic ties, in which case the layer thicknesses are not measured but inferred

We will change the phrase to "the layer thickness".

L136: "captures the. . ..." This is in the eye of the beholder. With the exception of the Bolling warming itself the TraCE run matches the abrupt transitions poorly – there is no YD to speak of.

We will reword this sentence as follows: "By design, TraCE-21ka captures the major glacial-to-modern temperature change, as well as some of the short-term, rapid climate changes, such as the Bølling-Allerød transition (Liu et al., 2009)."

L145: why not use P-E? is evaporation negligible?

As we write later in the paper (lines 257 to 262), "accumulation is closely related to total precipitation at our ice-core sites". We demonstrate this with high-resolution model results (Langen et al., 2015, 2017) that show at most of the ice-core sites in Greenland, precipitation and accumulation are nearly equivalent, suggesting that evaporation is negligible. Dye3 is the only

site where there is a difference between accumulation and precipitation, and this is due primarily to melt, not evaporation. This is why we extract P from TraCE-21ka (as mentioned on line 145 of the paper), and use it, rather than P-E, for our proxy system model (described in Sect. 2.3.2 of the paper, lines 256-269).

L217: "highly correlated" is a strong statement. Do you have a reference? Normally d18O and site temperature are not highly correlated at most sites on observational time scales (< 0.5).

The referee is correct that the correlations are low on interannual timescales, but it is well established to be high on longer timescales (Jouzel et al., 1997).

L224: Based on the recent literature, I think that post-depositional alternation may be the largest complication in interpreting the d18O record. Please mention.

We disagree. As we note in the paper, diffusion in the firn column is irrelevant for our timescales of 50 years (Cuffey and Steig, 1998). The reviewer may be thinking of post-depositional processes involving water exchange between the snowpack and the atmosphere (e.g., Steen-Larsen et al., 2011), but it is not established that this has any significant effect on long-term relationships. Indeed, if anything, such processes improve the relationship between temperature and $\delta^{18}O$ as they tend to reduce the bias caused by the fact that snow does not accumulated continuously, but as discrete events.

L241: Can you plot $T_{site}$ and $T^*_{site}$ together for the last 21ka at a key site (e.g. Summit). That will let the reader judge the impact of using $T^*$ instead of T.

Thank you for the suggestion. We have done this in Fig. R8, which we will include in a section of the supplementary information.

How is the seasonality of the posterior linked to the seasonality of the prior?

As we stated previously, we only use seasonality to compute $T^*$; it is in no other way included in the data assimilation process or results. Both our prior ensemble and our reconstructions are made up of mean-annual 50-year averages. We will clarify this in the revised paper.

L261: "grid-cell closest to site" is this also done for T, or do you use 2D linear interpolation or similar? Are there cases where multiple cores share a closest grid cell?

Yes, this is also done for selecting which $T^*$ value to use in the $\delta^{18}O$ PSM. We will clarify that in the revised paper. At the resolution of TraCE-21ka, only the GISP2 and GRIP ice cores have the same closest grid cell.

L295: maybe a sentence on how this was estimated?

Thank you for bringing our attention to this point of confusion. We will clarify this in the revised paper. We use the same method as was explained for $\delta^{18}O$.

L313-314: But [Dahl-Jensen et al., 1998] estimates it a lot colder at GRIP, more like -22K cooling at the LGM (25ka). This should be mentioned.

Dahl-Jensen et al. (1998) found a colder temperature at GRIP for the LGM, but the time period we discuss is 20-15ka, which is five to ten thousand years later, when the Dahl-Jensen et al. (1998) estimate shows that it has significantly warmed.

L340: Maybe reference [Buchardt et al., 2012] who did very detailed analyses of this.

Thank you. We are aware of this study, and will refer to it in our discussion of the temperature-precipitation relationship on shorter timescales.

L416: are other d18O records really independent? They suffer the same biases from seasonality, source effects, etc. For true independence, compare to d15N-N2.

We agree with the referee that it is important to compare our results to other types of proxy records. In this reply we have compared our temperature reconstructions to those from Buizert et al. (2014) and provided revisions that we will make to the paper.

L438: TraCE has no HTM anywhere! (one of its many problems. . .)

We will change the wording to say, "TraCE-21ka has no obvious HTM in this location or any location around Greenland."

L476: This is more of a discussion than a conclusion item. Consider moving it. Also, see my comment above, the 5ka timing could be an artifact of ice sheet elevation changes.

We will move this paragraph to the discussion, and we will include a discussion of elevation effects (see lines 323 to 348 of this document).

Figure 4: please add panels (e) and (f) with the T and P change over an abrupt transition (e.g. the Bolling onset). In panel (c), only show the cores that actually constrain the LGM (so not Agassiz, camp century and Renland). Why are the field so much smoother than the TraCE CCSM3 model resolution? Baffin bay has a large temp response with no cores to constrain it – can we trust this?

Thank you for these suggestions. We will add spatial plots of at least one of the abrupt transitions. We will also be sure to only include the ice core locations that contribute to the reconstruction of each time period. As we agreed previously in this reply, we will restore the plots to their original resolution because we agree with the reviewer that this is helpful. We had originally smoothed the plots due to rendering issues, but we have fixed the previous problem.

The goal in using a method like data assimilation is to spread the information from point proxy locations to locations without proxy records. This allows us to reconstruct spatially-complete climate fields. With few proxies and many locations, the problem is underconstrained; however, we have shown that the method is skillful for some locations where proxies are not assimilated (see Sect. 4.1, lines 364 to 382 in the paper).

Fig 5: the "noise" in T (i.e. high frequency signals) at all core sites seem strongly correlated. How come? Could it be that the posterior is more or less reflecting the mean d18O of the various sites?

Each core has some influence on the reconstruction at every location. Thus, the time series at each location is a weighted sum of the time series of each core. This would indeed make the higher-frequency noise correlated at different locations. We will say this in the revised paper.

Fig 6: The largest features in the plot are not directly constrained by any cores. Do you trust these?

For our reply, please see lines 513 to 516 in this document.

Figs 7 and 8 are very technical and could be moved to the supplement.

530

These figures demonstrate how well our reconstruction performs at locations without any assimilated information. We think this is an important aspect of our paper, and we wish to keep these figures in the main text.

**References**

535    Box, J. E.: Greenland ice sheet mass balance reconstruction. Part II: Surface mass balance (1840–2010), Journal of Climate, 26, 6974–6989, 2013.

Buizert, C., Gkinis, V., Severinghaus, J. P., He, F., Lecavalier, B. S., Kindler, P., Leuenberger, M., Carlson, A. E., Vinther, B., Masson-Delmotte, V., et al.: Greenland temperature response to climate forcing during the last deglaciation, Science, 345, 1177–1180, 2014.

Buizert, C., Keisling, B., Box, J., He, F., Carlson, A., Sinclair, G., and DeConto, R.: Greenland-Wide Seasonal Temperatures During the Last 540    Deglaciation, Geophysical Research Letters, 45, 1905–1914, 2018.

Cuffey, K. M. and Clow, G. D.: Temperature, accumulation, and ice sheet elevation in central Greenland through the last deglacial transition, Journal of Geophysical Research: Oceans, 102, 26 383–26 396, 1997.

Cuffey, K. M. and Steig, E. J.: Isotopic diffusion in polar firn: implications for interpretation of seasonal climate parameters in ice-core records, with emphasis on central Greenland, Journal of Glaciology, 44, 273–284, 1998.

545    Cuffey, K. M., Clow, G. D., Alley, R. B., Stuiver, M., Waddington, E. D., and Saltus, R. W.: Large arctic temperature change at the Wisconsin-Holocene glacial transition, Science, 270, 455–458, 1995.

Dahl-Jensen, D., Mosegaard, K., Gundestrup, N., Clow, G. D., Johnsen, S. J., Hansen, A. W., and Balling, N.: Past temperatures directly from the Greenland ice sheet, Science, 282, 268–271, 1998.

Edwards, T., Fettweis, X., Gagliardini, O., Gillet-Chaulet, F., Goelzer, H., Gregory, J., Hoffman, M., Huybrechts, P., Payne, A., Perego, 550    M., et al.: Probabilistic parameterisation of the surface mass balance–elevation feedback in regional climate model simulations of the Greenland ice sheet, The Cryosphere, 8, 181–194, 2014.

Guillevic, M., Bazin, L., Landais, A., Kindler, P., Orsi, A., Masson-Delmotte, V., Blunier, T., Buchardt, S. L., Capron, E., Leuenberger, M., Martinerie, P., Prié, F., and Vinther, B. M.: Spatial gradients of temperature, accumulation and $\delta$ 18 O-ice in Greenland over a series of Dansgaard-Oeschger events., Climate of the Past, 9, https://doi.org/10.5194/cp-9-1029-2013, 2013.

555    Hakim, G. J., Emile-Geay, J., Steig, E. J., Noone, D., Anderson, D. M., Tardif, R., Steiger, N., and Perkins, W. A.: The last millennium climate reanalysis project: Framework and first results, Journal of Geophysical Research: Atmospheres, 121, 6745–6764, 2016.

Jouzel, J., Alley, R. B., Cuffey, K., Dansgaard, W., Grootes, P., Hoffmann, G., Johnsen, S. J., Koster, R., Peel, D., Shuman, C., et al.: Validity of the temperature reconstruction from water isotopes in ice cores, Journal of Geophysical Research: Oceans, 102, 26 471–26 487, 1997.

Kindler, P., Guillevic, M., Baumgartner, M. F., Schwander, J., Landais, A., and Leuenberger, M.: Temperature reconstruction from 10 to 120 560    kyr b2k from the NGRIP ice core, Climate of the Past, 10, 887–902, 2014.

Langen, P., Mottram, R., Christensen, J., Boberg, F., Rodehacke, C., Stendel, M., Van As, D., Ahlstrøm, A., Mortensen, J., Rysgaard, S., et al.: Quantifying energy and mass fluxes controlling Godthåbsfjord freshwater input in a 5-km simulation (1991–2012), Journal of Climate, 28, 3694–3713, 2015.

Langen, P. L., Fausto, R. S., Vandecrux, B., Mottram, R. H., and Box, J. E.: Liquid water flow and retention on the Greenland ice sheet in the 565    regional climate model HIRHAM5: Local and large-scale impacts, Frontiers in Earth Science, 4, 110, 2017.

Liu, Z., Otto-Bliesner, B., He, F., Brady, E., Tomas, R., Clark, P., Carlson, A., Lynch-Stieglitz, J., Curry, W., Brook, E., et al.: Transient simulation of last deglaciation with a new mechanism for Bølling-Allerød warming, Science, 325, 310–314, 2009.

McFarlin, J. M., Axford, Y., Osburn, M. R., Kelly, M. A., Osterberg, E. C., and Farnsworth, L. B.: Pronounced summer warming in northwest Greenland during the Holocene and Last Interglacial, Proceedings of the National Academy of Sciences, 115, 6357–6362, 2018.

570    Steen-Larsen, H. C., Masson-Delmotte, V., Sjolte, J., Johnsen, S. J., Vinther, B. M., Bréon, F.-M., Clausen, H., Dahl-Jensen, D., Falourd, S., Fettweis, X., et al.: Understanding the climatic signal in the water stable isotope records from the NEEM shallow firn/ice cores in northwest Greenland, Journal of Geophysical Research: Atmospheres, 116, 2011.

Vinther, B. M., Buchardt, S. L., Clausen, H. B., Dahl-Jensen, D., Johnsen, S. J., Fisher, D., Koerner, R., Raynaud, D., Lipenkov, V., Andersen, K. K., et al.: Holocene thinning of the Greenland ice sheet, Nature, 461, 385, 2009.

[Figure]

**Figure R1.** Slope ($^{\circ}$C ‰$^{-1}$) of the linear $\delta^{18}$O-temperature relationship for each grid cell. The $\delta^{18}$O-temperature relationship between grid cells is not shown. This is an example from one of the prior ensembles. For reference, open circles show the locations of ice-core records used in this study.

[Figure]

**Figure R2.** Covariance (°C ‰) of $\delta^{18}$O and temperature for each grid cell. The $\delta^{18}$O-temperature relationship between grid cells is not shown. This is an example from one of the prior ensembles. For reference, open circles show the locations of ice-core records used in this study.

[Figure]

**Figure R3.** The magnitude of the Bølling-Allerød warming in our main reconstruction. The time definition of this event is the mean of 14.55 to 14.35 ka minus the mean of 14.9 to 14.7 ka, which is the same as in Buizert et al. (2014). For reference, open circles show the locations of ice-core records used in this reconstruction.

[Figure]

**Figure R4.** The magnitude of the Younger Dryas cooling in our main reconstruction. The time definition of this event is the mean of 12.5 to 12.3 ka minus the mean of 13.5 to 13.3 ka, which is the same as in Buizert et al. (2014). For reference, open circles show the locations of ice-core records used in this reconstruction.

[Figure]

**Figure R5.** Magnitude of the Holocene warming in our main reconstruction. The time definition of this event is the mean of 11.2 to 11.0 ka minus the mean of 11.9 to 11.7 ka, which is the same as in Buizert et al. (2014). For reference, open circles show the locations of ice-core records used in this reconstruction.

[Figure]

**Figure R6.** Difference between the Younger Dryas and the Older Dryas in our main reconstruction. The time definition of this event is the mean of 12.5 to 12.0 ka minus the mean of 16.5 to 15.5 ka, which is the same as in Buizert et al. (2014). For reference, open circles show the locations of ice-core records used in this reconstruction.

[Figure]

**Figure R7.** Difference between the Younger Dryas and the Older Dryas in our S3 sensitivity reconstruction, which uses the definition $\delta^{18}O = 0.335T$. The time definition of this event is the mean of 12.5 to 12.0 ka minus the mean of 16.5 to 15.5 ka, which is the same as in Buizert et al. (2014). For reference, open circles show the locations of ice-core records used in this reconstruction.

[Figure]

**Figure R8.** Temperature (T) and temperature weighted by monthly precipitation (T*) from TraCE-21ka at Summit, Greenland. Both variables are shown as anomalies with respect to 1850-2000 CE and have been averaged to 50-year resolution. T* was computed before the anomaly was taken.

**Table R1.** The mean slope for the linear $\delta^{18}$O-temperature relationship used for the main reconstruction in this study (**black**) and the mean slope for the relationship used in the S4 sensitivity experiment in this study (red). We also include estimates from previous work, inlcuding, slopes found by Buizert et al. (2014) (purple) as estimated from their Fig. 3 for time periods between 20 and 10 ka; slopes found by Guillevic et al. (2013) (green) from their Table 3 for Dansgaard-Oeschger events 8, 9, and 10; slopes found by Kindler et al. (2014) (blue) from their Fig. 5 and estimated from their Fig. 6 for 120 to 10 ka; and slopes found by Cuffey and Clow (1997) (orange) for time periods between 50 and 0.5 ka. The cores are arraged from North (top) to South (bottom).

| Core Name | Slope Range (°C ‰$^{-1}$) | Slope Average (°C ‰$^{-1}$) |
|---|---|---|
| Agassiz | 0.618 - 0.656 | 0.640 |
| | 0.412 - 0.437 | 0.425 |
| Camp Century | 0.439 - 0.468 | 0.456 |
| | 0.293 - 0.312 | 0.304 |
| NEEM | 0.450 - 0.480 | 0.465 |
| | 0.300 - 0.320 | 0.310 |
| | ∼0.25 - ∼0.75 | ∼ 0.51 |
| | 0.51 - 0.63 | 0.57 |
| NGRIP | 0.454 - 0.489 | 0.470 |
| | 0.303 - 0.326 | 0.313 |
| | ∼0.3 - ∼0.4 | ∼0.37 |
| | 0.34 - 0.47 | 0.42 |
| | ∼0.3 - ∼0.57 | 0.52 |
| GISP2 | 0.442 - 0.493 | 0.467 |
| | 0.294 - 0.329 | 0.311 |
| | ∼0.1 - ∼0.3 | ∼0.26 |
| | 0.38 | |
| | 0.251 - 0.465 | 0.324 |
| GRIP | 0.442 - 0.493 | 0.467 |
| | 0.294 - 0.329 | 0.311 |
| | 0.49 | |
| Renland | 0.546 - 0.595 | 0.571 |
| | 0.364 - 0.397 | 0.381 |
| Dye3 | 0.424 - 0.475 | 0.444 |
| | 0.283 - 0.317 | 0.296 |

**Table R2.** Comparison of abrupt climate transitions in our main reconstruction and sensitivity reconstructions, S1, S2, S3, and S4. We use the same time definitions as in Buizert et al. (2014). Note that the main reconstruction and S4 do not warm as rapidly into the Holocene as S1, S2, and S3. Thus, our use of a single time definition may not allow us capture the full transition for all of these reconstructions.

| Core Name | Reconstruction Name | Bølling-Allerød Transition | Younger Dryas | Holocene Transition |
|---|---|---|---|---|
| Agassiz | Main ($\delta^{18}$O$= 0.67T^*$) | 12.83 | -9.10 | 11.64 |
| | S1 ($\delta^{18}$O$= 0.67T$) | 8.67 | -5.16 | 7.78 |
| | S2 ($\delta^{18}$O$= 0.5T$) | 11.20 | -6.96 | 10.50 |
| | S3 ($\delta^{18}$O$= 0.335T$) | 15.41 | -9.99 | 14.84 |
| | S4 ($\delta^{18}$O$= 0.67T^*$, stronger P seasonality) | 15.49 | -11.05 | 13.06 |
| Camp Century | Main | 11.71 | -8.51 | 10.53 |
| | S1 | 8.04 | -4.94 | 7.13 |
| | S2 | 10.40 | -6.60 | 9.65 |
| | S3 | 14.34 | -9.41 | 13.71 |
| | S4 | 14.07 | -10.36 | 11.46 |
| NEEM | Main | 10.17 | -7.53 | 9.10 |
| | S1 | 7.11 | -4.51 | 6.34 |
| | S2 | 9.22 | -5.99 | 8.33 |
| | S3 | 12.74 | -8.47 | 12.16 |
| | S4 | 12.18 | -9.20 | 9.71 |
| NGRIP | Main | 9.62 | -7.17 | 8.57 |
| | S1 | 6.76 | -4.33 | 6.03 |
| | S2 | 8.77 | -5.73 | 8.14 |
| | S3 | 12.13 | -8.09 | 11.57 |
| | S4 | 11.52 | -8.77 | 9.07 |
| GISP2 | Main | 7.78 | -6.03 | 6.88 |
| | S1 | 5.66 | -3.88 | 5.18 |
| | S2 | 7.39 | -5.06 | 6.92 |
| | S3 | 10.26 | -7.03 | 9.81 |
| | S4 | 9.27 | -7.41 | 7.03 |
| GRIP | Main | 7.78 | -6.03 | 6.88 |
| | S1 | 5.66 | -3.88 | 5.18 |
| | S2 | 7.39 | -5.06 | 6.92 |
| | S3 | 10.26 | -7.03 | 9.81 |
| | S4 | 9.27 | -7.41 | 7.03 |
| Renland | Main | 8.51 | -6.27 | 7.77 |
| | S1 | 5.93 | -3.78 | 5.39 |
| | S2 | 7.70 | -5.02 | 7.22 |
| | S3 | 10.64 | -7.08 | 10.22 |
| | S4 | 10.15 | -7.62 | 8.53 |
| Dye3 | Main | 6.45 | -5.64 | 5.41 |
| | S1 | 5.33 | -4.34 | 5.26 |
| | S2 | 7.05 | -5.45 | 6.84 |
| | S3 | 9.92 | -7.29 | 9.60 |
| | S4 | 7.62 | -7.10 | 4.79 |

**Table R3.** Comparison of abrupt climate transitions in our main reconstruction (black) and from Buizert et al. (2014) (purple). Uncertainties are given as standard deviations, and were computed using summation in quadrature. We use the same time definitions as in Buizert et al. (2014).

| Core Name | Bølling-Allerød Transition | Younger Dryas | Holocene Transition |
|---|---|---|---|
| Agassiz | $12.83 \pm 3.49$ | $-9.10 \pm 3.65$ | $11.64 \pm 3.46$ |
| Camp Century | $11.71 \pm 3.23$ | $-8.51 \pm 3.42$ | $10.53 \pm 3.24$ |
| NEEM | $10.17 \pm 3.01$ | $-7.53 \pm 3.2$ | $9.10 \pm 3.05$ |
| | $8.89 \pm 3.70$ | $-4.92 \pm 3.45$ | $8.41 \pm 2.97$ |
| NGRIP | $9.62 \pm 2.90$ | $-7.17 \pm 3.09$ | $8.57 \pm 2.95$ |
| | $11.18 \pm 5.08$ | $-8.10 \pm 4.89$ | $10.89 \pm 4.59$ |
| GISP2 | $7.78 \pm 2.77$ | $-6.03 \pm 2.96$ | $6.88 \pm 2.84$ |
| | $14.37 \pm 3.55$ | $-9.23 \pm 3.59$ | $12.46 \pm 3.00$ |
| GRIP | $7.78 \pm 2.77$ | $-6.03 \pm 2.96$ | $6.88 \pm 2.84$ |
| Renland | $8.51 \pm 2.55$ | $-6.27 \pm 2.70$ | $7.77 \pm 2.57$ |
| Dye3 | $6.45 \pm 3.84$ | $-5.64 \pm 4.10$ | $5.41 \pm 3.96$ |

---

## Author Response (AR1)

**Included in this document:**

**1) Reply to Referee Comments**

We thank all three referees for their time and insights. In the following, we address each of the referees' comments (in black) and, where applicable, include the edited section of the paper. Our replies are inline in blue and edited sections are indented. Please note that some edits were influenced by multiple referee comments.

**Referee #1**

**Referee #1 General Comment**

The authors present a new reconstruction of temperature and precipitation over Greenland covering the past 20 000 years using for the first time over such a long period a data assimilation technique successfully applied recently over the past millennia. The paper is very clear, justify nearly all the choices in a very rigorous way and provides comprehensive estimates of the uncertainties. I have thus no doubt that, in addition to the new reconstruction that can be used for instance to drive ice sheet models, this study opens new fields of application of data assimilation of multi-millennial timescales.

Thank you.

However, I consider that the impact of the choice of the prior is not enough discussed and this issue must be addressed before publication. If I understand well, the prior ensemble is made of 100 states obtained by averaging 50 years of model data. Those states are selected randomly over the full length of the simulation (line 175). This method is reasonable if the climate variations are weak, such as during the past millennia, but is it valid for very large changes as observed during the glacial interglacial periods? I may have missed something but, if I am right, a state obtained in the model in the late Holocene can be used to reconstruct the last glacial climate, which may be hard to justify. For instance, the authors argue that it is important to take into account the changes in seasonality of precipitation (e.g. line 233) but I wonder how this could be achieved by selecting model states that are coming from very different periods. I would suggest using as prior only years that are close to the period that is reconstructed so that only glacial states are used to reconstruct glacial climate for instance.

The reviewer's general comment broadly concerns our selection of the prior ensemble from the entire TraCE-21ka simulation rather than from specific time periods that align with the reconstruction time. This is an excellent point, which we have thought about carefully, but had not elaborated upon in the paper.

The reviewer states that, "If I understand well, the prior ensemble is made of 100 states obtained by averaging 50 years of model data. Those states are selected randomly over the full length of the simulation (line 175)." If we are understanding each other correctly, then yes, one state in the 100-member prior ensemble is an average over 50 years of the model data; these 100 states are selected randomly from the full length of the simulation. This implies that both glacial and Holocene states are likely to be contained within the same prior ensemble that is used to reconstruct all time steps over the last 20,000 years. To be clear, a prior ensemble could in principle contain only Holocene states; however, this is not the case for any of the ten prior ensembles we use in the paper. Thus, in reconstructing a time step in the glacial, for example, both glacial and Holocene states are part of the prior ensemble.

We agree with the reviewer that conditionally chosen prior ensembles would be preferable, but for the timescale under consideration this is not yet feasible. To explain the reasoning behind our choices in the paper, we elaborate on the pros and cons of four prior ensemble options we considered before deciding on the one that we use in this study (#4). In the revised supplementary information, we have included the following as Sect. S1:

S1 Prior ensemble considerations

Here we elaborate on the pros and cons of four prior ensemble options we considered before deciding on the one that we use in this study (#4).

1. For offline data assimilation (i.e., no information passed between assimilation time steps), a justifiable method for choosing the prior ensemble would be to use a 100-member ensemble of 20,000-year climate simulations. These climate simulations would be TraCE-21ka-like (i.e., results from fully-coupled GCMs at T31 resolution or higher), and have varied initial conditions, boundary conditions, and model physics. The prior ensemble for any assimilation time step would be taken from the same time step in the climate simulations, which would lead to a prior ensemble that varies smoothly in time and is a justifiable initial guess for the climate evolution over the past 20,000 years. Though this option is simple, it is not feasible because the computational cost of running even one TraCE-21ka-like simulation remains near computational limits.

2. Given that there is only one TraCE-21ka-like simulation, another method would be to select states from TraCE-21ka that are closest in time to the reconstruction time step. For example, if we were reconstructing the 50-year average centered on the year 5,000 CE, then we would select the 100 states from TraCE-21ka that are closest in time to 5,000 CE. Given that we are working with 50-year averages, this means we would select all the states between 7,500 and 2,500 CE. This method, which we call the "running-window" method, provides a prior that varies smoothly in time and is a justifiable initial estimate for the climate evolution.

For the running-window method, the variance of the prior ensemble would tend to be small. A prior with small variance would lead to underweighting of the proxy records during assimilation. To avoid this issue, we could use the well-accepted approach of inflating the prior variance (Anderson and Anderson, 1999). However, the use of inflation adds an additional tunable parameter; in this case, it would add an additional parameter per time step. Although inflation can, in principle, be constrained using the ensemble calibration ratio (computed for excluded proxies), we have too few proxy records to meaningfully constrain this parameter without overfitting.

In addition to estimating numerous inflation factors, the running-window method limits us to one estimate of the spatial covariance structure per time step. Thus, we have no way to quantify the uncertainty associated with the prior covariance structure. This could be fixed by expanding the running window and randomly selecting multiple prior ensembles; however, if the running window is expanded enough to create meaningfully different prior ensembles, then Holocene states will leak into glacial prior ensembles (and *vice versa*) and the method essentially becomes the method we use in the paper.

3. To reduce the number of inflation parameters, we could split TraCE-21ka into several distinct time periods. From these time periods, we would randomly select prior ensembles that are only used for the reconstruction of associated assimilation time steps. For example, if we split TraCE-21ka into glacial, transitional, and Holocene periods, then we'd make a glacial prior ensemble that is only used to reconstruct the glacial, a transitional prior ensemble that is only used to reconstruct the transition, and a Holocene prior ensemble that is only used to reconstruct the Holocene. This reduces the number of inflation factors we must estimate to a total of three. A disadvantage, however, is that this makes the prior discontinuous in time, which frequently leads to a discontinuous reconstruction. To adjust the reconstruction and make it continuous requires another source of information. Such post-processing adds an extra layer of complexity.

90      4. The method used in our study ensures a continuous reconstruction and removes the need for inflation factors. This
        method uses the same prior ensemble for all time steps (thus it is continuous) and the includes both glacial and Holocene
        states, which provides enough variance to appropriately weight the proxy records (thus no inflation is needed). Though
        the time-invariant prior is a poor estimate of the climate evolution over the last 20,000 years, the proxy records are given
        enough weight to result in a posterior that captures the large climate changes. In addition, we can quantify the uncertainty
95      associated with the spatial covariance pattern by producing multiple posterior ensembles that each stem from a different
        prior ensemble. In the paper, we use ten different prior ensembles to quantify this uncertainty. Overall, this method is
        both feasible and simple, thus providing a first step in developing paleoclimate data assimilation for applications on
        glacial-interglacial timescales.

100     In this study, one state in a 100-member prior ensemble is an average over 50 years of the model data; these 100 states
        are selected randomly from the full length of the simulation. This implies that both glacial and Holocene states are likely
        to be contained within the same prior ensemble that is used to reconstruct all time steps over the last 20,000 years. A
        prior ensemble could in principle contain only Holocene states; however, this is not the case for any of the ten prior
        ensembles we use in the paper. Thus, in reconstructing a time step in the glacial, for example, both glacial and Holocene
105     states are part of the prior ensemble.

We would also like to specifically address the following comment: "the authors argue that it is important to take into account
the changes in seasonality of precipitation (e.g. line 233) but I wonder how this could be achieved by selecting model states that
are coming from very different periods." We agree with the reviewer that the best way to account for changes in precipitation
110 seasonality is to do it in a time-varying manner, but we use a time-invariant prior for reasons given above. This means that
the mean precipitation seasonality is constant in time and determined by the states in our prior ensemble. What is new in our
paper is that for each reconstruction, we use ten different prior ensembles, which gives us ten different estimates of the mean
precipitation seasonality. In addition, our approach accounts for spatial variations in precipitation seasonality.

115 We do not wish to mislead readers into thinking that we account for time-varying precipitation seasonality. Instead, we
account for a mean precipitation seasonality, which is determined by the states in our prior ensemble. To clarify this, we have
edited lines 242-244 in the original paper (lines 293-302 in the revised paper) to say the following:

        With $T^*_{site}$ in our PSM, we find that the $\delta^{18}$O-$T_{site}$ slope is spatially variable (e.g., Fig. S5), ranging between 0.42 and
120     0.66 ‰$^{\circ}$C$^{-1}$ at the ice-core sites (Table S1), and tending to be less than the modern spatial relationship of 0.67 ‰$^{\circ}$C$^{-1}$
        at most locations around Greenland. Due to the data assimilation method outlined above, these slopes vary both in space
        and across iterations, the latter being due to the varying prior ensembles. These slopes do not vary in time in the prior,
        but they do in the posterior (note that $\delta^{18}$O-$T_{site}$ slopes mentioned throughout this paper refer to the prior ensemble).
        By using ten different prior ensembles, we capture the uncertainty in the $\delta^{18}$O-temperature relationship from variations
125     in the precipitation seasonality. These TraCE-21ka-derived estimates lie within the range of slopes estimated for sites
        around Greenland for a variety of time periods (Table S1). Differences seen in Table S1 reflect both the different methods
        used and the time period considered. Some estimates, such as Guillevic et al. (2013) and Buizert et al. (2014) are for
        abrupt transitions, such as Dansgaard-Oeschger events, while others find mean slopes over longer periods of time, such
        as Kindler et al. (2014) and this investigation.

130 **Referee #1 Specific Comment #1**

More specifically, still related to the prior, the authors explain (line 135) that 'For paleoclimate data assimilation, it is impor-
tant that the climate simulation capture a range of possible climate states over the time period of interest.' They should thus
first discuss the results of the TraCE-21ka simulation as it seems from Figure 12 that it underestimates the magnitude of the
changes. More generally, the authors do not discuss at all the biases of the climate model. They correct for biases in the modern
135 state by using anomalies compared to 1850-2000 (line 146) but this seems to be a small change compared to the signal during
the whole simulation (line 366). Besides, the response to forcing is very different between different models as illustrated by

the Paleoclimate Model Intercomparison Project. How this model behavior, which can also bias results for distant past, is influencing the results? Another way to phrase this point is that the model biases are not constant over time while the proposed correction assumes the stationarity of the biases.

The reviewer makes a good point that we should expand our discussion of the TraCE-21ka simulation, especially with respect to model biases.

Model-bias corrections rely on observations. In the modern, there are numerous observations from ground-based and satellite systems. From the past, there are relatively few observations, which are from proxy records. Given an assumption of stationary model bias (unchanging in time), we can use the modern observations to compute the bias correction; however, given an assumption of non-stationary model bias, we must rely on paleoclimate proxy records as well. In our paper, we assume a stationary model bias and apply the delta-change method (Teutschbein and Seibert, 2012). This leaves the proxy records available for data assimilation. Ideally, we would subsample the proxy records and use one subsample for data assimilation, another to correct for model bias *a priori*, and another to assess the influence of model bias on our reconstruction *a posteriori*. This, however, is not possible with a small number of proxy records. Therefore, we have chosen to reserve all proxy records for data assimilation and to assume a stationary bias correction.

In addition to a mean bias in the model, there may also be biases in the variance. The reviewer specifically points out this issue with TraCE-21ka: it has a small glacial-Holocene climate change relative to the ice-core records. This is a bias that is best addressed with information from proxy records, which we have reserved as independent observations for data assimilation.

As the reviewer alluded to, another opportunity to assess the influence of model bias is to simply select our prior ensemble from a different model. By examining how the results are affected by a variety of different model simulations, we could assess the sensitivity of our results to different models (and thus model biases). Doing this analysis in a rigorous manner is not yet possible because TraCE-21ka is the only-available continuous 20,000-year simulation completed with a fully-couple GCM at a T31 resolution or higher.

We have included this discussion as a paragraph in section 2.2 (lines 147-155 in the revised paper):

From TraCE-21ka, we use two-meter air temperature for temperature ($T$) and the sum of large-scale stable precipitation and convective precipitation for precipitation ($P$). To correct for model bias in TraCE-21ka, we assume that the bias is stationary in time and apply the delta-change method (Teutschbein and Seibert, 2012) by taking the anomaly of temperature and the fraction of precipitation relative to the mean of our reference period (1850-2000 CE). An assumption of a stationary model bias is required because, with a small number of proxy records, we cannot afford to subsample them for the purposes of bias correction, data assimilation, and evaluation. After the bias correction, we average the TraCE-21ka variables (which originially have monthly resolution) to 50-year resolution, as we did for the ice-core records. In this process, we average 50 consecutive years (600 months) such that no year (or month) is used in more than one 50-year average. This averaging results in 440 time steps spaced 50 years apart.

**Referee #1 Specific Comment #2**

Estimating the skill of the reconstruction compared to a constant prior (line 204) is a too low target for me. If the reconstruction was only showing a warming between the glacial period and the Holocene, it would already be skillful compared to this initial estimate and this does not require a very sophisticated technique. The skill of the reconstruction should be evaluated against the transient TraCE-21ka simulation to see if the data assimilation brings some skill compared to the simulation not constrained by data.

We agree with the reviewer that we should compare our reconstruction skill against that of other 20,000-year reconstructions or simulations. The skill metrics are most comparable if there is either a $\delta^{18}O$ or T* variable available in the reconstruction

or simulation. Thus, this comparison is straightforward with TraCE-21ka, but not for other reconstructions like that of Buizert
et al. (2018). We have computed the correlation coefficient, coefficient of efficiency (CE), and root mean square error (RMSE)
for TraCE-21ka, and have added this information both as text (in Sect. 3.2 of the revised paper) and figures (Figs. S8-S11).
These figures are also included in this document.

We have added the following text (to line 209 of the original paper and line 258 of the revised paper) to introduce our
evaluation against TraCE-21ka:

> For further comparison, we additionally compute the correlation coefficient, CE, and RMSE between the TraCE-21ka
> simulation and the proxy records.

We have also edited section 4.1 of the original paper (section 3.2 of the revised paper), which discusses the results of the
evaluation. Please see our reply to specific comment #6 for the edited text (lines 325 to 375 of this document).

**Referee #1 Specific Comment #3**

The authors explain at the end of the conclusion (line 485) that using a model that directly simulates isotopes would likely
improve their results. It would be interesting to discuss that earlier because, for instance, they mention a different relationship
between reconstructed precipitation and temperature at different time scales (line 352) but what is the potential role of a differ-
ent relationship between temperature and $\delta^{18}$O on this conclusion?

We agree with the reviewer that the isotope-temperature relationship is an interesting one to explore, a topic which we touch
on with our sensitivity experiments described in section 2.3.1 of the paper (lines 245-255 of the original paper and lines 303-
313 of the revised paper). We tested the sensitivity of our results to different isotope-temperature relationships, primarily the
magnitude of the slope in a linear relationship, but also the spatial pattern of that slope. The reconstructions resulting from
these experiments are discussed in Sect. 4.2 and Fig. 10 of the original paper (Sect. 3.3 and Fig. 9 of the revised paper).

An isotope-enabled model, as we mention in the conclusion, would provide another estimate of the magnitude and spatial
pattern of the isotope-temperature relationship. The advantage of an isotope-enabled model is that it incorporates the variety of
processes that can affect water isotope ratios, whereas, with our experiments, we focus on one primary process: precipitation
seasonality. Isotope-enabled models may have biases, and the experiments we perform are important for assessing the sensi-
tivity of reconstructions to the assumed isotope-temperature relationships. As the reviewer suggests, we have brought up this
discussion earlier in the paper by adding the following sentences to the end of section 2.3.1 (lines 314-316 of the revised paper):

> These sensitivity tests are equivalent to testing different assumptions about the $\delta^{18}$O-temperature relationship. The avail-
> ability of a 20,000 year-long isotope-enabled climate simulation would allow us to determine this relationship from
> model physics, which incorporate a variety of processes that can affect water isotopes, including precipitation seasonal-
> ity.

The reviewer also astutely suggests that we test the effect of the isotope-temperature relationship on the precipitation-
temperature relationship. We have analyzed the temperature-precipitation relationship between all possible combinations of
our main reanalysis and sensitivity reanalyses and have added the following paragraph to the paper (in section 4.1, lines 508-
516 of the revised paper). Table S3 is included in this document and has also been added to the Supplementary Information.

> We examine how the sensitivity experiments (Figs. 9 and 10) affect the scaling factor ($\beta$) in the precipitation-temperature
> relationship. We pair the five temperature reconstructions (main, S1-S4) and three precipitation reconstructions (low,
> moderate, and high) into fifteen possible combinations and conduct the same analysis as described above. Across these
> fifteen combinations, we find that the spatial pattern of $\beta$ is robust (Fig. 11a). The exact magnitude depends primarily
> on the temperature reconstruction and how cold it is in the glacial, with colder temperatures giving lower $\beta$ values.

To a lesser degree, the magnitude also depends on the precipitation reconstruction, with wetter scenarios giving lower $\beta$ values. As previously, we find that the low-pass filtered datasets have the same or nearly the same $\beta$ value as the unfiltered dataset, while the high-pass filtered datasets have consistently lower $\beta$ values. As an example of this, Table S3 shows the $\beta$ value found for the Kangerlussuaq region for all fifteen combinations and three filtering options.

**Referee #1 Specific Comment #4**

If I am right, when the technique described in section 2.3 is applied for the past millennia, records related to both the temperature and hydrology are assimilated together, as the covariance between the variables can bring interesting information and reduces the uncertainties. Here, it is claimed that having independent temperature and precipitation reconstructions is an advantage. This also means that precipitation and temperature changes could not be dynamically consistent in the proposed reconstruction? Maybe the authors do not want to rely on the covariance between those two variables as simulated by the climate model but they should explain why and, in that case, explain in a bit more detail the added value brought by the assimilation using this model results as prior.

The reviewer is correct that previous work (e.g., Hakim et al., 2016; Tardif et al., 2019) has used a similar method to assimilate both temperature and precipitation-sensitive proxy records into a reconstruction of multiple climate variables over the last millennium. We agree with the reviewer that our choice to separate the proxy records and reconstruct temperature and precipitation independently requires more explanation in the paper.

We have chosen to independently reconstruct temperature and precipitation because the relationship between these two variables is highly non-linear and non-stationary over the last 20,000 years. Precipitation generally follows the expected thermodynamic relationship on glacial-interglacial timescales (Robin, 1977); however, there have been times in the last 20,000 years, as shown by ice-core records, that precipitation has deviated significantly (even having opposite sign) from the thermodynamic expectation (Cuffey and Clow, 1997). For this reason, we choose not to impose the climate-model-derived mean temperature-precipitation relationship on our reconstruction.

We would also like to address the reviewer's concern that the temperature and precipitation may not be dynamically consistent given our method of reconstructing them independently. The reviewer is correct that our reconstructions may not achieve dynamic consistency through the modeled relationships. However, the reconstructions are dynamically consistent insofar as the empirical $\delta^{18}O$ and accumulation records from the ice-core records are dynamically consistent, as they must be.

We have made the following edits to the paragraph at lines 175-183 of the original paper (lines 213-232 of the revised paper) to reflect this discussion.

The prior ensemble is an initial estimate of possible climate states, which we form using 100 randomly-chosen 50-year averages from the TraCE-21ka simulation. States from both the glacial and the Holocene make up a prior ensemble. The same prior is used for all time steps in the reconstruction, leading to a prior that is constant in time. For a longer discussion on the reasoning behind our choice to use a constant prior, please refer to the Supplementary Information, Sect. S1. Proxy records are assimilated into the prior using Eq. 1, which produces the posterior ensemble, a new estimate of possible climate states. We assimilate $\delta^{18}O$ to reconstruct temperature and separately assimilate accumulation to reconstruct precipitation. This approach maintains independence between temperature and precipitation, which avoids imposing linearity and stationarity on the relationship between these two variables. As Cuffey and Clow (1997) show, not only is this relationship non-linear on long timescales but it is also not well-approximated by simple thermodynamic expectations. Separating these variables ensures that the relationship between temperature and precipitation is consistent with the empirical relationship between $\delta^{18}O$ and accumulation from ice cores, rather than being derived exclusively from the climate model.

We repeat the data assimilation process over multiple iterations, with each iteration using one of ten different 100-member prior ensembles and excluding one proxy record. Each of the ten prior ensembles is made up of a different random selection of 50-year averages from TraCE-21ka. Thus, each prior ensemble has a different variance and spatial covariance structure. Each proxy record is excluded from a total of ten iterations, where each of these iterations uses a different one of the ten prior options. Every iteration is uniquely identifiable by which prior ensemble is used and which proxy record is excluded. For a reanalysis, the total number of iterations is thus ten times the number of proxy records, such that for temperature we have 80 iterations and for precipitation we have 50 iterations. A reanalysis is a compilation of the 100-member posterior ensembles from these iterations, resulting in a temperature reanalysis having 8,000 ensemble members and a precipitation reanalysis having 5,000 ensemble members.

**Referee #1 Specific Comment #5**

Line 153, it is said that 'The offline method is appropriate when characteristic memory in the system is significantly shorter than the time step' (here 50 years). Is this valid here, for Bølling-Allerød and Younger Dryas events for instance?

The reviewer raises an excellent point that periods with strong forcing, such as the Bølling-Allerød and Younger Dryas, have a longer characteristic memory than periods with weaker forcing, such as the late Holocene. In general, models have little predictive skill on decadal or longer timescales, except perhaps for areas strongly influenced by the thermohaline circulation (Latif and Keenlyside, 2011, and references therein). During periods of strong forcing, model predictive skill may increase if both the forcing and the response are appropriately represented by the model; however, the predictive skill may also decrease if model uncertainty is large (Hawkins and Sutton, 2009). Rather than assuming our method is appropriate for all times in the last 20,000 years, we would ideally make use of an ensemble of long climate simulations or online data assimilation; however, both alternative approaches are not feasible at this time due to computational cost.

To clarify this point in the paper, we have edited the first paragraph of section 2.3 (lines 149-153 of the original paper, lines 179-190 of the revised paper) to say the following:

To combine the ice-core and climate-model data, we use an offline data assimilation method similar to that described in Hakim et al. (2016). If no covariance localization is used, as in this study, this method can be summed up as a linear combination of randomly-selected model states that are weighted according to new information provided by the proxy records. "Offline" refers to the absence of a forecast model that evolves the climate state between assimilation time steps. The offline method is appropriate when model predictive-skill is small given the assimilation time step (Hakim et al., 2016, and references therein). Model predictive-skill is generally poor on decadal to longer timescales (Latif and Keenlyside, 2011, and references therein) except possibly during times of strong forcing, such as the Bølling-Allerød (14.7-12.7 ka) and the Younger Dryas (12.7-11.7 ka) (Hawkins and Sutton, 2009). Because each of our time steps is an average over 50 years, as dictated by the resolution to which we average the proxy records, the offline method is appropriate except possibly during these large-forcing events. For these events, an online method may be appropriate (assuming that the models correctly capture both the forcing and the response); however, online ensemble data assimilation over glacial-interglacial cycles using a fully-coupled earth system model is impractical due to the computational cost and is beyond the scope of this study.

**Referee #1 Specific Comment #6**

Line 375. The reanalysis skill over the full period is clear compared to a constant climate but this would be informative to quantify it more precisely for the two selected 5000-year periods. Stating that it is lower than for the full period is not enough I think. From the figures 7 and 8, it seems that the CE is negative for nearly all the points. Stating line 377 that 'the reanalysis shows overall improvement over the prior ensemble' is also a weak conclusion as discussed in point 2.

We agree with the reviewer that the evaluation over the two 5,000-year periods warrants more discussion. We have included this discussion as well as the comparison of our reconstruction skill to that of TraCE-21ka (in reply to specific comment #2, lines 182 to 196).

325    3.2 Independent proxy evaluation

[revised manuscript text omitted]

**Referee #2**

The authors present new reconstructions of temperature and precipitation over the last 20,000 years over Greenland. For this, they apply a data assimilation technique on $\delta^{18}$O and accumulation records from Greenland ice cores and use the temperature and precipitation outputs from the TraCE-21ka simulation to extend the information to all the continent. The paper is in general clear enough for that people not having skills in data assimilation can read and understand quite easily the methodology presented in this manuscript. In my knowledge, the technique presented here is innovative for such a long period, and the different assumptions are presented and tested in a very rigorous way. This manuscript is worthy for publication in Climate of the Past, after having considered the comments below.

Thank you.

**Referee #2 General Comment #1**

Compared to the ice cores part, which is well discussed in the Methods section and in the Supplementary Material, the results of the TraCE-21ka simulation are not discussed enough in my opinion. More details about the similarities/differences with other PMIP simulations and/or climate reconstructions for PI and LGM could be discussed for example. How do the last 1000/2000 years fit well with last millennium simulations or reconstructions from isotopic proxies? Moreover, the rapid climate transitions are not so well captured by the TraCE-21ka, especially the Younger Dryas. This point should be discussed in terms of potential consequences for the reconstructions by data assimilation. The spatial resolution (T31 and 26 atmospheric vertical levels for the atmosphere if I am not wrong) should be clearly stated, and the uncertainties related to this aspect could be discussed (even if it is mentioned later). For example, is the limited number of grid points over Greenland a problem for the paleo DA technique? The ice sheet boundary conditions are also of major importance in this type of simulation. The expected differences if a more recent ice-sheet reconstruction would be prescribed could be discussed. Last point: what about the precipitation seasonality in TraCE-21ka over Greenland? Is it consistent with observations? Is the seasonality different for Holocene and glacial periods? I guess it could have an important impact on the $\delta$18O PSM and the final temperature reconstruction...

We agree with the referee that our discussion of the TraCE-21ka simulation is limited compared to our discussion of the ice-core records. This is primarily because TraCE-21ka is described extensively in the literature (e.g., Liu et al., 2009, 2012; He, 2011; He et al., 2013), whereas a number of aspects of the ice-core network, particularly the accumulation records, are novel or have been discussed little in previous work. In section 2.2 of the paper, we do discuss attributes of TraCE-21ka that are especially relevant to our method, including, the glacial to Holocene mean state changes, temperature and precipitation seasonality, and the ice-sheet boundary conditions. We agree with the referee that we are missing a statement about the spatial resolution of TraCE-21ka (indeed it is T31) and more details concerning seasonality and the ice sheets. We have included these details in the revised paper.

2.2 Climate-model simulation

We use TraCE-21ka, a simulation of the last 22,000 years of climate (22 ka to -.04 ka), which was run using the fully-coupled CCSM3 at T31 resolution (approximately 3.75 degrees horizontally) with transient ice-sheet, orbital, greenhouse gas, and meltwater flux forcings (Liu et al., 2009, 2012; He et al., 2013). For paleoclimate data assimilation, it is important that the climate simulation capture a range of possible climate states over the time period of interest. By design, TraCE-21ka captures the major glacial-to-Holocene temperature changes, as well as some of the short-term, rapid climate changes, such as the Bølling-Allerød transition (Liu et al., 2009). Many higher-resolution climate simulations are transient only over the last millennium (e.g., Bothe et al., 2015) or provide a snapshot of a certain time, such as last glacial maximum or the mid Holocene (e.g., Harrison et al., 2014). Individually, these millennial-length simulations have too little variability to capture the range of climate states across the glacial-interglacial transition. If combined, the biases in each simulation would need to be individually addressed, which is beyond the scope of this study.

From TraCE-21ka, we use two-meter air temperature for temperature ($T$) and the sum of large-scale stable precipitation and convective precipitation for precipitation ($P$). To correct for model bias in TraCE-21ka, we assume that the bias is stationary in time and apply the delta-change method (Teutschbein and Seibert, 2012) by taking the anomaly of temperature and the fraction of precipitation relative to the mean of our reference period (1850-2000 CE). An assumption of a stationary model bias is required because, with a small number of proxy records, we cannot afford to subsample them for the purposes of bias correction, data assimilation, and evaluation. After the bias correction, we average the TraCE-21ka variables (which originially have monthly resolution) to 50-year resolution, as we did for the ice-core records. In this process, we average 50 consecutive years (600 months) such that no year (or month) is used in more than one 50-year average. This averaging results in 440 time steps spaced 50 years apart.

TraCE-21ka includes changes in orbital forcing, which contribute to changes in the seasonality of temperature and precipitation. The strength of these seasonal cycles influences the mean-annual relationship between $\delta^{18}$O and temperature (Steig et al., 1994; Werner et al., 2000; Krinner and Werner, 2003). TraCE-21ka consistently shows stronger temperature and precipitation seasonality in the glacial (20 to 15 ka) than in the Holocene (5 to 0 ka) at each of the eight ice-core sites considered in this study (Fig. S3). Relative to the annual mean, the glacial had warmer and wetter summers and colder and drier winters. The Holocene also shows such a seasonal cycle; however, there is a smaller difference between the summers and winters. Any seasonal signal with wetter summers than winters will bias the $\delta^{18}$O towards summer values. According to TraCE-21ka, this effect is amplified in the glacial. As we discuss in Sect. 2.3.1, the particularly strong summer bias in the glacial affects the mean-annual $\delta^{18}$O-temperature relationship in ways that are consistent with findings from borehole thermometry at the GISP2, GRIP, and Dye3 ice-core sites (Cuffey and Clow, 1997; Jouzel et al., 1997).

In addition to a change in the strength of the seasonal cycle, TraCE-21ka shows a temporal shift, with summer temperature and precipitation peaking earlier in the glacial (around June and July) than in the Holocene (from July to September) (Fig. S3). In the glacial, both variables peak around June and July, with only two exceptions: precipitation peaks in August at the Renland and Dye3 ice-core sites. In contrast, Holocene temperature peaks slightly later, in July, while precipitation peaks even later, in August and occasionally September. Both variables and both time periods show winter minimums in February, again with the two exceptions of Renland and Dye3, which show later precipitation minimums. In this study, the timing of the seasonal peaks is relevant because it affects the precipitation-weighted temperature, defined in Eq. 7.

TraCE-21ka also includes prescribed transient ice sheets as a boundary condition. The transience is important for capturing the influence of elevation change on the ice-core records; however, the low horizontal resolution of TraCE-21ka leads to difficulties in capturing elevation changes at the edges of the ice sheet, in southern Greenland, and at coastal ice caps. In addition, the ice sheets in TraCE-21ka are independent of climate, updated only every 500 years during the simulation, and taken from ICE-5G (Peltier, 2004), a now outdated ice-sheet reconstruction (Roy and Peltier, 2018).

We want to emphasize that our paper represents the first attempt to use the ensemble Kalman filter approach to reconstruct climate over glacial-interglacial timescales. Ideally, there would be more 20,000-year (or longer) TraCE-21ka-like simulations (i.e., from fully-coupled GCMs at T31 resolution or higher). With other simulations from different models, we could address the influence of model bias, spatial resolution, boundary conditions, and initial conditions (for a discussion of model bias, see

our reply to Referee #1's specific comment #1, lines 141-174 of this document). If we had other simulations, then we agree with the referee that it would warrant a comparison between simulations and a discussion of how the differences affect our
465 results. With only one TraCE-21ka-like simulation available, we cannot conduct a meaningful comparison with how other climate-model simulations would affect our results. Our focus instead is on demonstrating that the method is feasible and skillful. Thus, we do not see it as productive to elaborate on how TraCE-21ka compares with PMIP or other simulations. In addition, the previous literature has extensively interrogated the results of the TraCE-21ka simulation (e.g., He, 2011; Buizert et al., 2014; Pedro et al., 2016; Zhang et al., 2017, 2018; Marsicek et al., 2018). We will add new text to refer the reader to this
470 previous work.

The referee additionally suggests that we include a discussion of how well TraCE-21ka captures rapid climate transitions, such as the Younger Dryas. We agree that this would be necessary if we were to choose our prior ensemble exclusively from time periods that are close to our reconstruction time (for examples of this, see our reply to Referee #1's general comment,
475 lines 29-129 of this document); however, our prior ensemble is time-invariant and thus our method is insensitive to the temporal evolution of TraCE-21ka (as long as the variance and spatial covariance structures of TraCE-21ka are preserved).

**Referee #2 General Comment #2**

The way how the prior ensemble is made should be clarified. To avoid misunderstanding, the authors should state at line 175 that the prior is constant in time (and not later at line 206). If I understand clearly, 10 different 100-member prior ensembles are
480 made (it should be said directly at the beginning). To form a prior ensemble, do you then randomly pick up 100 snapshots from the resampled TraCE-21ka temperature and precipitation outputs (see my minor comment for the line 146)? Or do you take randomly from the yearly TraCE-21ka outputs 50 consecutive years of data that you average in time for a member, other 50 consecutive years of model outputs for another time period for the second member, and so on. . .? Or another way? Anyway, it needs clarification (in the way of the section 2.3 of Hakim et al. 2016 for example). I have the same remark as the first referee:
485 what would be the difference if you would use, for instance, a "glacial prior" to reconstruct climate variables from glacial period and a "Holocene prior" for the warmer period instead of a constant prior for all the 20,000 years? In link with my first major remark, what would be the impact on the seasonality of precipitation, that influences the reconstructions of the authors?

We thank the referee for calling our attention to these points of confusion. We now state earlier in the paper that the prior is
490 constant in time (line 215 of the revised paper). We have also clarified how we average TraCE-21ka by making the following edit (around line 146 of the original paper and lines 152-155 of the revised paper):

After the bias correction, we average the TraCE-21ka variables (which originially have monthly resolution) to 50-year resolution, as we did for the ice-core records. In this process, we average 50 consecutive years (600 months) such that
495 no year (or month) is used in more than one 50-year average. This averaging results in 440 time steps spaced 50 years apart.

We have also reorganized and edited a paragraph in Sect. 2.3 (starting at line 175 of the original paper and line 213 of the revised paper) to clarify how we form our ten different 100-member prior ensembles. Please see lines 264-285 in this document
500 for the edits.

The referee additionally raises concerns over our random selection of the prior ensemble from the full TraCE-21ka simulation and the effect this has on the seasonality of precipitation and our reconstructions. For this we refer to our reply to the general comments from Referee #1 (lines 29 to 129 in this document).

505 **Referee #2 Minor Comments**

We thank the referee for these specific edits and the effort to make the paper clearer and more concise.

Line 14: "requires understanding its sensitivity to changes. . ."
Thank you. This edit has been be made in the revised paper.

510

Line 18: I would put into brackets the terms "and arid" and "and wet".
We think that it is important to equally emphasize the thermal and hydrologic differences between glacial and interglacial periods. For this reason, we have not make the suggested revision.

515    Line 31: the spatial resolution, especially for paleoclimate simulations, brings also uncertainties.
We have edited this to say: "In contrast, climate-model simulations are spatially-complete estimates of past climate, but they are subject to uncertainty due to model dynamics, boundary conditions, and spatial resolution."

Line 37 and passim: I think this is TraCE-21ka and not TraCE21ka.
520    We have made this change throughout the paper.

Paragraph lines 106-116: Does the matching of $\delta^{18}O$ from Dye3 to the $\delta^{18}O$ record from NGRIP bring a dependency when evaluating the posterior against Dye3 $\delta^{18}O$ record?
While dating uncertainty is non-zero, it is clear from multiple independent lines of evidence that the major $\delta^{18}O$ variations
525    in all Greenland cores are nearly synchronous. Most important is the variance, rather than the timing, of $\delta^{18}O$, and this is not affected by the minor changes to the timescales imposed here. (Note that our matching of the $\delta^{18}O$ from Dye3 to the $\delta^{18}O$ record from NGRIP is similar to the methods used to create the GICC05 depth-age scale and apply it to other ice cores.)

Section 2.2: I understand when you use the term "transient ice-sheet" that the prescribed ice-sheet is changed over time. But
530    some people can misunderstand and think that it is done dynamically with a coupled ice-sheet model. I would use an expression like "prescribed transient ice-sheet boundary conditions" for example.
We have changed the sentence to say: "TraCE-21ka also includes prescribed transient ice-sheets as a boundary condition."

Line 146: What is the initial temporal resolution of TraCE-21ka outputs? Monthly mean? When you talk about "average of
535    50-year resolution", do you mean "resampling" every 50 model years? If you take the last 20,000 years, it makes something like 400 time steps, right?
We thank the referee for calling our attention to this point of confusion. We have edited section 2.2 to clarify (see lines 427 to 430 of this document). The referee is correct that we end up with about 440 time steps after averaging. Our method is to take 50 consecutive years (600 months) of TraCE-21ka and to average them. No year (or month) is used in more than one 50-year
540    average.

Lines 230-232: Other model studies like Gierz et al. 2017 (JAMES) for the LIG and Cauquoin et al. 2019 (CP) for 6k-PI climates have shown that the seasonality of precipitation affects the $\delta^{18}O$-temperature relationship over Greenland.
We thank the referee for this comment. We did not mean to imply that Werner et al. (2000) is the only modeling study that has
545    shown that precipitation seasonality affects the $\delta^{18}O$-temperature relationship in Greenland. We have added the recommended citations.

Line 286: "that the proxy y and prior estimate of the proxy H(xb)".
Thank you. This edit has been made in the revised paper.

550
Lines 311-316: What does it give compared to the TraCE-21ka results?
We agree with the referee that it's important to compare our results to that of TraCE-21ka; however, we have made a point of saving comparisons with TraCE-21ka and Buizert et al. (2018) for the discussion (Sect. 4.3 of the revised paper).

555    Line 326: "from nearly +2 °C in northern. . ."
Thank you. This edit has been made in the revised paper.

Line 367: "has a large effect on our evaluation."
Thank you. This edit has been made in the revised paper.

560
Line 380: "the ECR is. . ."
Thank you. This edit has been made in the revised paper.

Line 404: the slower warming trends are hard to see. Make a zoom in the figure or give numbers.
565    Thank you for the suggestion. We have edited the figure (Fig. 10 in the original paper and Fig. 9 in the revised paper) to show a zoom-in on the Younger Dryas to Holocene transition. The edited version is included in this document.

Line 428: For S4 and high P cases, say clearly that it refers to the "sensitivity" curves on figure 12.
Thank you, we have clarified that "sensitivity" in the legend label refers to S4 and high P (Fig. 12 of the original paper and Fig.
570    13 of the revised paper).

Line 486: you can add the reference Okazaki and Yoshimura 2017 (CP).
We agree that this is a relevant citation and have added it to the revised paper.

575    Line 488: Add the references Cauquoin et al. 2019 (CP) and Okazaki and Yoshimura 2019 (JGR Atmos).
We agree that these are relevant citations and have added them to the revised paper.

Figure 4: add maybe contours for more clarity. And change the scale for the precipitation fraction at the peak warmth in the Holocene (panel b).
580    Thank you for the suggestion. We have added contours for clarification and changed the color scale in panel (b). The edited version is included in this document.

Figures 7, 8, S3, and S4: quite normal that the correlation is improved for the full period compared to the constant prior climate state.
585    Yes, we agree and have stated this in the paper (lines 205-209 of the original paper and lines 254-259 of the revised paper). If the referee thinks it is necessary, we can state it again later in the text or in the figure captions. We have additionally be compared the skill of our results to TraCE-21ka, as suggested by Referee #1.

**Referee #3**

Review of Badgeley et al. 2020 on Greenland paleo data assimilation
590
Badgeley et al. present temperature and precipitation fields for the last 20,000 years over Greenland generated using a paleo data-assimilation technique. This is an interesting and potentially very valuable new approach to investigating past climates. The paper is well written and clearly illustrated, and I am generally enthusiastic about the work.

595    Thank you.

While the methodology represents a big step forward, the paper is also a step backwards in other regards as it assumes a constant linear scaling of d18O to site temperature for all sites and periods based on the spatial d18O-T relationship. This assumption has been disproven in the last 2 decades through careful work in the ice core community (including some of the
600    papers cited here). This assumption will dominate all the spatial and temporal patterns in the temperature reconstructions,

and deserves more careful consideration than it is given here. The authors suggest that this problem is alleviated by using the precipitation weighted temperature, but they do not demonstrate this. Below I recommend some comparisons that should be done before the paper is suitable for publication.

605 My main concern is the use of a single linear d18O-T scaling based on the spatial d18O-T pattern at all sites and locations. While water isotopes are a valuable proxy, its temperature interpretation has proved very difficult. Borehole thermometry and d15N gas thermometry are the most reliable methods to get absolute (calibrated) temperature changes, and both suggest a d18O slope that is around half of the spatial relationship (0.67 permil/K) used here (as the authors acknowledge).

610 We agree with the referee that it would be overly-simplistic to assume "a constant linear scaling of $\delta^{18}$O to site temperature for all sites and periods based on the spatial $\delta^{18}$O-T relationship". This is not, however, what we do; we allow the $\delta^{18}$O-temperature relationship to vary spatially by relying on precipitation-weighted temperature ($T^*$) from TraCE-21ka. As we write in our paper (lines 229-232 of the original paper, lines 279-283 of the revised paper), "Numerous studies have suggested that precipitation seasonality is the largest source of nonlinearity in the $\delta^{18}$O-$T_{site}$ relationship (e.g., Steig et al., 1994; Pausata

615 and Löfverström, 2015); changes in precipitation seasonality are thought to be the primary reason that the effective $\delta^{18}$O-$T_{site}$ relationship for the glacial-interglacial transition has such a low slope (Werner et al., 2000; Gierz et al., 2017; Cauquoin et al., 2019)." We convert $T^*$ to $\delta^{18}$O using the equation $0.67T^* = \delta^{18}$O, and then compute the best-fit slope between $\delta^{18}$O and temperature to find their relationship. This effectively assumes a constant linear scaling between the *instantaneous* $\delta^{18}$O and site temperature, while allowing for changes in precipitation to affect the time-averaged relationship. These TraCE-21ka-derived

620 slopes vary between 0.42 and 0.66 ‰ °C$^{-1}$ at the core sites (Table S1), and are less than the modern spatial relationship of 0.67 ‰ °C$^{-1}$ at most locations around Greenland (e.g., Fig. S5). The table and figure have been included in the supplementary information.

The TraCE-21ka-derived slopes vary both in space and across prior ensembles (Fig. S5 and Table S1). By using ten dif-
625 ferent prior ensembles, we capture the uncertainty in the $\delta^{18}$O-temperature relationship, as determined by TraCE-21ka. We also examine a wider range of slope estimates in our sensitivity experiments. The slopes for S4 are shown in Table S1, and the slopes for S1, S2, and S3 are 0.67, 0.5, and 0.335 ‰ °C$^{-1}$, respectively. As described in our paper (lines 389-405 of the original paper and lines 446-464 of the revised paper), the magnitude of the slope affects the magnitude of the anomalies (Fig. 10 in the original paper and Fig. 9 in the revised paper). The spatial pattern of the slope also has an effect. For example, in the
630 early Holocene, the spatially-variable slopes result in a reconstruction that warms more slowly. In addition, the reconstructions with spatially-varying slopes show stronger north-south gradients than those with spatially-constant slopes. This north-south gradient shows up especially in the abrupt transitions, with larger changes in the north relative to the south (Table S2). In the paper, we had not discussed the impact of the spatially-varying slopes on the spatial pattern of the reconstructions; we will include this discussion in the revised paper by making the following revisions (Sect. 4.2, lines 400-405 of the original paper
635 and Sect. 3.3, lines 457-464 of the revised paper).

The temperature results are also sensitive to the spatial pattern of the $\delta^{18}$O-$T$ relationship. We find this by comparing the results from the S1-S3 scenarios that assume a spatially-uniform relationship to results from the main reanalysis and S4 scenario that assume a spatially-variable relationship. The S1-S3 scenarios have a characteristic shape to their time
640 series (Fig. 9), and, although the main reanalysis and S4 scenario generally fit this characteristic shape in the glacial and middle-late Holocene, in the early Holocene the main reanalysis and S4 diverge and show slower warming trends than the S1-S3 scenarios (Fig. 9b). In addition, the reconstructions with spatially-varying $\delta^{18}$O-$T$ relationships show stronger north-south gradients during times of abrupt temperature change than those with spatially-constant relationships (Table S2). These findings indicate that there is new information added by using a PSM that includes spatial variability in pre-
645 cipitation seasonality.

The reviewer is correct that we effectively assume that the $\delta^{18}$O-temperature relationship is constant in time. We could in principle account for temporal changes in the $\delta^{18}$O-temperature relationship by using a time-varying prior; however, this

method is complicated by discontinuities and/or extra assimilation parameters. We use the same prior ensemble for all time steps of the reconstruction, which avoids discontinuities in the reconstruction and does not require us to constrain extra assimilation parameters with so few proxy records. A consequence of this method is that the $\delta^{18}$O-temperature relationship, which is derived from the prior ensemble, is constant in time. Unfortunately, we are restricted in our ability to both use a time-varying prior and avoid the complications stated above until more long, fully-coupled climate simulations become available (see our reply to Reviewer #1, lines 29 to 129 in this document, for more details).

As the referee mentioned, borehole thermometry and $\delta^{15}$N gas thermometry are reliable methods to getting at the $\delta^{18}$O-temperature relationship; however, as we say on lines 233-237 of the original paper (lines 283-287 of the revised paper), we do not rely on these methods because borehole thermometry and $\delta^{15}$N gas thermometry are not available at all sites. Instead, we rely on the TraCE-21ka-derived relationships described above, which we compare to relationships found previously using borehole and $\delta^{15}$N gas thermometry (Table S1). It is known that the $\delta^{18}$O-temperature relationship varies temporally depending on the length of time considered and the date (Jouzel et al., 1997). Table S1 shows that indeed, different investigations have estimated a variety of slopes for a variety of time periods. Differences in the estimated slopes are likely to result from the method used in a particular investigation and the time period considered. Some estimates, such as Guillevic et al. (2013) and Buizert et al. (2014) are for abrupt transitions, such as Dansgaard-Oeschger events, while others find mean slopes over longer periods of time, such as Kindler et al. (2014) and our own paper. The slopes that we derive from TraCE-21ka mostly fall within the ranges found by previous studies, even though our slopes are estimated for a time period that is not addressed in the other investigations, the last 20,000 years.

We have included a fuller description and discussion of the slopes in revisions of Sect. 2.3.1:

With T$^*_{site}$ in our PSM, we find that the $\delta^{18}$O-T$_{site}$ slope is spatially variable (e.g., Fig. S5), ranging between 0.42 and 0.66 ‰$^\circ$C$^{-1}$ at the ice-core sites (Table S1), and tending to be less than the modern spatial relationship of 0.67 ‰$^\circ$C$^{-1}$ at most locations around Greenland. Due to the data assimilation method outlined above, these slopes vary both in space and across iterations, the latter being due to the varying prior ensembles. These slopes do not vary in time in the prior, but they do in the posterior (note that $\delta^{18}$O-T$_{site}$ slopes mentioned throughout this paper refer to the prior ensemble). By using ten different prior ensembles, we capture the uncertainty in the $\delta^{18}$O-temperature relationship from variations in the precipitation seasonality. These TraCE-21ka-derived estimates lie within the range of slopes estimated for sites around Greenland for a variety of time periods (Table S1). Differences seen in Table S1 reflect both the different methods used and the time period considered. Some estimates, such as Guillevic et al. (2013) and Buizert et al. (2014) are for abrupt transitions, such as Dansgaard-Oeschger events, while others find mean slopes over longer periods of time, such as Kindler et al. (2014) and this investigation.

I suspect this assumption will lead to underestimated temperature variability in the posterior. The authors should check this for the abrupt transitions at the three sites (GISP2, NEEM, NGRIP) where d15N-based temperature changes are known (Buizert et al. 2014).

We thank the referee for this suggestion. In Table S4 we compare the magnitudes of three abrupt transitions (warming into the Bølling-Allerød, cooling into the Younger Dryas, and warming out of the Younger Dryas) against the $\delta^{15}$N-derived temperature estimates from Buizert et al. (2014) at three locations. We note that these three sites are all in central and northern Greenland and are not necessarily representative of southern Greenland or the coastal ice caps (e.g., Agassiz and Renland). The comparison shows that our approach does not underestimate temperature variability as compared to Buizert et al. (2014), with some of our results showing larger changes and some showing smaller changes. The specifics of the comparison are dependent on the location and the time period. For example, our reconstruction shows greater variability than the $\delta^{15}$N-derived reconstruction at NEEM, but less variability at NGRIP and GISP2. In addition, the two reconstructions agree better during the Younger Dryas cooling at GISP2, but are in better agreement during the Younger Dryas and Bølling-Allerød warmings at

NEEM and NGRIP.

However, there is also a clear spatial gradient, as first noted by [Guillevic et al., 2013], a paper that should be cited and discussed. Guillevic observes that d18O changes are largest towards the north (i.e. NEEM), and smaller towards the south (i.e. Summit). However, the actual temperature changes have the opposite gradient – smallest in the north and largest in the south. This means that the d18O-T relationship has an enormous spatial gradient, from ~0.6 at NEEM to ~0.3 at Summit. The Guillevic temperature gradient is seen in many (all?) climate model simulations and should thus be considered very robust.

Thank you for the suggestion. We agree that the Guillevic et al. (2013) paper should be cited. We are aware of the discrepancy between the spatial gradient in the $\delta^{18}$O and in the $\delta^{15}$N-derived temperature; however, we do not agree that this discrepancy has been resolved. The Guillevic temperature gradient (larger temperature changes in the south than in the north) is based on four central to northern ice core records (GISP2 and GRIP, however, are at essentially the same site near Summit). Guillevic et al. (2013) find that the temperature changes at NGRIP and the Summit cores are statistically indistinguishable; thus most of their spatial pattern is driven by the difference between NEEM and NGRIP/GISP2/GRIP. We cannot know without more constraints whether this north-south pattern that appears between NEEM and these three other cores holds for other parts of Greenland, such as southern Greenland. As we show in Sect. S3 (Sect. S4 of the revised paper) of our supplementary information, a southern data point is key to reconstructing southern Greenland climate.

The referee notes the reproduction of Guillevic et al.'s north-south pattern by climate models as evidence for the interpretation that this pattern extends to southern Greenland. We acknowledge that this has been found in some climate simulations, for example, Buizert et al. (2014) show that the TraCE-21ka simulation has this pattern during the Bølling-Allerød transition. The model simulations that find this pattern have a particular forcing, such as abrupt changes in freshwater in the North Atlantic (e.g., Liu et al., 2009) or specific sea ice configurations (e.g., Li et al., 2010). Other models show that abrupt climate transitions can be forced by other mechanisms, such as gradual changes in freshwater forcing (e.g., Obase and Abe-Ouchi, 2019), which lead to different spatial patterns. We have added a discussion of the spatial pattern of abrupt temperature changes to the revised paper (see lines 746 to 797 in this document).

These patterns are such that when using a single constant slope (as the authors do), the larger temperature changes would appear to be in the north, as is indeed the case in their reconstructions (Fig 4a, 4c). However, the Guillevic result would actually suggest the opposite pattern in temperature. The authors need to plot the magnitude of abrupt climate warming in their reanalysis (either the 14.7 ka or 11.6 ka transition), and compare it to the d15N-based values. My hunch is that they will find the opposite pattern from the Guillevic result.

As noted above, the reviewer is not correct that we use a single slope for all ice-core sites. We use a spatially-varying slope (for an example, see Fig. S5). We will be sure to clarify this in the revised paper.

We have reviewed the spatial pattern of three abrupt transitions in our main reconstruction: warming into the Bølling-Allerød, cooling into the Younger Dryas, and warming out of the Younger Dryas, respectively. These results are shown in Table S4 for the eight ice core locations and are compared to results from Buizert et al. (2014). Our results show increasing variability to the north accross seven of the eight cores, while the Guillevic et al. (2013) and Buizert et al. (2014) results show increasing variability to the south across just three cores. So yes, the referee is correct that our results show the opposite pattern from the Guillevic et al. (2013) results; however, the reason for this difference is not due to a spatially-constant $\delta^{18}$O-tempreature relationship. Instead, our spatial pattern is a result of how the temporal information from the proxy records is spread throughout the domain via the spatial covariance pattern between temperature and precipitation-weighted temperature in the prior ensemble.

To show this, we have performed three additional experiments, O8, N3O5, and N3O5_BA, in which we have assimilated both $\delta^{18}$O and $\delta^{15}$N-derived temperature from Buizert et al. (2014) and tested a time-period specific prior ensemble. The experiments and their results are given in the following section that has been added to the paper. New figures that are referenced

in this section are included in this document.

**4.2 Spatial patterns during abrupt climate change events**

[revised manuscript text omitted]

It has also been documented that the d18O-T slope is strongly variable in time, changing by almost a factor of 2 [Kindler et al., 2014].

Please refer to lines 647-654 of this document for an explanation of why we use a method that has a constant $\delta^{18}$O-T relationship in time in the prior ensemble. We note, however, at the $\delta^{18}$O-T relationship is free to vary in time in the posterior ensemble.

It would be unreasonable to ask the authors to redo all the work abandoning a key assumption; rather I think they should do a careful comparison to d15N-based estimates of abrupt climate change to assess how well their method captures both the magnitude and spatial pattern of abrupt temperature changes – and the implications this may have for the LGM and Holocene optimum patterns shown in Fig 4a and 4c. Perhaps they can provide some suggestions for future work on ways to assimilate the d15N-based climate constraints directly.

We agree with the referee that it would be ideal to assimilate more than just $\delta^{18}$O records. Though including a large variety of proxy records is beyond the scope of this paper, we have done three new reconstructions to show how our results are affected by assimilating $\delta^{15}$N-derived temperature records from Buizert et al. (2014). Please see lines 746-797 of this document for more information on these new reconstructions.

If the reconstructed N-S temperature gradient during abrupt change is indeed opposite to the Guillevic gradient, this should be clearly stated in the abstract.

We will edit the relevant part of our abstract to say the following. Major edits are in *italics*. Note that some of the edits are in response to later comments (see lines 985-987).

Reconstructions of past temperature and precipitation are fundamental to modeling the Greenland Ice Sheet and assessing its sensitivity to climate. Paleoclimate information is sourced from proxy records and climate-model simulations; however, the former are spatially incomplete while the latter are sensitive to model dynamics and boundary conditions. Efforts to combine these sources of information to reconstruct spatial patterns of Greenland climate over glacial-interglacial cycles have been limited by assumptions of fixed spatial patterns and a restricted use of proxy data. We avoid these limitations by using paleoclimate data assimilation to create independent reconstructions of temperature and precipitation for the last 20,000 years. Our method uses *oxygen-isotope ratios of ice and accumulation rates* from long ice-core records and extends *this information* to all locations across Greenland using spatial relationships derived from a transient climate-model simulation. *Standard evaluation metrics for this method show that our results capture climate at locations without ice-core records. Our results differ from previous work in the reconstructed spatial pattern of temperature change during abrupt climate transitions; this indicates a need for additional proxy data and additional transient climate-model simulations.* We investigate the relationship between precipitation and temperature, finding that it is frequency dependent and spatially variable, suggesting that thermodynamic scaling methods commonly used in ice-sheet modeling are overly simplistic. Our results demonstrate that paleoclimate data assimilation is a useful tool for reconstructing the spatial and temporal patterns of past climate on timescales relevant to ice sheets.

The authors suggest that using precipitation-weighted temperatures alleviates the problems associated with using a linear d18O-T scaling. To validate this claim, at the very least they should show a comparison of the 21ka histories of TraCE 2m temperature and TraCE precipitation-weighted temperature at a key site (e.g. Summit), to show how different these two really

are. Ideally, they would show more clearly how this impacts the reconstructed magnitude of the abrupt climate change events (that are most strongly constrained by the d15N data).

We have included a figure in the Supplementary Information as the referee suggests (Fig. S4), which shows that there is a difference between the temperature and precipitation-weighted temperature. The figure is only for the grid cell closest to Summit, but this is the case for all locations around Greenland, though the magnitude of the difference varies by location.

Through our temperature sensitivity experiments, we have tested the impact that precipitation-weighted temperature has on our reconstructions (see lines 389-405 of the original paper or line 446-464 of the revised paper for a discussion of the results of our sensitivity experiments). We thank the reviewer for their suggestion that we discuss this in more detail. Previously in this reply (lines 625-645 of this document), we discussed the impact that a spatially-varying $\delta^{18}$O-temperature relationship has on our results, and provided a revised section of the paper. This is relevant because the spatial variations in the $\delta^{18}$O-temperature relationship are a result of using precipitation-weighted temperature.

**Referee #3 General Comments**

Please describe the data assimilation method in more general terms understandable to the non-initiated, so the reader won't have to track down the Hakim reference. Can we think of the posterior as a cleverly weighted sum of the randomly selected model timesteps put into the prior? Is there some relationship between the posterior and the 21ka climate simulation – for example, is the posterior solution for the LGM very similar to the TraCE simulation of the LGM? Is the posterior LGM solution strongly weighted towards LGM model years randomly selelcted in the prior?

The referee is correct that this data assimilation method can be thought of as "a cleverly weighted sum of the randomly selected model timesteps put into the prior" because the ensemble mean is indeed a weighted sum of the ensemble members *if* there is no covariance localization, as is the case in our paper. The referee is also correct that it can be determined whether the posterior mean for a time step in the glacial is more strongly weighted towards ensemble members that were selected from the glacial period in TraCE-21ka; however, these weights are not a direct output of our method.

Thank you for the suggestion to describe data assimilation in more general terms. We agree that it's a good idea to provide a summary of the method. We will make the following edit to the first paragraph in Sect. 2.3, which describes the data assimilation method. The edit is in *italics*.

To combine the ice-core and climate-model data, we use an offline data assimilation method similar to that described in Hakim et al. (2016). *If no covariance localization is used, as in this study, this method can be summed up as a linear combination of randomly-selected model states that are weighted according to new information provided by the proxy records.*

The TraCE simulation has quite a coarse grid I imagine? Please specify the exact resolution. I imagine it may even put multiple of the ice core sites in a single grid box. Perhaps the grid box resolution could be drawn onto figure 1? It seems that the spatial fields in Fig 4 are much smoother than the model would be. Did you apply smoothing or some other technique?

Yes, we agree that we should specify the exact resolution of TraCE-21ka. We have added this to Sect. 2.2 of the paper. The spatial resolution is T-31, or about 3.75 degrees, and the temporal resolution is monthly, but we average it to 50-year resolution (resulting in about 440 time steps).

We thank the reviewer for calling our attention to the fact that we forgot to say in the figure captions that the spatial fields are smoothed. We agree that it could be helpful to see the original resolution, so we have converted our plots back to the original

resolution.

How meaningful is it to use global climate simulations and constrain them only in Greenland? From a global perspective, Greenland is essentially a single location and the global climate field is not at all constrained. How well-behaved is the far-field response in the reanalysis? And does this somehow impact the reconstruction? I think doing this with global proxy databases (such as [Shakun et al., 2012] would be a great next step (beyond the scope of this paper of course).

Global reconstructions are beyond the scope of this paper, which focuses on reconstructing Greenland climate using proxy records from Greenland. We agree that a next step in applying paleoclimate data assimilation to galcial-interglacial timescales is to use a global proxy database to reconstruct global climate variables.

Seasonality is very briefly addressed, but it deserves more attention as it is an important climate parameter. Please specifically address seasonality in both the prior and posteriors. Will the reconstructions made available online have T and/or P seasonality in them, and if so, describe how this seasonality is derived. I imagine the seasonality of the posterior can be derived via the assimilation method?

We agree that seasonality deserves more attention. As we note in our reply to Referee #2, we have included a description of TraCE-21ka's seasonality in Sect. 2.2 of the paper. Seasonality is only used to compute $T^*$; it is in no other way included in the data assimilation process or results. Our results are mean-annual 50-year averages, which we state in the paper (line 369-370 of the revised paper). The referee is correct that climate variables for specific seasons or the seasonal cycle itself can theoretically be reconstructed using this data assimilation method.

The authors find an unusually late timing of the Holocene optimum around 5ka – much later than other ice-core based estimate from both d18O and melt layers. Looking at Fig 2, it appears that Camp Century (and perhaps Dye 3) are the only cores that suggest such timing, and since the temperature reanalysis is fully determined by ice core d18O, it follows that those two cores must be responsible for this timing (do you agree with this assessment?). However, as pointed out by [Vinther et al., 2009], these sites experience strong thinning in the first half of the Holocene, which will shift their apparent climatic optimum towards a later age (as early Holocene climatic warmth is masked by a cooler site temperature at higher elevation). Could the late (5ka) timing of the climatic optimum in your reanalysis be an artifact of the thinning history of the Greenland ice sheet? Please discuss briefly in the text.

We agree with the referee that the timing of the Holocene thermal maximum (HTM) in our reconstructions tends to be on the later end of the range of previous findings (see lines 317-325 of the original paper or lines 379-387 of the revised paper). We also agree that this signal appears to result from the Camp Century, Dye3, and perhaps the NEEM $\delta^{18}$O records. The referee mentions how thinning in the early Holocene (Vinther et al., 2009) would affect our results. As we state in lines 127 and 460 of the original paper (lines 131 and 615 of the revised paper), our reconstructions are for climate at the ice-sheet surface, meaning that they include the lapse-rate effect of changing surface elevation. This is opposed to climate reconstructions at a reference elevation. Thus, our reconstruction of a later HTM is consistent with previous findings of an earlier HTM (given a fixed reference elevation) (e.g., McFarlin et al., 2018) and early Holocene thinning (Vinther et al., 2009). This is something we have thought about in great detail, but decided not to include in the paper; however, as suggested by the referee, we will make the following edits to the discussion section(Sect. 5, lines 455-465 of the original paper and Sect. 4.3, lines 610-627 of the revised paper). New text is in *italics*.

An important distinction among various different paleoclimate reconstructions for Greenland is in the treatment of elevation changes. Any paleoclimate reconstruction from ice-core records is complicated by ice-sheet elevation changes. In Vinther et al. (2009), it is assumed that the climate history is the same at all locations around Greenland, and that any differences among the ice core paleotemperature records is a result of that elevation change. In B18, past elevation changes are assumed to be negligible. In our reconstruction, the impact of elevation change on the spatial covariances of temperature and precipitation is implicitly accounted for as part of the data assimilation methodology. Formally,

*our reconstruction is of surface climate, not climate at a fixed elevation. Consequently, our reanalysis may not be directly comparable to other paleoclimate reconstructions. For example, the HTM is commonly reconstructed as an early Holocene event in records that are at a fixed or nearly-fixed elevation. In our reanalysis, the early Holocene is cooler than the mid Holocene. Changes in the ice-surface elevation could account for this apparent discrepancy. Thinning in the early Holocene (Vinther et al., 2009) would result in a lowering of the ice surface and an apparent warming at the ice surface due to lapse rate effects. This warming signal would be captured in ice-core records. If the warming trend due to surface lowering occurs at the same time as an overall climate cooling, then the climate signal would be dampened or possibly reversed.*

*Our method depends on the accuracy of the climate-elevation relationships in our prior – i.e. in the TraCE-21ka climate model simulation, which probably does not capture such relationships with particularly high fidelity since the model resolution is low and the climate and ice-sheet models are not coupled. Future work could take advantage of the probabilistic relationships among accumulation, temperature, and surface elevation as simulated in fine-scale regional climate models (Edwards et al., 2014).*

The data assimilation is fully dependent upon the accuracy of the TraCE-21 climate model simulation in capturing Greenland climate. Therefore, the paper needs a short evaluation of how well this model actually simulates Greenland T and P in the modern day. The TraCE T and P fields should be compared to modern-day Greenland reconstructions thereof; I would recommend the works by Box et al. on this topic [Box, 2013; Box and Colgan, 2013; Box et al., 2009; Box et al., 2013], but general reanalysis products such as NCEP or ERA5 are suitable also.

*The statement that "data assimilation is fully dependent upon the accuracy of the TraCE-21ka climate model simulation in capturing Greenland climate" is incorrect. Our results are dependent only on the spatial covariance patterns of the temperature anomalies and fractional precipitation in TraCE-21ka (referenced to 1850-2000 CE) and the proxy records. For our use of TraCE-21ka, what matters is how well TraCE-21ka captures the spatial pattern and variance of past climate anomalies, which is unknown except by comparison with the ice-core data (which our method does implicitly). The fidelity of the modern-day climatology is not particularly relevant.*

All the figures show relative temperature changes and accumulation changes (relative to the reference period, which is not defined as far as I can tell). But when forcing ice sheet models absolute values are needed. Are these absolute values taken from the last time-slice of the TraCE simulations, or is something better used?

*Before we use TraCE-21ka in our assimilation, we subtract (for temperature) or divide (for precipitation) by the mean of TraCE-21ka over the period 1850-2000 CE. Before we assimilate each proxy record, we subtract (for $\delta^{18}$O) or divide (for accumulation rate) by the mean of that record over the same period, 1850-2000 CE. Thus, the reconstructions are of temperature anomalies and fractional precipitation, which are referenced to the 1850-2000 CE mean climate as recorded by the ice-core records. We thank the referee for pointing out that this was not entirely clear. We have stated this more clearly and earlier on in the revised paper (lines 133, 150, and each of the relevant figure captions in the revised paper).*

*Now that our reconstructions are complete, they may be turned into absolute values for applications such as ice-sheet modeling. This can be done by adding a 1850-2000 CE temperature climatology to the temperature fields and multiplying the precipitation fields by a 1850-2000 CE precipitation climatology (e.g., from Box, 2013). We do not do this in our paper.*

**Referee #3 Minor Comments**

L8: What are "independent ice core records"? d18O? Again, I think the reconstructions should be compared during the abrupt temperature transitions at NEEM NGRIP and GISP2, which is where d15N-N2 provides a very robust estimate of the magnitude of change. Those are the truly independent ice core records to compare to.

Our use of the term "independent ice-core records" refers to the comparison of our results against $\delta^{18}$O records that are excluded from that reconstruction iteration. This is explained later in the text around line 184 of the original paper and line 233 of the revised paper, so we will edit the abstract to clarify what is meant. See lines 823 to 837 of this document for these edits.

In our results and discussion we compare to findings from previous work; however, the referee is correct that we do not directly compare to temperature reconstructions derived from $\delta^{15}$N of N$_2$. We have now done this comparison, discussed it previously in this reply, and revised the paper by including a new section, Sect. 4.2 (see lines 746-797 of this document for the revisions).

L24: This is somewhat misleading, because you'll always need to do such precip corrections unless you are doing a fully coupled ice-climate simulation. As the ice elevation in the ice sheet simulation evolves, it differs from the reference elevation at which the climate field is defined; this needs to be corrected for via clausius-clapeyron or similar. So also with your forcing the ice sheet models will need to apply thermodynamic precip corrections.

We agree that ice-sheet models need to correct precipitation fields for changing surface elevation using some assumption about the relationship between precipitation and elevation. Our point, however, is about ice-sheet simulations that base their precipitation histories entirely on their temperature histories using a thermodynamic relation. Thus, the time series and spatial pattern the precipitation anomalies perfectly match those from the temperature fields, which we know to be false from ice-core records. We have tried to clarify this distinction in the revised paper:

Despite such evidence, paleo ice-sheet models typically assume precipitation fields that are parameterized in time using a thermodynamic relationship that is constant for all locations and timescales (e.g., Huybrechts, 2002; Greve et al., 2011).

L28: Many more d15N studies to cite here: [Guillevic et al., 2013; Kindler et al., 2014; Severinghaus and Brook, 1999; Severinghaus et al., 1998]

Thank you. We appreciate the recommended citations. We have included them throughout the revised paper where appropriate.

L38: "restricted to a single climate model realization"; wouldn't this critique apply to your study as well? It appears that both use the exact same climate model run.

Yes, the referee is correct, and we have removed the quoted language from the revised paper. Our current results rely on a single climate simulation; however, our method is easily generalizable to any climate simulation of the past 20,000 years. The method in Buizert et al. (2018) is also generalizable to other climate simulations as long as these simulations are accompanied by single-fociring experients.

L70: "measured layer thickness" is not really true. For several cores you use volcanic ties, in which case the layer thicknesses are not measured but inferred

We have changed the phrase to "the layer thickness".

L136: "captures the. . .." This is in the eye of the beholder. With the exception of the Bolling warming itself the TraCE run matches the abrupt transitions poorly – there is no YD to speak of.

We have reworded this sentence as follows: "By design, TraCE-21ka captures the major glacial-to-modern temperature change, as well as some of the short-term, rapid climate changes, such as the Bølling-Allerød transition (Liu et al., 2009)."

L145: why not use P-E? is evaporation negligible?

As we write later in the paper (lines 257-262 of the original paper and lines 318-323 of the revised paper), "accumulation is closely related to total precipitation at our ice-core sites". We demonstrate this with high-resolution model results (Langen et al., 2015, 2017) that show at most of the ice-core sites in Greenland, precipitation and accumulation are nearly equivalent, suggesting that evaporation is negligible. Dye3 is the only site where there is a difference between accumulation and precipitation, and this is due primarily to melt, not evaporation. This is why we extract P from TraCE-21ka, and use it, rather than P-E, for our proxy system model (described in Sect. 2.3.2).

L217: "highly correlated" is a strong statement. Do you have a reference? Normally d18O and site temperature are not highly correlated at most sites on observational time scales (< 0.5).

The referee is correct that the correlations are low on interannual timescales, but it is well established to be high on longer timescales (Jouzel et al., 1997).

L224: Based on the recent literature, I think that post-depositional alternation may be the largest complication in interpreting the d18O record. Please mention.

We disagree. As we note in the paper, diffusion in the firn column is irrelevant for our timescales of 50 years (Cuffey and Steig, 1998). The reviewer may be thinking of post-depositional processes involving water exchange between the snowpack and the atmosphere (e.g., Steen-Larsen et al., 2011), but it is not established that this has any significant effect on long-term relationships. Indeed, if anything, such processes improve the relationship between temperature and $\delta^{18}O$ as they tend to reduce the bias caused by the fact that snow does not accumulated continuously, but as discrete events.

L241: Can you plot $T_{site}$ and $T^*_{site}$ together for the last 21ka at a key site (e.g. Summit). That will let the reader judge the impact of using $T^*$ instead of T.

Thank you for the suggestion. We have done this in Fig. S4, which we have included in the supplementary information.

How is the seasonality of the posterior linked to the seasonality of the prior?

As we stated previously, we only use seasonality to compute $T^*$; it is in no other way included in the data assimilation process or results. Both our prior ensemble and our reconstructions are made up of mean-annual 50-year averages.

L261: "grid-cell closest to site" is this also done for T, or do you use 2D linear interpolation or similar? Are there cases where multiple cores share a closest grid cell?

Yes, this is also done for selecting which $T^*$ value to use in the $\delta^{18}O$ PSM. We have clarified that in the revised paper. At the resolution of TraCE-21ka, only the GISP2 and GRIP ice cores have the same closest grid cell.

L295: maybe a sentence on how this was estimated?

Thank you for bringing our attention to this point of confusion. We have clarifed this in the revised paper. We use the same method as was explained for $\delta^{18}O$.

L313-314: But [Dahl-Jensen et al., 1998] estimates it a lot colder at GRIP, more like -22K cooling at the LGM (25ka). This should be mentioned.

Dahl-Jensen et al. (1998) found a colder temperature at GRIP for the LGM, but the time period we discuss is 20-15ka, which is five to ten thousand years later, when the Dahl-Jensen et al. (1998) estimate shows that it has significantly warmed.

L340: Maybe reference [Buchardt et al., 2012] who did very detailed analyses of this.

Thank you. We are aware of this study, and have included it in our discussion of the temperature-precipitation relationship on shorter timescales (lines 484-494 of the revised paper).

L416: are other d18O records really independent? They suffer the same biases from seasonality, source effects, etc. For true independence, compare to d15N-N2.

We agree with the referee that it is important to compare our results to other types of proxy records. In this reply we have compared our temperature reconstructions to those from Buizert et al. (2014) and revised the paper.

L438: TraCE has no HTM anywhere! (one of its many problems...)

We have changed the wording to say, "TraCE-21ka has no obvious HTM in this location or any location around Greenland."

L476: This is more of a discussion than a conclusion item. Consider moving it. Also, see my comment above, the 5ka timing could be an artifact of ice sheet elevation changes.

We have moved this paragraph to the discussion (Sect. 4.3 of the revised paper). We have also included a discussion of elevation effects (see lines 919-950 of this document).

Figure 4: please add panels (e) and (f) with the T and P change over an abrupt transition (e.g. the Bolling onset). In panel (c), only show the cores that actually constrain the LGM (so not Agassiz, camp century and Renland). Why are the field so much smoother than the TraCE CCSM3 model resolution? Baffin bay has a large temp response with no cores to constrain it – can we trust this?

Thank you for these suggestions. We have added spatial plots of the abrupt temperature transitions (Figs. 12, S12, and S13). We have also included only the ice core locations that contribute to the reconstruction of each time period. As we mentioned previously in this reply, we have restored the plots to their original resolution because we agree with the reviewer that this is helpful information. We had originally smoothed the plots due to rendering issues, but we have fixed the previous problem.

The goal in using a method like data assimilation is to spread the information from point proxy locations to locations without proxy records. This allows us to reconstruct spatially-complete climate fields. With few proxies and many locations, the problem is underconstrained; however, we have shown that the method is skillful for some locations where proxies are not assimilated (see Sect. 4.1, lines 364-382 of the original paper and Sect. 3.2, lines 401-439 of the revised paper).

Fig 5: the "noise" in T (i.e. high frequency signals) at all core sites seem strongly correlated. How come? Could it be that the posterior is more or less reflecting the mean d18O of the various sites?

Each core has some influence on the reconstruction at every location. Thus, the time series at each location is a weighted sum of the time series of each core. This would indeed make the higher-frequency noise correlated at different locations.

1125     Fig 6: The largest features in the plot are not directly constrained by any cores. Do you trust these?

         For our reply, please see lines 1115-1118 in this document.

         Figs 7 and 8 are very technical and could be moved to the supplement.
1130
         These figures demonstrate how well our reconstruction performs at locations without any assimilated information. We think
         this is an important aspect of our paper, and we wish to keep these figures in the main text.

[Figure]

**Figure 4.** Spatial pattern of the reanalysis mean for temperature (panels (a), (c)) and precipitation (panels (b), (d)) with contours for clarity. (a) and (b) are averaged over 1,000 years around the peak warmth in the Holocene, 5.5-4.5 ka, while (c) and (d) are averaged over 5,000 years in the late glacial, 20-15 ka. Anomalies and fractions are with respect to the mean of 1850-2000 CE. Points show ice-core locations used for each reanalysis with closed circles indicating $\delta^{18}O$ records and open circles indicating accumulation records. Grey stars show the locations of the EGRIP ice-core site, Summit, and South Dome, which are referenced in Figs. 5 and 11.

[Figure]

**Figure 9.** The main temperature (T) reanalysis (ensemble mean and $5^{th}$ to $95^{th}$ percentile shading) and ensemble mean for four sensitivity scenarios, S1-S4. Each sensitivity scenario reflects a different assumption about precipitation seasonality, with S1-S3 assuming a spatially-uniform seasonality and S3-S4 assuming stronger seasonality than the main reanalysis. Anomalies are with respect to the mean of 1850-2000 CE. These time series are for the location closest to Summit, which is representative of the results around Greenland. Panel (a) shows the full time period, and panel (b) the Younger-Dryas through the Holocene, showing that S1, S2, and S3 warm more quickly than the main reconstruction and S4.

[Figure]

**Figure 12.** Spatial pattern of the abrupt cooling event into the Younger Dryas. Panel (a) shows results from experiment O8, assimilating all eight $\delta^{18}$O records, panel (b) shows results from experiment N3O5, assimilating all three $\delta^{15}$N-derived temperature records and the remaining five $\delta^{18}$O records (those that do not overlap with the $\delta^{15}$N sites), and panel (c) shows results from experiment N3O5_BA, which is similar to the N3O5 experiment except the prior ensemble is selected from the 1,000 years surrounding the Bølling-Allerød warming. Unfilled black circles show locations of assimilated $\delta^{18}$O records, while filled circles with white outlines show locations of assimilated $\delta^{15}$N-derived temperature records. Filled circles in panels (b) and (c) show the $\delta^{15}$N-derived temperature values as reported by Buizert et al. (2014) on the same color scale as the rest of the panel. The temporal definition of this event is the same as defined in Buizert et al. (2014).

[Figure]

**Figure S3.** A comparison of TraCE-21ka glacial and Holocene seasonality at each ice-core site considered in this study. The mean glacial (20 to 15 ka) seasonality is shown as solid lines and the mean Holocene (5 to 0 ka) seasonality is shown as dashed lines. Panel (a) shows the monthly temperature anomaly referenced to the annual mean. Panel (b) shows the fraction of total annual temperature that fell each month. For both panels, the reference line (black) shows no seasonal cycle. Both the magnitude and the timing of the seasonal cycle change between the glacial and the Holocene, with the glacial generally showing a stronger seasonal cycle and an earlier summer peak. ref = reference line, Ag. = Agassiz, C.C. = Camp Century, and Ren. = Renland.

[Figure]

**Figure S4.** Temperature (T) and temperature weighted by monthly precipitation (T$^*$) from TraCE-21ka at Summit, Greenland. Both variables are shown as anomalies with respect to 1850-2000 CE and have been averaged to 50-year resolution. T$^*$ was computed before the anomaly was taken.

[Figure]

**Figure S5.** Slope ($^\circ$C ‰$^{-1}$) of the linear $\delta^{18}$O-temperature relationship for each grid cell. The $\delta^{18}$O-temperature relationship between grid cells is not shown. This is an example from one of the prior ensembles. For reference, open circles show the locations of ice-core records used in this study.

[revised manuscript text omitted]

**Tables**

**Table S1.** The mean slope for the linear $\delta^{18}$O-temperature relationship used for the main reconstruction in this study (black, "main") and the mean slope for the relationship used in the S4 sensitivity experiment in this study (red, "S4"). We also include estimates from previous work, inlcuding, slopes found by Buizert et al. (2014) (purple, "B14") as seen in their Fig. 3 for five discontinuous time periods between 20 and 10 ka; slopes found by Guillevic et al. (2013) (green, "G13") from their Table 3 for Dansgaard-Oeschger events 8, 9, and 10; slopes found by Kindler et al. (2014) (blue, "K14") from their Fig. 5 and estimated from their Fig. 6 for 120 to 10 ka; and slopes found by Cuffey and Clow (1997) (orange, "C97") for time periods between 50 and 0.5 ka. The cores are arraged from North (top) to South (bottom).

| Core Name | Slope Range (°C ‰$^{-1}$) | Slope Average (°C ‰$^{-1}$) |
|---|---|---|
| Agassiz | main: 0.618 - 0.656 | 0.640 |
| | S4: 0.412 - 0.437 | 0.425 |
| Camp Century | main: 0.439 - 0.468 | 0.456 |
| | S4: 0.293 - 0.312 | 0.304 |
| NEEM | main: 0.450 - 0.480 | 0.465 |
| | S4: 0.300 - 0.320 | 0.310 |
| | B14: 0.25 - 0.76 | 0.50 |
| | G13: 0.51 - 0.63 | 0.57 |
| NGRIP | main: 0.454 - 0.489 | 0.470 |
| | S4: 0.303 - 0.326 | 0.313 |
| | B14: 0.29 - 0.41 | 0.36 |
| | G13: 0.34 - 0.47 | 0.42 |
| | K14: 0.3 - 0.57 | 0.52 |
| GISP2 | main: 0.442 - 0.493 | 0.467 |
| | S4: 0.294 - 0.329 | 0.311 |
| | B14: 0.11 - 0.30 | 0.25 |
| | G13: 0.38 | |
| | C97: 0.251 - 0.465 | 0.324 |
| GRIP | main: 0.442 - 0.493 | 0.467 |
| | S4: 0.294 - 0.329 | 0.311 |
| | G13: 0.49 | |
| Renland | main: 0.546 - 0.595 | 0.571 |
| | S4: 0.364 - 0.397 | 0.381 |
| Dye3 | main: 0.424 - 0.475 | 0.444 |
| | S4: 0.283 - 0.317 | 0.296 |

**Table S2.** Comparison of abrupt climate transitions in our main reconstruction and sensitivity reconstructions, S1, S2, S3, and S4. We use the same time definitions as in Buizert et al. (2014). Note that the main reconstruction and S4 do not warm as rapidly out of the Younger Dryas as S1, S2, and S3. Thus, our use of a single time definition may not allow us capture the full transition for all of these reconstructions.

| Core Name | Reconstruction Name | Bølling-Allerød warming | Younger Dryas cooling | Younger Dryas warming |
|---|---|---|---|---|
| Agassiz | Main ($\delta^{18}O= 0.67T^*$) | 12.83 | -9.10 | 11.64 |
|  | S1 ($\delta^{18}O= 0.67T$) | 8.67 | -5.16 | 7.78 |
|  | S2 ($\delta^{18}O= 0.5T$) | 11.20 | -6.96 | 10.50 |
|  | S3 ($\delta^{18}O= 0.335T$) | 15.41 | -9.99 | 14.84 |
|  | S4 ($\delta^{18}O= 0.67T^*$, stronger P seasonality) | 15.49 | -11.05 | 13.06 |
| Camp Century | Main | 11.71 | -8.51 | 10.53 |
|  | S1 | 8.04 | -4.94 | 7.13 |
|  | S2 | 10.40 | -6.60 | 9.65 |
|  | S3 | 14.34 | -9.41 | 13.71 |
|  | S4 | 14.07 | -10.36 | 11.46 |
| NEEM | Main | 10.17 | -7.53 | 9.10 |
|  | S1 | 7.11 | -4.51 | 6.34 |
|  | S2 | 9.22 | -5.99 | 8.33 |
|  | S3 | 12.74 | -8.47 | 12.16 |
|  | S4 | 12.18 | -9.20 | 9.71 |
| NGRIP | Main | 9.62 | -7.17 | 8.57 |
|  | S1 | 6.76 | -4.33 | 6.03 |
|  | S2 | 8.77 | -5.73 | 8.14 |
|  | S3 | 12.13 | -8.09 | 11.57 |
|  | S4 | 11.52 | -8.77 | 9.07 |
| GISP2 | Main | 7.78 | -6.03 | 6.88 |
|  | S1 | 5.66 | -3.88 | 5.18 |
|  | S2 | 7.39 | -5.06 | 6.92 |
|  | S3 | 10.26 | -7.03 | 9.81 |
|  | S4 | 9.27 | -7.41 | 7.03 |
| GRIP | Main | 7.78 | -6.03 | 6.88 |
|  | S1 | 5.66 | -3.88 | 5.18 |
|  | S2 | 7.39 | -5.06 | 6.92 |
|  | S3 | 10.26 | -7.03 | 9.81 |
|  | S4 | 9.27 | -7.41 | 7.03 |
| Renland | Main | 8.51 | -6.27 | 7.77 |
|  | S1 | 5.93 | -3.78 | 5.39 |
|  | S2 | 7.70 | -5.02 | 7.22 |
|  | S3 | 10.64 | -7.08 | 10.22 |
|  | S4 | 10.15 | -7.62 | 8.53 |
| Dye3 | Main | 6.45 | -5.64 | 5.41 |
|  | S1 | 5.33 | -4.34 | 5.26 |
|  | S2 | 7.05 | -5.45 | 6.84 |
|  | S3 | 9.92 | -7.29 | 9.60 |
|  | S4 | 7.62 | -7.10 | 4.79 |

**Table S3.** Scaling factors ($\beta$) for the temperature-precipitation relationship in the Kangerlussuaq region. The results for the main reanalysis are in bold.

| Temperature scenario | Precipitation scenario | No Filtering | Low-Pass (5,000 years$^{-1}$) | High-Pass (5,000 years$^{-1}$) |
|---|---|---|---|---|
| Main | Low | 0.09 | 0.09 | 0.04 |
| **Main** | **Moderate** | **0.08** | **0.08** | **0.04** |
| Main | High | 0.08 | 0.08 | 0.05 |
| S1 | Low | 0.12 | 0.11 | 0.05 |
| S1 | Moderate | 0.11 | 0.11 | 0.06 |
| S1 | High | 0.11 | 0.11 | 0.07 |
| S2 | Low | 0.09 | 0.09 | 0.04 |
| S2 | Moderate | 0.09 | 0.08 | 0.04 |
| S2 | High | 0.09 | 0.08 | 0.05 |
| S3 | Low | 0.06 | 0.06 | 0.03 |
| S3 | Moderate | 0.06 | 0.06 | 0.03 |
| S3 | High | 0.06 | 0.06 | 0.04 |
| S4 | Low | 0.07 | 0.07 | 0.03 |
| S4 | Moderate | 0.07 | 0.07 | 0.03 |
| S4 | High | 0.07 | 0.07 | 0.04 |

**Table S4.** Comparison of abrupt climate transitions in our main reconstruction (black) and from Buizert et al. (2014) (purple) values in the second row for NEEM, NGRIP, and GISP2). Uncertainties are given as standard deviations. We use the published values and uncertainties from Buizert et al. (2014). We use the same time definitions as in Buizert et al. (2014).

| Core Name | Bølling-Allerød warming | Younger Dryas cooling | Younger Dryas warming |
|---|---|---|---|
| Agassiz | $12.83 \pm 3.49$ | $-9.10 \pm 3.65$ | $11.64 \pm 3.46$ |
| Camp Century | $11.71 \pm 3.23$ | $-8.51 \pm 3.42$ | $10.53 \pm 3.24$ |
| NEEM | $10.17 \pm 3.01$ | $-7.53 \pm 3.2$ | $9.10 \pm 3.05$ |
|  | $8.9 \pm 1.2$ | $-4.8 \pm 0.6$ | $8.4 \pm 1.5$ |
| NGRIP | $9.62 \pm 2.90$ | $-7.17 \pm 3.09$ | $8.57 \pm 2.95$ |
|  | $10.8 \pm 1.0$ | $-10.9 \pm 1.0$ | $11.4 \pm 1.6$ |
| GISP2 | $7.78 \pm 2.77$ | $-6.03 \pm 2.96$ | $6.88 \pm 2.84$ |
|  | $14.4 \pm 0.95$ | $-9.2 \pm 0.45$ | $12.4 \pm 1.7$ |
| GRIP | $7.78 \pm 2.77$ | $-6.03 \pm 2.96$ | $6.88 \pm 2.84$ |
| Renland | $8.51 \pm 2.55$ | $-6.27 \pm 2.70$ | $7.77 \pm 2.57$ |
| Dye3 | $6.45 \pm 3.84$ | $-5.64 \pm 4.10$ | $5.41 \pm 3.96$ |

**2) Summary of Changes Made to the Manuscript**

Here we summarize the major changes made to the manuscript and Supplementary Information. All manuscript edits are shown in the marked-up version below.

1. In the Supplementary Information we added Sect. S1, which is a discussion of the pros and cons of four different prior ensemble options. The point of the section is to clarify our choice of a time-constant prior ensemble over time-period specific prior ensembles.

2. We expanded Sect. 2.2, "Climate-model simulation", to include a more thorough description of the TraCE-21ka simulation, especially on the topics of seasonality and prescribed ice sheet evolution. This change was made to provide more context for a major component of the data assimilation method: the climate simulation from which the prior ensemble is drawn.

3. In Sect. 2.3.1, we expanded our discussion of the $\delta^{18}$O-temperature slope as derived from the prior ensemble. Two new figures and one new table were added for visual clarification (Figs. S4 and S5; Table S1). These changes were made to clarify that the slopes we use are spatially-variable and that this is due to the inclusion of precipitation-weighted temperature (T$^*$) in the $\delta^{18}$O proxy system model (PSM).

4. We rearranged the results (Sect. 3) and discussion (previously Sect. 5, now Sect. 4) to increase clarity and flow of the paper. The evaluation section (previously Sect. 4) was combined with the results section (now Sects. 3.2 and 3.3). The discussion of the relationship between temperature and precipitation was moved from the results to the discussion section (it is now Sect. 4.1).

5. We added a new section (Sect. 4.2) describing three new experiments to show the impact of including $\delta^{15}$N-derived temperature records and a time-period specific prior ensemble from the Bølling-Allerød. Three new figures and one new table accompany this new section (Figs. 12, S12, and S13; Table S4). A new Supplementary Information section (Sect. S6) was also added to detail the methods of these new experiments.

6. Numerous minor edits were made in response to all three referees. These are described in the replies above.

**3) Marked-up Manuscript**

[revised manuscript text omitted]

---

## Author Response (AR2)

**Included in this document:**

1) Reply to Referee Comments (pages 1-3)

2) Summary of Changes made to the Manuscript (page 3)

3) Marked-up Manuscript (starts after page 4) *Note that the page numbers start over in this section.

5    4) References (page 4)

**1) Reply to Referee Comments**

We thank both referees for their time. In the following, we address referee comments (in black) and, where applicable, include the edited section of the paper. Our replies are inline in blue and edited sections are indented.

**Referee #1**

10   Referee #1 had no comments.

**Referee #2**

Review of the revised version of Badgeley et al: Greenland temperature and precipitation over the last 20,000 years using data assimilation

15   The authors have done an extremely detailed and comprehensive overhaul of their paper in response to the reviewer comments. I wish all authors were as responsive to reviewer reports!

They address most of my comments and remarks satisfactorily – in many cases actually going much beyond what I would have thought necessary.

I have a few minor comments that I would like the authors to address, after which the work is suitable for publication.

Regarding seasonality:

25   The authors clarified my confusion on this issue well. I had expected the approach to give seasonal temperatures in some way, which it does not. I think it would be important to emphasize this in the abstract and conclusion, by stating that they are reconstructing *annual mean* temperature and precipitation. Particularly for issues like the HTM (which I think of as a summer signal) this distinction is important.

30   Suggested changes:

Line 6: We avoid these limitations by using paleoclimate data assimilation to create independent reconstructions of *annual mean* temperature and precipitation for the last 20,000 years.

35   Line 630: Our approach, combining ice-core records with a climate-model simulation, provides complete spatial reanalyses of both *annual mean* temperature and precipitation covering the last 20,000 years.

We are glad to have clarified the confusion and are happy to include more emphasis on this point as suggested. We have edited lines 6 and 630 to say "mean-annual temperature and precipitation".

Regarding abrupt climate change:

One of my main concerns was the spatial pattern of abrupt climate change, and the authors have done a great job of both visualizing this, and providing the patterns in the supplementary tables. Thanks for this. The additional tests in Section 4.2 clearly show that the water isotopes want to push the N-S abrupt climate change spatial pattern in the opposite direction from the d15N.

Thank you. The reviewer makes a good point that the $\delta^{18}O$ result has a stronger North-South pattern (greater changes in the North), than if the $\delta^{15}N$-derived temperature records are included. We find, however, that this impact is much less important than the choice of prior ensemble. We have included this clarification in Sect. 4.2 (see lines 70 to 93 of this response).

I think the section could use a stronger discussion at the end of this section.

As I stated before, all GCM studies I am aware of agree with the d15N pattern. Their suggestion that this is an artifact of the GCM forcing is somewhat unfair – I am not aware of a single model study that has the largest abrupt D-O magnitude in North Greenland (and I've looked at a many). The paper that they cite as an example of an alternative forcing (Obase and Abe-Ouchi 2019) actually has the largest abrupt temperature change in South Greenland also (See their fig 4a, lower right panel. The colors are saturated in red, but the contour lines clearly show that the abrupt warming is greatest over the N-Atlantic, and that the largest abrupt Greenland warming occurs in the south).

I would encourage just a simple statement of the fact that the spatial abrupt change pattern of the reanalysis does not match that seen in d15N or in climate models simulations. Based on the reliability of d15N as an abrupt climate change proxy, and the overwhelming GCM agreement on this point, I think it would be fair to state that this single aspect of the d18O-based reanalysis is probably unreliable.

The referee is correct that the Obase and Abe-Ouchi (2019) agrees with larger changes in Southern Greenland. We have heeded the referee's suggestion to remove the discussion on model forcing and have replaced it with a simpler conclusion. We have edited the last three paragraphs of Sect. 4.2 (lines 539 to 561 of the revised paper) to say the following. New text is in *italics*.

For both the "O8" and "N3O5" experiments, the spatial pattern for the abrupt climate change events are similar to our main reanalysis, with the largest magnitude temperature changes in northern and northwestern Greenland, decreasing magnitude with decreasing latitude, and slightly larger change in the central east and southeast than corresponding western regions. For example, the spatial pattern of the Younger Dryas cooling is nearly the same regardless of which grouping of records is assimilated (Figs. 12a and b). This overall finding is robust to different combinations of these proxy records; for example, if we assimilate just the three $\delta^{15}N$-derived temperature records and no $\delta^{18}O$ records (results not shown) the pattern is not substantially changed. This pattern of temperature change differs from spatial patterns inferred previously from $\delta^{15}N$ for various abrupt climate events (Guillevic et al., 2013; Buizert et al., 2014); however, the O8 and N3O5 experiments show that these differences are not due to the assimilation of $\delta^{18}O$ rather than $\delta^{15}N$-derived temperatures. *The replacement of $\delta^{18}O$ records with $\delta^{15}N$-derived temperature records does not change the overall spatial pattern of these abrupt events, though it does result in a reconstruction with slightly larger temperature changes in the south and slightly smaller changes in the north. This effect, however, is less important than the choice of prior ensemble, as is shown by the third experiment, N3O5_BA.*

*For N3O5_BA, we restrict the prior ensemble to the Bølling-Allerød warming, which produces a reconstructed spatial pattern similar to those reported by Buizert et al. (2014) for each of the three abrupt climate events.* In TraCE-21ka, the Bølling-Allerød warming is forced by a sudden termination of freshwater forcing in the North Atlantic (Liu et al., 2009).

This forcing leads to large temperature fluctuations in southern Greenland that decrease with increasing latitude. With this covariance pattern dominating the prior ensemble, the N3O5_BA reconstruction indeed shows the largest temperature changes in southern Greenland, followed by central and then northern Greenland (Figs. 12c, S12c, and S13c).

90 *These experiments suggest that current ice-core records are insufficient to place a strong constraint on the spatial pattern of abrupt climate events. Reconstructions of these events may be improved with additional data, especially from southern Greenland (Sect. S4), and with prior ensembles that are designed to sample over uncertainty in the forcing and boundary conditions unique to these events.*

95 And yes, another ice core in South Greenland would be highly desirable.

Figures S12 and S13 are identical, as far as I can tell.

The figures appear very similar, but are in fact different. We have added a sentence to the Supplementary Information (lines
100 215 to 218) pointing out that the reconstructions of these two warming events are nearly identical, but that we include them both for completeness.

**2) Summary of Changes Made to the Manuscript**

We made three primary changes to the manuscript:

1. In the abstract and conclusion, we clarified that we reconstruct *mean-annual* temperature and precipitation.
105 2. We revised the discussion in Sect. 4.2.
3. We fixed the contours on Fig. 4, which had previously been offset to the southwest.

All text edits made to the manuscript are shown in the marked-up version below.

**3) Marked-up Manuscript**

**References**

[revised manuscript text omitted]

---

## Author Response (AR3)

**Included in this document:**

1)  Summary of Changes made to the Manuscript (page 1)

2)  Marked-up Manuscript (starts after page 1) *Note that the page numbers start over in this section.

**1) Summary of Changes Made to the Manuscript**

5   We made formatting changes according to the Climate of the Past manuscript preparation guidelines. We also added a pre-assigned DOI for the data. The data DOI, however, will not go live until the data has finished the submission process at the Arctic Data Center. All text edits made to the manuscript are shown in the marked-up version below.

**2) Marked-up Manuscript**

[revised manuscript text omitted]